# HOW I LEARNED TO STOP WORRYING AND LOVE RETRAINING

**Max Zimmer**[1], **Christoph Spiegel**[1] **& Sebastian Pokutta**[1,2]
[1]Department for AI in Society, Science, and Technology, Zuse Institute Berlin, Germany
[2]Institute of Mathematics, Technische Universität Berlin, Germany
`{zimmer,spiegel,pokutta}@zib.de`

## ABSTRACT

Many Neural Network Pruning approaches consist of several iterative training and pruning steps, seemingly losing a significant amount of their performance after pruning and then recovering it in the subsequent retraining phase. Recent works of Renda et al. (2020) and Le & Hua (2021) demonstrate the significance of the learning rate schedule during the retraining phase and propose specific heuristics for choosing such a schedule for IMP (Han et al., 2015). We place these findings in the context of the results of Li et al. (2020) regarding the training of models within a fixed training budget and demonstrate that, consequently, the retraining phase can be massively shortened using a simple linear learning rate schedule. Improving on existing retraining approaches, we additionally propose a method to adaptively select the initial value of the linear schedule. Going a step further, we propose similarly imposing a budget on the initial dense training phase and show that the resulting simple and efficient method is capable of outperforming significantly more complex or heavily parameterized state-of-the-art approaches that attempt to sparsify the network during training. These findings not only advance our understanding of the retraining phase, but more broadly question the belief that one should aim to avoid the need for retraining and reduce the negative effects of 'hard' pruning by incorporating the sparsification process into the standard training.

## 1 INTRODUCTION

Modern Neural Network architectures are commonly highly over-parameterized (Zhang et al., 2016), containing millions or even billions of parameters, resulting in both high memory requirements as well as computationally intensive and long training and inference times. It has been shown however (LeCun et al., 1989; Hassibi & Stork, 1993; Han et al., 2015; Gale et al., 2019; Lin et al., 2020; Blalock et al., 2020) that modern architectures can be compressed dramatically by *pruning*, i.e., removing redundant structures such as individual weights, entire neurons or convolutional filters. The resulting *sparse* models require only a fraction of storage and floating-point operations (FLOPs) for inference, while experiencing little to no degradation in predictive power compared to the *dense* model. Although it has been observed that pruning might have a regularizing effect and be beneficial to the generalization capacities (Blalock et al., 2020), a very heavily pruned model will normally be less performant than its dense (or moderately pruned) counterpart (Hoefler et al., 2021).

One approach to pruning consists of removing part of a network's weights from the model architecture after a standard training process, seemingly losing most of its predictive performance, and then retraining to compensate for that pruning-induced loss. This can be done either One Shot, that is pruning and retraining only once, or the process of pruning and retraining can be repeated iteratively. Although dating back to the early work of Janowsky (1989), this approach was most notably proposed by Han et al. (2015) in the form of ITERATIVE MAGNITUDE PRUNING (IMP). In its full iterative form, for example formulated by Renda et al. (2020), IMP can require the original train time several times over to produce a pruned network, resulting in hundreds of retraining epochs on top of the original training procedure and leading to its reputation for being computationally impractical (Liu et al., 2020; Ding et al., 2019; Hoefler et al., 2021; Lin et al., 2020; Wortsman et al., 2019). This as well as the belief that IMP achieves sub-optimal states (Carreira-Perpinán & Idelbayev, 2018; Liu et al., 2020) is one of the motivating factors behind methods that similarly start with an initially dense

model but incorporate the sparsification into the training. We refer to such *dense-to-sparse* methods as *pruning-stable* (Bartoldson et al., 2020).

Motivated by recent results of Li et al. (2020) regarding the training of Neural Networks under constraints on the number of training iterations, we challenge these commonly held beliefs by rethinking the retraining phase of IMP within the context of *Budgeted Training* and demonstrate that it can be massively shortened by using a simple linearly decaying learning rate schedule. We further demonstrate the importance of the learning rate scheme during the retraining phase and improve upon the results of Renda et al. (2020) and Le & Hua (2021) by proposing a simple and efficient approach to also choose the initial value of the learning rate, a problem which has not been previously addressed in the context of pruning. We also propose likewise imposing a budget on the initial dense training phase of IMP, turning it into a method capable of efficiently producing sparse, trained networks without the need for a pretrained model by effectively leveraging a cyclic linear learning rate schedule. The resulting method is able to outperform significantly more complex and heavily parameterized state-of-the-art approaches, that aim to reach pruning-stability at the end of training by incorporating the sparsification into the training process, while using less computational resources.

**Contributions.** The major contributions are as follows:

1. We empirically find that the results of Li et al. (2020) regarding the Budgeted Training of Neural Networks apply to the retraining phase of IMP, providing further context for the results of Renda et al. (2020) and Le & Hua (2021). Building on this, we find that the runtime of IMP can be drastically shortened by using a simple linear learning rate schedule with little to no degradation in model performance.

2. We propose a novel way to choose the initial value of this linear schedule without the need to tune additional hyperparameters in the form of ADAPTIVE LINEAR LEARNING RATE RESTARTING (ALLR). Our approach takes the impact of pruning as well as the overall retraining time into account, improving upon previously proposed retraining schedules on a variety of learning tasks.

3. By considering the initial dense training phase as part of the same budgeted training scheme, we derive a simple yet effective method in the form of BUDGETED IMP (BIMP) that can outperform many pruning-stable approaches given the same number of iterations to train a network from scratch.

We believe that our findings not only advance the general understanding of the retraining phase, but more broadly question the belief that methods aiming for pruning-stability are generally preferable over methods that rely on 'hard' pruning and retraining both in terms of the quality of the resulting networks and in terms of the speed at which they are obtained. We also hope that BIMP can serve as a modular and easily implemented baseline against which future approaches can be realistically compared.

**Outline.** Section 2 contains a summary of existing literature and network pruning approaches. It also contains a reinterpretation of some of these results in the context of Budgeted Training as well as a technical description of the methods we are proposing. In Section 3 we will experimentally analyze and verify the claims made in the preceding section. We conclude this paper with some relevant discussion in Section 4.

## 2 PRELIMINARIES AND METHODOLOGY

While the sparsification of Neural Networks includes a wide variety of approaches, we will focus on the analysis of *Model Pruning*, i.e., the removal of redundant structures in a Neural Network. We focus on performing *unstructured* pruning, that is the removal of individual weights, further also providing experiments for its *structured* counterpart, where entire groups of elements, such as convolutional filters, are removed. We will also focus on approaches that follow the *dense-to-sparse* paradigm, i.e., that start with a dense network and then either sparsify the network *during* training or *after* training, as opposed to methods that *prune before training* (e.g. Lee et al., 2019) or *dynamic sparse training* methods (e.g. Evci et al., 2020) where the networks are sparse throughout the entire training process. For a full and detailed survey of pruning algorithms we refer the reader to Hoefler et al. (2021).

Pruning-unstable methods are exemplified by ITERATIVE MAGNITUDE PRUNING (IMP) (Han et al., 2015). In its original form, it first employs standard network training, adding a common $\ell_2$-regularization term on the objective, and then removes all weights from the network with magnitude below a certain threshold. The network at this point commonly loses some or even all of its learned predictive power, so it is then retrained for a fixed number of epochs. This prune-retrain cycle is usually repeated a number of times; the threshold at every pruning step is determined as the appropriate percentile such that, at the end of a given number of iterations, a desired target sparsity is met. Renda et al. (2020) suggested the following complete approach: train a network for $T$ epochs and then iteratively prune 20% percent of the remaining weights and retrain for $T_{rt} = T$ epochs until the desired sparsity is reached. For a goal sparsity of 98% and $T = 200$ original training epochs, the algorithm would therefore require 18 prune-retrain-cycles for a massive 3800 total retrain epochs.

## 2.1 RETHINKING RETRAINING AS BUDGETED TRAINING

There has been some recent interest in the learning rate schedule used during retraining. The original approach by Han et al. (2015) is commonly referred to as FINE TUNING (FT): suppose we train for $T$ epochs using the learning rate schedule $(\eta_t)_{t \leq T}$ and retrain for $T_{rt}$ epochs per prune-retrain-cycle, then FT retrains the pruned network using a constant learning rate of $\eta_T$, i.e., the last learning rate used during the original training. Renda et al. (2020) note that the learning rate schedule during retraining can have a dramatic impact on the predictive performance of the pruned network and propose LEARNING RATE REWINDING (LRW), where one retrains the pruned network for $T_{rt}$ epochs using the last $T - T_{rt}$ learning rates $\eta_{T-T_{rt}+1}, \ldots, \eta_T$ during each cycle. Le & Hua (2021) further improved upon these results by proposing SCALED LEARNING RATE RESTARTING (SLR), where the pruned network is retrained using a proportionally identical schedule, i.e., by compressing $(\eta_t)_{t \leq T}$ into the retraining time

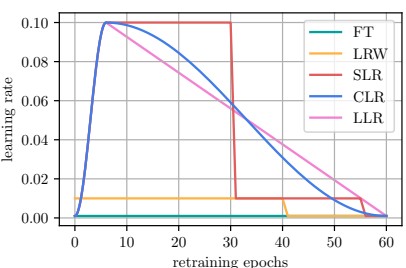

Figure 1: The different learning rate schedules for IMP when retraining for 60 epochs, assuming a stepped learning rate schedule during an initial training lasting for 200 epochs.

frame of $T_{rt}$ epochs with a short warm-up phase. They also introduced CYCLIC LEARNING RATE RESTARTING (CLR) based on the 1-cycle learning rate schedule of Smith & Topin (2017), where the original schedule (commonly a stepped one) is replaced with a cosine based one starting at the same initial learning rate $\eta_1$, likewise including a short warm-up phase. Figure 1 depicts the aforementioned schedules for a retraining budget of 60 epochs.

LRW was proposed as a variant of WEIGHT REWINDING (WR) (Frankle et al., 2019), suggesting that its success is due to some connection to the Lottery Ticket Hypothesis. Le & Hua (2021) already gave a more grounded motivation when introducing SLR by noting that its main feature is the "usage of large learning rates".[1] By proposing CLR and motivating it through the 1-cycle learning rate schedule of Smith & Topin (2017), they also already demonstrated that there is no particular significance to basing the learning rate schedule on the one used during the original training.

We think that the results of Li et al. (2020) regarding the training of Neural Networks within a fixed iteration budget (Budgeted Training) provide some relevant further context for the varying success achieved by these methods as well as indications on how one can further improve upon them, in particular when retraining is assumed to be significantly shorter than the original training, that is when $T_{rt} \ll T$. Li et al. (2020) study training when a resource budget is given in the form of a fixed number of epochs or iterations that the network will be trained for, instead of following the common assumption that training is executed until asymptotic convergence is achieved to some

---

[1]Note that a large initial and then (often exponentially) decaying learning rate has become the standard practice for regular training (Leclerc & Madry, 2020). The conventional approach to explaining the success of such schedules from an optimization perspective is that an initially large learning rate accelerates training and avoids local minima, while the gradual decay helps to converge to an optimum without oscillating around it. However, there are also indications that the usage of large learning rates and the separation of training into a large- and small-step regime help from a generalization perspective (Jastrzębski et al., 2017; Li et al., 2019; You et al., 2019; Leclerc & Madry, 2020).

satisfactory degree. Specifically, they empirically determine what learning rate schedules are best suited to achieve the highest possible performance within a given budget. The two major takeaways from their results are as follow:

1. *Compressing any given learning rate schedule to fit within a specific budget significantly outperforms simply truncating it once the budget is reached.* Li et al. (2020) refer to this compression as BUDGET-AWARE CONVERSION (BAC). This clearly aligns with the findings of Le & Hua (2021) that SLR outperforms LRW, since SLR is simply the BAC of the original training schedule while LRW is a truncated version of it (albeit truncated 'from the back' instead of the front).

2. *Certain learning rate schedules are more suited for a wider variety of budgets than others.* In particular, their results indicate that a linear schedule performs best when a tight budget is given, closely followed by a cosine based approach. This provides an explanation for why CLR outperforms SLR when the original learning rate schedule follows more traditional recommendations.

Put succinctly, the empirical results of Li et al. (2020) regarding the learning rate schedule in a budgeted training context seem to closely resemble the development and improvement of retraining schedules in the context of pruning. Hence, we claim that retraining should first and foremost be considered under the aspect of Budgeted Training and that lessons derived in the latter setting are generally applicable in this context. Motivated by the findings of Li et al. (2020), we therefore propose LINEAR LEARNING RATE RESTARTING (LLR) leveraging a linear learning rate schedule during retraining: LLR linearly decays the learning rate during each retrain cycle from an initial value of $\eta_1$ to zero after a short warm-up phase. This effectively results in a cyclic learning rate schedule when pruning and retraining in the iterative setting, which has previously been found to help generalization (Smith, 2017). Going one step further, we also propose dynamically adapting the initial value of the retraining schedule by relating it not just to the initial learning rate during the original training but also to the impact of the previous pruning step, resulting in ADAPTIVE LINEAR LEARNING RATE RESTARTING (ALLR). While previous works have focused on the actual schedule of the learning rate during retraining, the initial value has only implicitly been dealt with. FT chooses the last learning rate value $\eta_T$, which is typically the smallest. On the other hand, SLR and CLR rely on the initial value corresponding to the maximum value $\eta_1$ of the original schedule, which the authors attribute the success of their methods to. The initial value of LRW is implicitly chosen in proportion to the retraining time by truncating the original schedule from the back.

Existing works have shown that to find minima that generalize well, the learning rate should exhibit a large-step and a small-step retraining regime (Jastrzębski et al., 2017; Li et al., 2019; You et al., 2019; Leclerc & Madry, 2020). When choosing the initial value of the retraining schedule, the two characteristics of a prune-retrain cycle have to be taken into account: its length and the impact of pruning. Given a tight retraining budget it might occur that large initial steps cannot be compensated adequately, while a too small learning rate (possibly over a long retraining) period might be insufficient to recover large pruning-induced performance degradation. An adaptive way of choosing the initial stepsize must therefore address the following question: how much of an *increase in loss* do we have to compensate for and do we have sufficient *time* to properly perform both a large-step and small-step learning rate regime? To that end, ALLR discounts the initial value $\eta_1$ by a factor $d \in [0, 1]$ to account for both the available retraining time (similar to LRW, where the magnitude of the initial learning rate naturally depends on $T_{rt}$) and the performance drop induced by pruning. Since measuring the decrease in train accuracy would require an additional evaluation epoch and is thus undesirable (cf. Appendix C.1 for an ablation study), ALLR achieves this goal by first measuring the relative $L_2$-norm change in the weights due to pruning, that is after pruning a $s \in (0, 1]$ fraction of the remaining weights, we compute the normalized distance between the weight vector $\mathcal{W}$ and its pruned version $\mathcal{W}^p$ in the form of

$$d_1 = \frac{\|\mathcal{W} - \mathcal{W}^p\|_2}{\|\mathcal{W}\|_2 \cdot \sqrt{s}} \in [0, 1], \tag{1}$$

where normalization by $\sqrt{s}$ ensures that $d_1$ can actually attain the full range of values in $[0, 1]$. We then determine $d_2 = T_{rt}/T$ to account for the length of the retrain phase and choose $d\,\eta_1$ as the initial learning rate for ALLR where $d = \max(d_1, d_2)$. This approach effectively interpolates between the recommendations of Renda et al. (2020) and Le & Hua (2021) based on a computationally cheap proxy. Appendix C.1 contains several ablation studies to justify our design choices.

In Section 3.1, we will verify our claims by empirically comparing the retraining schedules, namely FT, LRW, SLR, CLR, LLR and ALLR, against one another as well as against tuned versions of their underlying (constant, stepped, cosine or linear) schedules. We then study in Section 3.2 to what degree the retraining phase of IMP can be shortened when leveraging the proposed schedules.

## 2.2 PRUNING-STABILITY: TRYING TO AVOID RETRAINING

Pruning-stable algorithms are defined by their attempt to find a well-performing pruned model from an initially dense one *during the training procedure* so that the ultimate 'hard' pruning step results in almost no drop in accuracy and the retraining phase becomes superfluous. They do so by inducing a strong implicit bias during some otherwise standard training setup, either by gradual pruning, i.e., extending the pruning mask dynamically, or by employing regularization- and constraint-optimization techniques to learn an almost sparse structure throughout training. Many methods also rely on some kind of 'soft' pruning for this, e.g., by zeroing out weights or strongly pushing them towards zero, but not fully removing them from the network architecture during training.

Let us briefly summarize a variety of methods that have been proposed in this category over the last couple of years: LC (Carreira-Perpinán & Idelbayev, 2018) and GSM (Ding et al., 2019) both employ a modification of weight decay and force the $k$ weights with the smallest score more rapidly towards zero, where $k$ is the number of parameters that will eventually be pruned and the score is the parameter magnitude or its product with the loss gradient. Similarly, DNW (Wortsman et al., 2019) zeroes out the smallest $k$ weights in the forward pass while still using a dense gradient. CS (Savarese et al., 2020), STR (Kusupati et al., 2020) and DST (Liu et al., 2020) all rely on the creation of additional trainable threshold parameters, which are applied to sparsify the model while being regularly trained alongside the usual weights. Here, the training objectives are modified via penalty terms to control the sparsification. GMP (Zhu & Gupta, 2017; Gale et al., 2019) follows a tunable pruning schedule which sparsifies the network throughout training by dynamically extending and updating a pruning mask. Finally, based on this idea, DPF (Lin et al., 2020) maintains a pruning mask which is extended using the pruning schedule of Zhu & Gupta (2017), but allows for error compensation by modifying the update rule to use the (stochastic) gradient of the pruned model while updating the dense parameters.

The two most commonly claimed advantages of pruning-stable methods compared to IMP are the following:

1. *They result in a pruned model faster when training from scratch since they avoid the expensive iterative prune-retrain cycles.* Ding et al. (2019) for example advertise that is there is "no need for time consuming retraining", Liu et al. (2020) try to "avoid the expensive pruning and fine-tuning iterations", Hoefler et al. (2021) state that sparsifying during training "is usually cheaper than the train-then-sparsify schedule", Lin et al. (2020) argue that IMP is "computationally expensive" and "outperformed by algorithms that explore different sparsity masks instead of a single one", Wortsman et al. (2019) try to "train a sparse Neural Network without retraining or fine-tuning" and Frankle & Carbin (2018) (in the context of the Lottery Ticket Hypothesis) state that "iterative pruning is computationally intensive, requiring training a network 15 or more times consecutively".

2. *They produce preferable results either because they avoid 'hard' pruning or due to the particular implicit bias they employ.* Liu et al. (2020) for example state that 'hard' pruning methods suffer from a "failure to properly recover the pruned weights" and Carreira-Perpinán & Idelbayev (2018) argue that learning the pruning set throughout training "helps find a better subset and hence prune more weights with no or little loss degradation".

While many of the previously listed methods perform well and achieve state-of-the-art results, so far little empirical evidence has been given to support that the claimed advantages of pruning-stability have actually been achieved. To verify this, we propose BUDGETED IMP (BIMP), where the same lessons we previously derived from Budgeted Training for the retraining phase of IMP are applied to the initial training of the network. More specifically, given a budget of $T$ epochs, we simply train a network for some $T_0 < T$ epochs using a linearly decaying learning rate schedule and then apply IMP with the proposed schedules on the output for the remaining $T - T_0$ epochs.

Table 1: ResNet-50 on ImageNet: Performance of the different learning rate translation schemes for One Shot IMP for target sparsities of 70%, 80% and 90% and retrain times of 2.22% (2 epochs), 5.55% (5 epochs) and 11.11% (10 epochs) of the initial training budget. The **first**, **second**, and **third** best values are highlighted. Results are averaged over two seeds with the standard deviation indicated.

| | Model sparsity 70% | | | Model sparsity 80% | | | Model sparsity 90% | | |
|---|---|---|---|---|---|---|---|---|---|
| Budget: | 2.22% | 5.55% | 11.11% | 2.22% | 5.55% | 11.11% | 2.22% | 5.55% | 11.11% |
| **FT** | 73.51 ±0.04 | 73.98 ±0.04 | 74.44 ±0.11 | 70.45 ±0.20 | 71.81 ±0.11 | 72.68 ±0.07 | 56.75 ±0.01 | 61.60 ±0.30 | 64.61 ±0.21 |
| **LRW** | 73.50 ±0.04 | 73.99 ±0.04 | 74.45 ±0.11 | 70.45 ±0.20 | 71.82 ±0.12 | 72.67 ±0.07 | 56.75 ±0.01 | 61.61 ±0.30 | 64.60 ±0.23 |
| **SLR** | 70.93 ±0.01 | 72.58 ±0.03 | 73.69 ±0.11 | 70.48 ±0.04 | 72.37 ±0.02 | 73.44 ±0.18 | 67.19 ±0.23 | 69.45 ±0.01 | 70.80 ±0.09 |
| **CLR** | 72.22 ±0.09 | 73.58 ±0.08 | 74.49 ±0.04 | 71.96 ±0.09 | 73.30 ±0.08 | 74.24 ±0.08 | 68.72 ±0.06 | 70.60 ±0.15 | 71.51 ±0.13 |
| **LLR (ours)** | 72.39 ±0.13 | 73.65 ±0.05 | 74.34 ±0.02 | 72.07 ±0.09 | 73.41 ±0.05 | 74.23 ±0.10 | 68.90 ±0.05 | 70.48 ±0.01 | 71.53 ±0.09 |
| **ALLR (ours)** | 73.69 ±0.03 | 74.37 ±0.05 | 74.89 ±0.04 | 72.96 ±0.15 | 74.02 ±0.08 | 74.71 ±0.04 | 69.56 ±0.07 | 71.19 ±0.01 | 71.99 ±0.07 |

The resulting method is capable of obtaining a pruned model from a random initialization within any given budget $T$ while still maintaining all of the key characteristics of IMP, most notably the fact that (1) we 'hard' prune and do not allow weights to recover in subsequent steps, (2) we do not impose any particular additional implicit bias besides a common weight decay term during either training or retraining, and (3) we follow a prescribed static training schedule with the exception of adapting the initial learning rate to the impact of the last pruning in the case of ALLR. This clearly delineates BIMP from the previously listed methods and allows us to compare them on equal terms by giving all methods the same budget of a total of $T$ epochs, independent of whether they are spent on 'normal' training or retraining. In Section 3.3 we thoroughly compare our proposed approach to previously listed pruning-stable methods in a fair setting. We remark that the implicit biases of many pruning-stable approaches can result in a substantial computational overhead that we are deliberately ignoring here by comparing methods on a per-epoch basis and therefore giving these methods the advantage in the comparison. However, we include the images-per-second throughput of the individual algorithms which highlights that BIMP is among the most efficient approaches.

Finally, let us remark that there has been a significant amount of attention on how to select the specific weights to be pruned. Ranking weights for pruning based on the magnitude of their current values has established itself as the approach of choice (Lee et al., 2019), where specific criteria have been proposed that take the particular network architecture into consideration (Zhu & Gupta, 2017; Gale et al., 2019; Evci et al., 2020; Lee et al., 2020). We have verified some of these results in Appendix C.2 and will stick to the simple global selection criterion used by Han et al. (2015) for BIMP.

# 3 EXPERIMENTAL RESULTS

Let us outline the general methodological approach to computational experiments in this section, including datasets, architectures and metrics. Experiment-specific details are found in the respective subsections. We note that, given the surge of interest in pruning, Blalock et al. (2020) proposed experimental guidelines in the hope of standardizing the experimental setup. We aim to follow these guidelines whenever possible. All experiments performed throughout this computational study are based on the PyTorch framework (Paszke et al., 2019), using the original code of the methods whenever available. All results and metrics were logged and analyzed using Weights & Biases (Biewald, 2020). We have made our code and general setup available at github.com/ZIB-IOL/BIMP for the sake of reproducibility.

We perform extensive experiments on image recognition datasets such as *ImageNet* (Russakovsky et al., 2015), *CIFAR-10/100* (Krizhevsky et al., 2009), the semantic segmentation tasks *COCO* (Lin et al., 2014) and *CityScapes* (Cordts et al., 2016) as well as neural machine translation (NMT) on *WMT16* (Bojar et al., 2016). In particular, we employed *ResNets* (He et al., 2015), *Wide ResNets* (WRN) (Zagoruyko & Komodakis, 2016), *VGG* (Simonyan & Zisserman, 2014), the transformer-based *MaxViT* (Tu et al., 2022) architecture, as well as *PSPNet* (Zhao et al., 2017) and *DeepLabV3* (Chen et al., 2017) in the case of CityScapes and COCO, respectively. For NMT, we used a *T5* transformer (Raffel et al., 2020) available through *HuggingFace* (Wolf et al., 2020). Exact parameters can be found in Appendix A, where we also define what can be considered a 'standard' training setup for each setting that we rely on whenever not otherwise specified. The focus of our analysis will

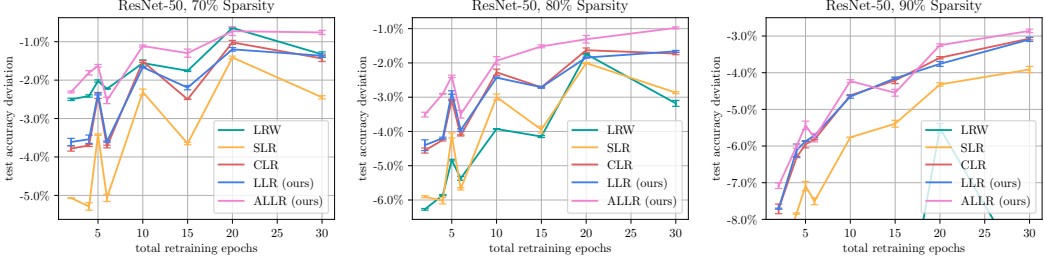

Figure 2: ResNet-50 on ImageNet: Performance of different retraining schedules compared to the dense model shown over the total number of epochs used for retraining including both One Shot and iterative magnitude pruning. Results are averaged over two seeds with max-min-bands indicated and the plots depict sparsity 70%, 80% and 90% from left to right.

be the tradeoff between the model sparsity and the final test performance, being the accuracy in the case of image classification, the mIoU (mean intersection over union) for segmentation, or the BLEU score (Post, 2018) for NMT. As a secondary measure, we will also consider the *theoretical speedup* (Blalock et al., 2020) induced by the sparsity, see Appendix A for full details. We use a validation set of 10% of the training data for hyperparameter selection.

## 3.1 LEARNING RATE SCHEDULES DURING RETRAINING

Table 1 contains part of the results regarding the comparison between FT, LRW, SLR, CLR and our proposed approaches LLR and ALLR for ImageNet in the One Shot setting. First of all, we find that retraining after pruning is in fact a Budgeted Training scenario, as the insights for normal dense training (Li et al., 2020) transfer to the retraining case. This is further observable when comparing the translated schedules to versions of constant, stepped exponential, cosine and linear learning rate schedules, where the initial learning rate after pruning was tuned using a grid search (cf. Appendix B.1, also containing additional results, exact parameter grids and the results for other datasets). In general, linear and cosine based schedules clearly outperform the constant and stepped ones, with a slight advantage of LLR over CLR.

However, for short retraining times and for the medium sparsity range, the fixed restarting schedules CLR and LLR fail to yield results competitive to FT and LRW, since a too large initial learning rate is detrimental given a restricted retraining budget. ALLR is able to consistently improve upon previous approaches, which becomes especially noticeable in the small retraining budget regime, where for larger budgets the approaches begin to converge. We think that ALLR is a suitable drop-in replacement when performing retraining. We similarly observe the strength of ALLR in Figure 2, depicting the highest achievable test accuracy for each number of total retraining epochs, including both the One Shot as well as iterative pruning case. Appendix B.1 includes the full results on different tasks and datsets, longer retraining budgets as well as the structured pruning setting, where we remove convolutional filters based on their respective norm (Li et al., 2016).

## 3.2 BUDGETING THE RETRAINING PHASE

In this part we will treat the number of retrain epochs per prune-retrain cycle $T_{rt}$ as well as the total amount of such cycles $J$ as tunable hyperparameters for IMP and try to determine the tradeoff between the predictive performance of the final pruned network and the total number of retrained epochs $J \cdot T_{rt}$. As a baseline performance for a pruned network, we will use the approach suggested by Renda et al. (2020) as it serves as a good benchmark for the current potential of IMP.

In Figure 3 we present the envelope of the results for ResNet-56 trained on CIFAR-10 with target sparsities of 90%, 95%, and 98%, respectively. The parameters for the retraining phase were optimized using a grid search over $T_{rt} \in \{10, 15, 20, ..., 60\}$ and $J \in \{1, 2, ..., 6\}$ using ALLR. We find that IMP is capable of achieving what has previously been considered its full potential with significantly less than the total number of retraining epochs usually budgeted for its iterative form.

Table 2: ResNet-56 on CIFAR-10 (above) and ResNet-50 on ImageNet (below): Comparison between BIMP and pruning-stable methods when training for goal sparsity levels of 90%, 95%, 99% (CIFAR-10) and 70%, 80%, 90% (ImageNet), denoted in the main columns. Each subcolumn denotes the Top-1 accuracy, the theoretical speedup and the actual sparsity achieved by the method. Further, we denote the images-per-second throughput during training, i.e., a higher number indicates a faster method. All results are averaged over multiple seeds and include standard deviations. The **first**, **second**, and **third** best values are highlighted.

**CIFAR-10**

| Method | # img/s | Model sparsity 90% | | | Model sparsity 95% | | | Model sparsity 99% | | |
|---|---|---|---|---|---|---|---|---|---|---|
| | | Accuracy | Speedup | Sparsity | Accuracy | Speedup | Sparsity | Accuracy | Speedup | Sparsity |
| **BIMP (ours)** | 3638 | 93.35 ±0.13 | 7 ±0.2 | 90.00 ±0.00 | 92.57 ±0.32 | 12 ±0.8 | 95.00 ±0.00 | 87.17 ±0.59 | 56 ±4.9 | 99.00 ±0.00 |
| **GMP** | 3536 | 92.84 ±0.42 | 10 ±0.0 | 90.00 ±0.00 | 92.12 ±0.17 | 20 ±0.0 | 95.00 ±0.00 | 86.72 ±0.04 | 77 ±0.0 | 99.00 ±0.00 |
| **GSM** | 3251 | 91.27 ±0.69 | 11 ±2.5 | 90.00 ±0.00 | 90.07 ±1.67 | 21 ±5.0 | 95.00 ±0.00 | 83.00 ±0.90 | 77 ±18.9 | 99.00 ±0.00 |
| **DPF** | 3560 | 93.32 ±0.11 | 7 ±0.0 | 90.00 ±0.00 | 92.68 ±0.14 | 12 ±0.1 | 95.00 ±0.00 | 86.76 ±0.33 | 63 ±3.7 | 99.00 ±0.00 |
| **DNW** | 3335 | 91.81 ±1.83 | 6 ±0.8 | 90.00 ±0.00 | 91.95 ±0.06 | 7 ±0.3 | 95.09 ±0.00 | 83.67 ±0.24 | 15 ±0.1 | 99.17 ±0.00 |
| **LC** | 3467 | 90.51 ±0.16 | 5 ±0.1 | 90.00 ±0.00 | 89.16 ±0.60 | 8 ±0.5 | 95.00 ±0.00 | 81.63 ±0.74 | 30 ±1.5 | 99.00 ±0.00 |
| **STR** | 2864 | 89.25 ±1.23 | 8 ±0.8 | 90.15 ±0.76 | 89.77 ±1.75 | 31 ±10.3 | 95.11 ±0.28 | 83.68 ±0.94 | 159 ±31.9 | 99.13 ±0.02 |
| **CS** | 2725 | 91.87 ±0.30 | 13 ±0.3 | 90.52 ±0.76 | 91.36 ±0.23 | 21 ±2.9 | 95.38 ±0.19 | 86.55 ±0.92 | 69 ±7.1 | 98.90 ±0.02 |
| **DST** | 1972 | 92.41 ±0.28 | 10 ±0.7 | 89.55 ±0.41 | 89.17 ±0.00 | 18 ±0.0 | 94.42 ±0.00 | 86.99 ±0.00 | 63 ±0.0 | 98.36 ±0.00 |

**ImageNet**

| Method | # img/s | Model sparsity 70% | | | Model sparsity 80% | | | Model sparsity 90% | | |
|---|---|---|---|---|---|---|---|---|---|---|
| | | Accuracy | Speedup | Sparsity | Accuracy | Speedup | Sparsity | Accuracy | Speedup | Sparsity |
| **BIMP (ours)** | 1454 | 75.62 ±0.02 | 2 ±0.0 | 70.00 ±0.00 | 75.08 ±0.16 | 3 ±0.0 | 80.00 ±0.00 | 73.53 ±0.05 | 6 ±0.0 | 90.00 ±0.00 |
| **GMP** | 1425 | 74.62 ±0.08 | 2 ±0.0 | 70.00 ±0.00 | 74.19 ±0.17 | 4 ±0.0 | 80.00 ±0.00 | 72.80 ±0.03 | 7 ±0.1 | 90.00 ±0.00 |
| **GSM** | 1349 | 73.69 ±0.70 | 2 ±0.1 | 70.00 ±0.00 | 72.75 ±0.62 | 4 ±0.3 | 80.00 ±0.00 | 70.08 ±0.94 | 9 ±0.8 | 90.00 ±0.00 |
| **DPF** | 1456 | 75.59 ±0.07 | 2 ±0.0 | 70.00 ±0.00 | 75.30 ±0.02 | 3 ±0.0 | 80.00 ±0.00 | 74.05 ±0.05 | 6 ±0.0 | 90.00 ±0.00 |
| **DNW** | 530 | 75.60 ±0.01 | 2 ±0.0 | 70.00 ±0.00 | 75.27 ±0.01 | 3 ±0.0 | 80.00 ±0.00 | 74.29 ±0.03 | 5 ±0.1 | 90.00 ±0.00 |
| **LC** | 1436 | 75.03 ±0.20 | 2 ±0.0 | 70.00 ±0.00 | 73.87 ±0.62 | 3 ±0.0 | 80.00 ±0.00 | 67.57 ±2.71 | 5 ±0.0 | 90.00 ±0.00 |
| **STR** | 1396 | 70.66 ±0.13 | 3 ±0.0 | 75.34 ±0.01 | 70.70 ±0.13 | 4 ±0.0 | 80.93 ±0.00 | 70.13 ±0.01 | 8 ±0.0 | 90.00 ±0.00 |
| **DST** | 1219 | 74.63 ±0.22 | 4 ±0.1 | 70.00 ±0.00 | 73.16 ±0.11 | 6 ±0.1 | 80.00 ±0.00 | 71.35 ±0.09 | 13 ±0.4 | 90.00 ±0.00 |

More concretely, for all three levels of sparsity, 90%, 95%, and 98%, IMP meets the baseline laid out by Renda et al. (2020) after only around 100 epochs of retraining, instead of requiring the 2000, 2800, and 3600 epochs used to establish that baseline, respectively. In fact, given the superiority of a linear learning rate schedule over a more commonly used stepped one in the case of budgeted training (Li et al., 2020), ALLR clearly continues and exceeds the stepped learning rate schedule baseline. Additional results and exact parameter grids are included in Appendix B.2.

Given that we have just established that the retraining phase of IMP takes well to enforcing a budget when using an appropriate learning rate schedule and that Li et al. (2020) already established that 'normal' training can be significantly shortened through a linear learning rate schedule, it is reasonable to assume that the same holds for the original training phase without strongly impacting the pruning and retraining part and therefore the ultimate product of IMP.

To verify this, we trained ResNet-56 on CIFAR-10 using a linearly decaying learning rate schedule from between 5% up to 100% of 200 epochs, which we consider the 'full' training, and then apply IMP with ALLR on the resulting network both One Shot and iteratively for target sparsities of 90%, 95%, and 98%. Figure 7 in the appendix shows the results for the iterative setting, where we retrain for three cycles of 15 epochs each.

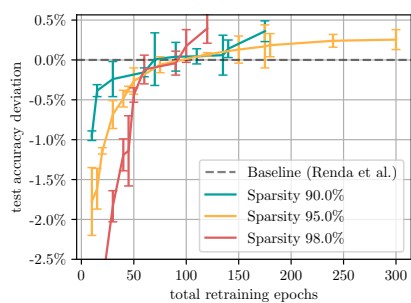

Figure 3: ResNet-56 on CIFAR-10: Envelope of the performance of IMP using ALLR compared to the baseline of Renda et al. (2020) shown over the total number of epochs used for retraining.

We can see that IMP takes well to budgeting the initial training period, both in the One Shot and in the iterative setting, with the target sparsity seemingly having little influence on how much the initial training can be compressed.

### 3.3 THE EFFICACY OF PRUNING-STABILITY

We conclude by comparing the performance of BIMP to pruning-stable approaches. To that end, we train models on ImageNet, CIFAR-100 and CIFAR-10 and give all methods an equal budget of 90 epochs (200 for CIFAR) to derive a pruned model from a randomly initialized one. For BIMP, we employ ALLR and treat the initial training length $T_0$ as well as the number of prune-retrain-cycles as hyperparameters, ensuring that the overall epoch budget $T$ is not exceeded. The hyperparameters of each of the pruning-stable methods were likewise tuned using manually defined grid searches, resorting to the recommendations of the original publications whenever possible, see Appendix B.1. For GMP, GSM, DPF, DNW, and LC we give the methods prespecified sparsity levels, the same as given to BIMP. Tuning the remaining methods in a way that allows for a fair comparison however is significantly more difficult, since none of them allow to clearly specify a desired level of sparsity but instead require tuning additional hyperparameters as part of the grid search. Despite our best efforts, we were only able to cover part of the desired sparsity range using STR and DST. For CS in the case of ImageNet, we were unable to tune the hyperparameters. In addition, we noticed that each of these methods can have some variance in the level of sparsity achieved even for fixed hyperparameters, so we list the standard deviation of the final sparsity with respect to the random seed initialization. We note that in the original works, LC and GSM were applied to pretrained models. To allow a fair comparison, we applied LC and GSM to both randomly initialized as well as pretrained models and chose the best results for each sparsity, giving them a larger budget than the others. Further, we noticed that some pruning-stable methods can profit from retraining. For CIFAR-10 and CIFAR-100, we hence retrain all methods excluding BIMP for 30 epochs using FT and use the accuracy reached after retraining when it exceeds the original one. All results are averaged over multiple seeds with standard deviation indicated.

Table 2 reports the final test performance, theoretical speedup, and actually achieved sparsity of all methods for CIFAR-10 and ImageNet, where we defer full results to Appendix B.4. The results show that BIMP is able to outperform many of the pruning-stable methods considered here. For ImageNet, DNW consistently performs on par or better than BIMP, albeit at the price of needing roughly twice as long for training, cf. the images-per-second throughput. Surprisingly, despite broad hyperparameter grid search, most methods seem to be in disadvantage compared to BIMP, with DPF being a both efficient and strong competitor. BIMP obtains these results within the same number of overall training epochs and we are ignoring the computational overhead of some of the more involved methods. We note that the authors of STR report better results on ImageNet, which we unfortunately were unable to replicate in our experimental setting (cf. Appendix B.1 for the exact hyperparameter grid).

In Appendix C.3, we have included an ablation study where we compare BIMP to several modifications of GMP not previously suggested in the literature, since GMP is the closest in design to BIMP out of all pruning-stable methods considered here. Most notably this includes variants of GMP with both a global and a cyclical linear learning rate schedule as well as a 'hard' pruning variant.

## 4 DISCUSSION AND OUTLOOK

The learning rate is often considered to be the single most important hyperparameter in Deep Learning which nevertheless still remains poorly understood, certainly from a theoretical perspective but also still from an empirical one. Our work therefore provides an important building block in which we established that, counter to the often explicitly stated belief that IMP is inefficient, many significantly more complex and sometimes strenuously motivated methods are outperformed by perhaps the most basic of approaches when proper care is taken of the learning rate.

Despite providing a strong retraining alternative with ALLR, we emphasize that the main goal of this work is not to suggest yet another acronym and claim that it is the be-all and end-all of network pruning, but to instead (a) hopefully focus the efforts of the community on understanding more basic questions before suggesting convoluted novel methods and (b) to emphasize that IMP can serve as a strong, easily implemented, and modular baseline. We think the modularity here is of particular importance, as individual aspects can easily be exchanged or modified, e.g., when, what, how, and how much to prune or how to retrain, in order to formulate rigorous ablation studies.

## REPRODUCIBILITY

Reproducibility is of utmost importance for any comparative computational study such as this. All experiments were based on the PyTorch framework and use publicly available datasets. The implementation of the ResNet-56 and ResNet-18 network architecture is based on github.com/JJGO/shrinkbench and github.com/charlieokonomiyaki/pytorch-resnet18-cifar10, respectively, the implementation of the WideResNet network architecture is based on github.com/meliketoy/wide-resnet.pytorch, the implementation of the VGG-16 network architecture is based on github.com/jaeho-lee/layer-adaptive-sparsity and the implementation of the Resnet-50 network architecture is taken from PyTorch. Regarding the pruning methods, the code was taken from the respective publications whenever possible. Regarding the different variants of magnitude pruning such as ERK or UNIFORM+, we closely followed the implementation of Lee et al. (2020) available at github.com/jaeho-lee/layer-adaptive-sparsity. For metrics such as the theoretical speedup, we relied on the implementation in the *ShrinkBench*-framework of Blalock et al. (2020), see github.com/JJGO/shrinkbench. We have made our code and general setup available at github.com/ZIB-IOL/BIMP for the sake of reproducibility.

## ACKNOWLEDGEMENTS

This research was partially supported by the DFG Cluster of Excellence MATH+ (EXC-2046/1, project id 390685689) funded by the Deutsche Forschungsgemeinschaft (DFG). We would like to thank Berkant Turan for his support in conducting the NMT experiment using the HuggingFace library.

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

# A  TECHNICAL DETAILS AND TRAINING SETTINGS

## A.1  TECHNICAL DETAILS AND GENERAL TRAINING SETTINGS

We define *pruning-stability*, *theoretical speedup* as well as several *pruning selection criteria* for IMP, i.e., different criteria that are used to select weights for pruning. We will analyze the impact of such criteria under different retraining schedules in Appendix C.2. Further, Table 3 shows the default training settings used throughout this work.

**Definition A.1** (Bartoldson et al. (2020)). Let $t_{\text{pre}}$ and $t_{\text{post}}$ be the test accuracy before pruning and after pruning the trained model, respectively. Assuming $t_{\text{post}} \leq t_{\text{pre}}$, we define the *pruning-stability* of a method as

$$\Delta_{\text{stability}} := 1 - \frac{t_{\text{pre}} - t_{\text{post}}}{t_{\text{pre}}} \in [0, 1].$$

Pruning-stable methods are sparsification algorithms that learn a sparse solution throughout training such that $\Delta_{\text{stability}} \approx 1$. For example, methods that perform the forward-pass using an already sparsified copy of the parameters (e.g. DNW by Wortsman et al., 2019), will have $\Delta_{\text{stability}} = 1$, since the 'hard' pruning step only consists of an application of the present pruning mask, which has no further effect. Methods that actively drive certain parameter groups towards zero more rapidly (such as Carreira-Perpinán & Idelbayev, 2018; Ding et al., 2019) will have a pruning-stability close to 1, since the projection of (magnitude) pruning at the end of training will perturb the parameters only slightly.

Crucial to our analysis are the tradeoffs between the model sparsity, the final performance (measured by *final test accuracy*, *BLEU scores* or *IoU*) and the *theoretical speedup* induced by the sparsity (Blalock et al., 2020). The theoretical speedup is a metric measuring the ratio in FLOPs needed for inference comparing the dense and sparse model. More precisely, let $F_d$ be the number of FLOPs the dense model needs for inference, and let similarly be $F_s$ the same number for the pruned model, given some sparsity $s$.[2] The theoretical speedup is defined as $F_d/F_s$ and depends solely on the position of the zero weights within the network and layers, not on the numerical values of non-zero parameters.

IMP in its original form treats all trainable parameters as a single vector and computes a global threshold below which parameters are removed, independent of the layer they belong to. This simple approach, which we will refer to as GLOBAL, has been subject to criticism for not determining optimal layer-dependent pruning rates and for being inconsistent (Liu et al., 2020). Fully-connected layers for example have many more parameters than convolutional layers and are therefore much less sensitive to weight removal (Han et al., 2015; Carreira-Perpinán & Idelbayev, 2018). Further, it has been observed that the position of a layer can play a role in whether that layer is amenable to pruning: often first and last layers are claimed to be especially relevant for the classification performance (Gale et al., 2019). On the other hand, in which layers pruning takes place significantly impacts the sparsity-induced theoretical speedup (Blalock et al., 2020). Lastly, the non-negative homogeneity of modern ReLU-based Neural Network architectures (Neyshabur et al., 2015) would also seem to indicate a certain amount of arbitrariness to this heuristic selection rule, or at least a strong dependence on the network initialization rule and optimizer used, as weights can be rescaled to force it to fully remove all parameters of a layer, destroying the pruned network without having affected the output of the unpruned network.

Determining which weights to remove is hence crucial for successful pruning and several methods have been designed to address this fact. Zhu & Gupta (2017) introduced the UNIFORM allocation, in which a global sparsity level is enforced by pruning each layer to exactly this sparsity. Gale et al. (2019) extend this approach in the form of UNIFORM+ by (a) keeping the first convolutional layer dense and (b) pruning at most 80% of the connections in the last fully-connected layer. Evci et al. (2020) propose a reformulation of the ERDŐS-RÉNYI KERNEL (ERK) (Mocanu et al., 2018) to take the layer and kernel dimensions into account when determining the layerwise sparsity distribution. In particular, ERK allocates higher sparsity to layers with more parameters. Finally, Lee et al. (2020) propose LAYER-ADAPTIVE MAGNITUDE-BASED PRUNING (LAMP), an approach which takes an $\ell_2$-distortion perspective by relaxing the problem of minimizing the output distortion at time of pruning with respect to the worst-case input. We note that we follow the advice of Evci et al. (2020)

---

[2]To compute the number of FLOPs, we sample a single batch from the test set. The code to compute the theoretical speedup has been adapted from the repository of the *ShrinkBench* framework (Blalock et al., 2020).

Table 3: Exact training configurations used throughout the experiments for IMP. We note that others have reported an accuracy of around 80% for WRN28x10 trained on CIFAR-100 that we were unable to replicate. The discrepancy is most likely due to an inconsistency in PyTorch's dropout implementation. For experiments involving Vision-Transformers, we used label smoothing as well as gradient clipping. For COCO and CityScapes architectures, we rely on pretrained backbones and report the common mean Intersection-over-Union (IoU) metric measured on the validation set. For the NMT task we report the BLEU score on the test set, where we limit the sequence length to 128 throughout.

| Dataset | Network (number of weights) | Epochs | Batch size | Momentum | Learning rate ($t$ = training epoch) | Unpruned test accuracy/IoU/BLEU |
|---|---|---|---|---|---|---|
| CIFAR-10 | ResNet-56 (850 K) ResNet-18 (11 Mio) VGG-16 (138 Mio) | 200 | 128 | 0.9 | $\eta_t = \begin{cases} 0.1 & t \in [1,90], \\ 0.01 & t \in [91,180], \\ 0.001 & t \in [181,200] \end{cases}$ | 93.5% ±0.3% 95.0% ±0.04% 93.8% ±0.2% |
| CIFAR-100 | WRN28x10 (37 Mio) | 200 | 128 | 0.9 | $\eta_t = \begin{cases} 0.1 & t \in [1,60], \\ 0.02 & t \in [61,120], \\ 0.004 & t \in [121,160], \\ 0.0008 & t \in [161,200] \end{cases}$ | 76.7% ±0.2% |
| ImageNet | ResNet-50 (26 Mio) | 90 | 256 | 0.9 | $\eta_t = \begin{cases} 0.1\frac{t}{5} & t \in [1,5], \\ 0.1 & t \in [5,30], \\ 0.01 & t \in [31,60], \\ 0.001 & t \in [61,80], \\ 0.0001 & t \in [81,90] \end{cases}$ | 76.17% ±0.03% |
| ImageNet | MaxViT (31 Mio) | 200 | 256 | 0.9 | $\eta_t = \begin{cases} 0.2\frac{t}{20} & t \in [1,20], \\ 0.2 & t \in [20,60], \\ 0.02 & t \in [61,120], \\ 0.002 & t \in [121,160], \\ 0.0002 & t \in [161,200] \end{cases}$ | 78.0% ±0.02% |
| COCO | DeepLabV3 (40 Mio) | 30 | 36 | 0.9 | $\eta_t = \begin{cases} 0.05 & t \in [1,20], \\ 0.005 & t \in [21,26], \\ 0.0005 & t \in [27,30] \end{cases}$ | 63.08 IoU ±0.3 |
| CityScapes | PSPNet (68 Mio) | 300 | 12 | 0.9 | $\eta_t = \begin{cases} 0.1\frac{t}{20} & t \in [1,20], \\ 0.1 & t \in [20,100], \\ 0.01 & t \in [101,200], \\ 0.001 & t \in [201,270], \\ 0.0001 & t \in [271,290] \\ 0.00001 & t \in [291,300] \end{cases}$ | 58.3 IoU ±0.5 |
| WMT16 (EN-DE) | T5-small (77 Mio) | 5 | 16 | 0.9 | $\eta_t = \begin{cases} 0.1t & t \in [0,1.0], \\ 0.1 & t \in [1,2[, \\ 0.01 & t \in [2,3[, \\ 0.001 & t \in [3,4[, \\ 0.0001 & t \in [4,5] \end{cases}$ | 24.56 BLEU ±0.007 |

and Dettmers & Zettlemoyer (2019) and do not prune biases and batch-normalization parameters, since they only amount to a negligible fraction of the total weights, however keeping them has a very positive impact on the performance of the learned model. Further, for the computations involving GMP, we similarly employ the global selection criterion since we found it to yield better results than UNIFORM+.

We will compare these approaches in Appendix C.2 with a focus on the impact of the retraining phase. Since Le & Hua (2021) found that SLR can be used to obtain strong results even when pruning convolutional filters randomly, i.e., by assigning random importance scores to the filters instead of using the magnitude criterion or others, we are interested in understanding the importance of the retraining technique when considering different sparsity distributions.

For experiments involving the pruning of convolutional filters instead of weights, we follow Li et al. (2016) and remove filters using an $L_2$-norm criterion, enforcing a uniform distribution of sparsity among the layers.

# B EXTENDED RESULTS AND COMPLETE TABLES

## B.1 LEARNING RATE SCHEDULES DURING RETRAINING

This section contains the complete results regarding the performance of different learning rate schedules for retraining. For fixed schedules, i.e., FT, LRW, SLR, CLR, LLR and ALLR, only the weight decay parameter is tuned and the best configuration reported. For the tuned schedules, i.e., constant, stepped (BAC), cosine and linear, we tune the weight decay, the initial value of the learning rate as well as the length of warm-up as follows. For ResNet-18 and ResNet-56 trained on CIFAR-10, weight decay is tuned using a grid search over 1e-4, 2e-4, and 5e-4 and for the tuned schedules in the lower half of the table the initial value is chosen using a grid search over $\{0.001, 0.005, 0.01, 0.05, 0.1, 0.5\}$, where in the iterative case we use the same value for each cycle, and the warm-up is tuned over either zero or ten percent of the retraining budget. For WideResNet on CIFAR-100, weight decay is tuned over 2e-4, and 5e-4, the initial value is chosen using a grid search over $\{0.0008, 0.004, 0.02, 0.05, 0.1, 0.5\}$, where we vary the warmup length between zero and ten percent of the retraining budget. For ResNet-50 on ImageNet, MaxViT on ImageNet, DeepLabV3 on COCO, PSPNet on CityScapes and T5-small on WMT16, we only present the results for fixed schedules. For ResNet-50 on ImageNet, we keep the weight decay fixed at 1e-4, while for the other four aforementioned architecture-dataset pairs, we set the weight decay to 1e-5.

As indicated in the caption, each table displays the results for one architecture-dataset pair either in the One Shot or the iterative setting. If no pruning method is indicated, we report the results of magnitude pruning with a global selection criterion. Whenever we perform filter pruning, we indicate it the caption of the table and rely on a uniform selection of filters by their $L_2$-norm.

Table 4: ResNet-56 on CIFAR-10 (One Shot): Performance of the different learning rate translation schemes (above) compared to tuned schedules (below) for IMP in the One Shot setting for target sparsity of 50%, 90%, 95%, 99% and a retrain time of 2.5%, 5%, 10%, and 25% of the initial training budget. The **first**, **second**, and **third** best values for the translation schemes are highlighted. Results are averaged over two seeds with the standard deviation indicated.

| | Model sparsity 50% | | | | Model sparsity 90% | | | |
|---|---|---|---|---|---|---|---|---|
| Budget: | 2.5% | 5% | 10% | 25% | 2.5% | 5% | 10% | 25% |
| FT | 93.42 ±0.33 | 93.36 ±0.42 | 93.40 ±0.36 | 93.62 ±0.21 | 90.72 ±0.22 | 91.34 ±0.07 | 91.51 ±0.09 | 92.02 ±0.11 |
| LRW | 93.40 ±0.30 | 93.34 ±0.43 | 93.42 ±0.32 | 93.38 ±0.01 | 90.76 ±0.23 | 91.32 ±0.08 | 91.47 ±0.02 | 92.44 ±0.18 |
| SLR | 92.00 ±0.23 | 92.62 ±0.18 | 92.88 ±0.28 | 93.29 ±0.02 | 91.05 ±0.09 | 91.61 ±0.04 | 92.09 ±0.08 | 92.27 ±0.26 |
| CLR | 92.72 ±0.14 | 93.14 ±0.12 | 93.42 ±0.23 | 93.63 ±0.03 | 91.66 ±0.09 | 92.18 ±0.02 | 92.27 ±0.13 | 92.56 ±0.28 |
| LLR | 92.62 ±0.04 | 93.06 ±0.35 | 93.37 ±0.16 | 93.55 ±0.26 | 91.67 ±0.06 | 92.19 ±0.20 | 92.30 ±0.20 | 92.64 ±0.11 |
| ALLR | 93.18 ±0.45 | 93.46 ±0.21 | 93.55 ±0.18 | 93.82 ±0.01 | 91.72 ±0.00 | 92.00 ±0.04 | 92.44 ±0.12 | 92.45 ±0.31 |
| constant | 93.42 ±0.34 | 93.36 ±0.35 | 93.40 ±0.28 | 93.55 ±0.25 | 90.92 ±0.11 | 91.33 ±0.01 | 91.55 ±0.01 | 91.91 ±0.04 |
| stepped | 93.54 ±0.37 | 93.58 ±0.33 | 93.47 ±0.33 | 93.64 ±0.28 | 91.93 ±0.10 | 92.30 ±0.18 | 92.43 ±0.01 | 92.61 ±0.15 |
| cosine | 93.50 ±0.32 | 93.48 ±0.41 | 93.62 ±0.25 | 93.71 ±0.22 | 92.06 ±0.02 | 92.36 ±0.06 | 92.67 ±0.15 | 92.66 ±0.02 |
| linear | 93.50 ±0.35 | 93.54 ±0.35 | 93.70 ±0.31 | 93.75 ±0.39 | 92.01 ±0.01 | 92.25 ±0.08 | 92.43 ±0.10 | 92.69 ±0.08 |

| | Model sparsity 95% | | | | Model sparsity 99% | | | |
|---|---|---|---|---|---|---|---|---|
| Budget: | 2.5% | 5% | 10% | 25% | 2.5% | 5% | 10% | 25% |
| FT | 87.61 ±0.09 | 88.50 ±0.17 | 89.59 ±0.03 | 90.27 ±0.54 | 64.68 ±2.53 | 68.97 ±2.14 | 73.48 ±1.13 | 77.54 ±0.31 |
| LRW | 87.61 ±0.04 | 88.56 ±0.16 | 89.60 ±0.08 | 91.03 ±0.08 | 64.66 ±2.54 | 68.98 ±2.14 | 73.44 ±1.17 | 81.38 ±0.16 |
| SLR | 88.38 ±0.45 | 89.54 ±0.23 | 90.43 ±0.08 | 90.75 ±0.48 | 77.05 ±0.16 | 78.98 ±0.04 | 80.23 ±0.28 | 81.42 ±0.03 |
| CLR | 89.45 ±0.38 | 90.22 ±0.05 | 90.76 ±0.10 | 91.04 ±0.14 | 77.96 ±0.08 | 79.40 ±0.08 | 80.71 ±0.38 | 81.95 ±0.06 |
| LLR | 89.52 ±0.08 | 90.31 ±0.34 | 90.92 ±0.28 | 91.16 ±0.40 | 78.11 ±0.07 | 79.63 ±0.55 | 81.04 ±0.27 | 81.58 ±0.12 |
| ALLR | 89.62 ±0.25 | 90.42 ±0.14 | 90.97 ±0.34 | 91.25 ±0.46 | 77.96 ±0.06 | 79.49 ±0.48 | 80.96 ±0.13 | 81.94 ±0.25 |
| constant | 88.19 ±0.66 | 88.81 ±0.06 | 89.56 ±0.04 | 90.34 ±0.59 | 73.33 ±1.04 | 74.91 ±0.40 | 77.59 ±0.49 | 78.90 ±0.45 |
| stepped | 89.81 ±0.25 | 90.31 ±0.08 | 90.89 ±0.13 | 91.09 ±0.45 | 77.33 ±0.35 | 79.01 ±0.13 | 80.75 ±0.06 | 81.86 ±0.26 |
| cosine | 89.84 ±0.03 | 90.50 ±0.40 | 91.17 ±0.21 | 91.32 ±0.37 | 77.81 ±0.14 | 79.75 ±0.25 | 80.87 ±0.11 | 82.11 ±0.21 |
| linear | 89.84 ±0.13 | 90.63 ±0.30 | 90.98 ±0.25 | 91.29 ±0.39 | 78.11 ±0.32 | 80.00 ±0.15 | 81.14 ±0.01 | 82.12 ±0.08 |

Table 5: ResNet-56 on CIFAR-10 (Iterative): Performance of the different learning rate translation schemes (above) compared to tuned schedules (below) for IMP in the iterative setting for target sparsity of 90%, 95%, 99% and retrain times as indicated in the Budget row. Here $2 \times 2.5\%$ indicates two prune-retrain cycles, each of which having length equal to 2.5% of the overall training budget. The first, second, and third best values for the translation schemes are highlighted. Results are averaged over two seeds with the standard deviation indicated.

| | **Model sparsity 90%** | | | | | |
|---|---|---|---|---|---|---|
| Budget: | $2 \times 2.5\%$ | $2 \times 5\%$ | $2 \times 10\%$ | $3 \times 2.5\%$ | $3 \times 5\%$ | $3 \times 10\%$ |
| **FT** | 90.96 ±0.09 | 91.49 ±0.01 | 91.83 ±0.11 | 91.22 ±0.10 | 91.83 ±0.04 | 91.81 ±0.30 |
| **LRW** | 90.88 ±0.14 | 91.47 ±0.07 | 91.99 ±0.08 | 91.28 ±0.08 | 91.60 ±0.04 | 91.74 ±0.20 |
| **SLR** | 91.33 ±0.01 | 92.08 ±0.07 | 92.41 ±0.10 | 91.62 ±0.39 | 92.05 ±0.01 | 92.49 ±0.23 |
| **CLR** | 92.05 ±0.08 | 92.47 ±0.08 | 92.69 ±0.42 | 91.92 ±0.11 | 92.73 ±0.09 | 92.69 ±0.08 |
| **LLR** | 91.79 ±0.13 | 92.27 ±0.38 | 92.76 ±0.22 | 92.10 ±0.35 | 92.60 ±0.31 | 92.81 ±0.35 |
| **ALLR** | 91.95 ±0.32 | 92.26 ±0.13 | 92.78 ±0.33 | 92.20 ±0.47 | 92.66 ±0.35 | 92.79 ±0.03 |
| **constant** | 91.10 ±0.21 | 91.45 ±0.11 | 91.86 ±0.10 | 91.20 ±0.13 | 91.52 ±0.04 | 91.88 ±0.19 |
| **stepped** | 91.80 ±0.26 | 92.10 ±0.33 | 92.62 ±0.08 | 91.81 ±0.22 | 92.52 ±0.43 | 92.77 ±0.13 |
| **cosine** | 92.01 ±0.01 | 92.30 ±0.15 | 92.77 ±0.54 | 92.22 ±0.21 | 92.74 ±0.11 | 92.89 ±0.08 |
| **linear** | 92.08 ±0.04 | 92.42 ±0.20 | 92.94 ±0.04 | 92.07 ±0.06 | 92.67 ±0.26 | 92.80 ±0.12 |
| | **Model sparsity 95%** | | | | | |
| Budget: | $2 \times 2.5\%$ | $2 \times 5\%$ | $2 \times 10\%$ | $3 \times 2.5\%$ | $3 \times 5\%$ | $3 \times 10\%$ |
| **FT** | 87.80 ±0.28 | 89.07 ±0.21 | 89.72 ±0.08 | 88.76 ±0.11 | 89.47 ±0.01 | 90.30 ±0.11 |
| **LRW** | 87.81 ±0.18 | 89.15 ±0.10 | 89.86 ±0.09 | 88.72 ±0.04 | 89.66 ±0.19 | 90.24 ±0.31 |
| **SLR** | 89.80 ±0.28 | 90.47 ±0.08 | 91.09 ±0.21 | 90.20 ±0.60 | 91.07 ±0.34 | 91.60 ±0.03 |
| **CLR** | 90.45 ±0.20 | 91.25 ±0.35 | 91.36 ±0.13 | 90.94 ±0.04 | 91.61 ±0.16 | 91.98 ±0.30 |
| **LLR** | 90.82 ±0.30 | 91.03 ±0.04 | 91.27 ±0.13 | 90.90 ±0.56 | 91.42 ±0.00 | 91.96 ±0.22 |
| **ALLR** | 90.69 ±0.08 | 91.39 ±0.02 | 91.44 ±0.51 | 91.05 ±0.09 | 91.56 ±0.17 | 91.90 ±0.04 |
| **constant** | 87.83 ±0.35 | 89.11 ±0.24 | 89.80 ±0.24 | 88.57 ±0.25 | 89.92 ±0.45 | 90.22 ±0.09 |
| **stepped** | 90.17 ±0.25 | 90.79 ±0.21 | 91.39 ±0.12 | 90.25 ±0.13 | 91.10 ±0.11 | 91.75 ±0.21 |
| **cosine** | 90.51 ±0.07 | 91.08 ±0.23 | 91.45 ±0.06 | 91.03 ±0.15 | 91.57 ±0.21 | 92.09 ±0.50 |
| **linear** | 90.56 ±0.19 | 91.14 ±0.08 | 91.48 ±0.23 | 90.95 ±0.28 | 91.34 ±0.18 | 91.81 ±0.36 |
| | **Model sparsity 99%** | | | | | |
| Budget: | $2 \times 2.5\%$ | $2 \times 5\%$ | $2 \times 10\%$ | $3 \times 2.5\%$ | $3 \times 5\%$ | $3 \times 10\%$ |
| **FT** | 65.95 ±1.50 | 71.08 ±1.70 | 76.46 ±0.23 | 75.42 ±0.16 | 78.19 ±0.10 | 79.92 ±0.30 |
| **LRW** | 65.78 ±1.43 | 71.45 ±1.18 | 76.35 ±0.18 | 75.50 ±0.11 | 77.94 ±1.12 | 79.94 ±0.87 |
| **SLR** | 81.11 ±0.69 | 82.11 ±0.39 | 82.60 ±0.17 | 82.60 ±0.04 | 83.50 ±0.15 | 84.34 ±0.35 |
| **CLR** | 81.41 ±0.18 | 82.08 ±0.43 | 82.86 ±0.08 | 83.21 ±0.16 | 84.24 ±0.21 | 84.88 ±0.04 |
| **LLR** | 81.25 ±0.01 | 81.92 ±0.08 | 82.60 ±0.03 | 83.36 ±0.15 | 84.18 ±0.11 | 84.48 ±0.20 |
| **ALLR** | 81.24 ±0.27 | 82.25 ±0.31 | 83.16 ±0.00 | 83.12 ±0.11 | 84.50 ±0.15 | 85.11 ±0.91 |
| **constant** | 73.91 ±0.13 | 75.37 ±0.44 | 78.38 ±0.86 | 78.03 ±1.27 | 80.05 ±0.33 | 80.73 ±0.52 |
| **stepped** | 81.12 ±0.06 | 82.18 ±0.54 | 82.76 ±0.04 | 82.80 ±0.08 | 83.54 ±0.08 | 84.48 ±0.18 |
| **cosine** | 81.42 ±0.23 | 82.09 ±0.17 | 82.56 ±0.30 | 82.89 ±0.62 | 84.16 ±0.24 | 84.97 ±0.43 |
| **linear** | 81.17 ±0.35 | 82.14 ±0.47 | 82.75 ±0.28 | 83.31 ±0.08 | 84.17 ±0.72 | 85.09 ±0.48 |

Table 6: ResNet-18 on CIFAR-10 (One Shot): Performance of the different learning rate translation schemes (above) compared to tuned schedules (below) for IMP in the One Shot setting for target sparsity of 70%, 90%, 95%, 98% and a retrain time of 5%, 10%, and 25% of the initial training budget. The first, second, and third best values for the translation schemes are highlighted. Results are averaged over two seeds with the standard deviation indicated.

| | Model sparsity 70% | | | Model sparsity 90% | | |
|---|---|---|---|---|---|---|
| Budget: | 5% | 10% | 25% | 5% | 10% | 25% |
| **FT** | 94.97 ±0.12 | 95.09 ±0.24 | 95.23 ±0.01 | 94.63 ±0.08 | 94.75 ±0.30 | 94.77 ±0.07 |
| **LRW** | 95.05 ±0.16 | 95.08 ±0.23 | 95.05 ±0.01 | 94.62 ±0.08 | 94.74 ±0.30 | 94.91 ±0.20 |
| **SLR** | 94.21 ±0.10 | 94.82 ±0.16 | 94.86 ±0.06 | 94.24 ±0.07 | 94.66 ±0.20 | 94.90 ±0.41 |
| **CLR** | 94.70 ±0.07 | 95.02 ±0.14 | 95.27 ±0.01 | 94.65 ±0.01 | 94.88 ±0.15 | 94.97 ±0.08 |
| **LLR** | 94.58 ±0.13 | 94.82 ±0.16 | 95.00 ±0.06 | 94.61 ±0.25 | 94.87 ±0.14 | 95.03 ±0.04 |
| **ALLR** | 94.99 ±0.03 | 95.08 ±0.08 | 95.34 ±0.15 | 94.66 ±0.01 | 94.94 ±0.15 | 95.02 ±0.07 |
| **constant** | 94.97 ±0.12 | 95.09 ±0.25 | 95.25 ±0.01 | 94.62 ±0.12 | 94.73 ±0.33 | 94.73 ±0.04 |
| **stepped** | 95.08 ±0.13 | 95.19 ±0.12 | 95.19 ±0.15 | 94.88 ±0.23 | 94.86 ±0.06 | 95.05 ±0.16 |
| **cosine** | 95.08 ±0.18 | 95.16 ±0.16 | 95.35 ±0.01 | 94.95 ±0.13 | 95.15 ±0.24 | 95.20 ±0.35 |
| **linear** | 95.18 ±0.14 | 95.20 ±0.11 | 95.34 ±0.13 | 94.98 ±0.18 | 95.06 ±0.26 | 95.18 ±0.20 |
| | Model sparsity 95% | | | Model sparsity 98% | | |
| Budget: | 5% | 10% | 25% | 5% | 10% | 25% |
| **FT** | 93.72 ±0.14 | 94.11 ±0.17 | 94.14 ±0.06 | 91.01 ±0.54 | 92.38 ±0.05 | 93.03 ±0.13 |
| **LRW** | 93.74 ±0.10 | 94.09 ±0.18 | 94.49 ±0.06 | 91.01 ±0.58 | 92.37 ±0.03 | 93.95 ±0.13 |
| **SLR** | 94.26 ±0.17 | 94.38 ±0.03 | 94.52 ±0.08 | 93.18 ±0.08 | 93.62 ±0.12 | 93.61 ±0.16 |
| **CLR** | 94.38 ±0.35 | 94.60 ±0.02 | 94.79 ±0.18 | 93.41 ±0.25 | 94.06 ±0.05 | 93.95 ±0.11 |
| **LLR** | 94.45 ±0.19 | 94.71 ±0.01 | 94.81 ±0.13 | 93.50 ±0.01 | 93.72 ±0.25 | 94.22 ±0.04 |
| **ALLR** | 94.43 ±0.13 | 94.59 ±0.25 | 94.68 ±0.07 | 93.48 ±0.06 | 93.86 ±0.23 | 93.97 ±0.08 |
| **constant** | 93.69 ±0.11 | 94.15 ±0.11 | 94.19 ±0.06 | 92.09 ±0.21 | 92.38 ±0.13 | 93.14 ±0.16 |
| **stepped** | 94.42 ±0.13 | 94.55 ±0.18 | 94.72 ±0.28 | 93.44 ±0.18 | 93.82 ±0.13 | 93.86 ±0.22 |
| **cosine** | 94.50 ±0.05 | 94.73 ±0.01 | 94.80 ±0.01 | 93.64 ±0.13 | 94.01 ±0.10 | 94.06 ±0.02 |
| **linear** | 94.68 ±0.17 | 94.70 ±0.01 | 94.98 ±0.10 | 93.67 ±0.10 | 94.09 ±0.01 | 94.22 ±0.03 |

Table 7: WideResNet on CIFAR-100 (One Shot): Performance of the different learning rate translation schemes (above) compared to tuned schedules (below) for IMP in the One Shot setting for target sparsity of 90%, 95%, 99% and a retrain time of 2.5%, 5%, 10% of the initial training budget. The **first**, **second**, and **third** best values for the translation schemes are highlighted. Results are averaged over two seeds with the standard deviation indicated.

| | Model sparsity 90% | | |
|---|---|---|---|
| Budget: | 2.5% | 5% | 10% |
| FT | 73.96 ±0.28 | 74.70 ±0.08 | 74.68 ±0.18 |
| LRW | 73.95 ±0.24 | 74.70 ±0.07 | 74.67 ±0.19 |
| SLR | 75.82 ±0.48 | 76.09 ±0.26 | 76.05 ±0.16 |
| CLR | 75.69 ±0.19 | 76.00 ±0.17 | 75.96 ±0.06 |
| LLR | 75.62 ±0.02 | 76.25 ±0.07 | 75.82 ±0.36 |
| ALLR | 75.71 ±0.05 | 75.84 ±0.07 | 75.85 ±0.41 |
| constant | 74.95 ±0.21 | 74.87 ±0.85 | 75.44 ±0.70 |
| stepped | 77.23 ±0.36 | 77.40 ±0.14 | 77.72 ±0.45 |
| cosine | 77.34 ±0.15 | 77.85 ±0.28 | 77.76 ±0.01 |
| linear | 77.22 ±0.21 | 77.20 ±0.25 | 77.42 ±0.02 |
| | Model sparsity 95% | | |
| Budget: | 2.5% | 5% | 10% |
| FT | 68.32 ±0.16 | 69.54 ±0.43 | 70.11 ±0.43 |
| LRW | 68.28 ±0.14 | 69.59 ±0.38 | 70.09 ±0.54 |
| SLR | 75.10 ±0.35 | 75.73 ±0.35 | 75.49 ±0.21 |
| CLR | 75.40 ±0.18 | 75.62 ±0.35 | 75.51 ±0.13 |
| LLR | 75.32 ±0.29 | 75.95 ±0.26 | 75.69 ±0.29 |
| ALLR | 75.02 ±0.69 | 75.84 ±0.25 | 75.61 ±0.21 |
| constant | 70.41 ±0.34 | 70.86 ±1.11 | 71.27 ±0.88 |
| stepped | 75.92 ±0.08 | 77.09 ±0.28 | 77.67 ±0.21 |
| cosine | 76.33 ±0.02 | 77.20 ±0.35 | 77.35 ±0.66 |
| linear | 76.58 ±0.28 | 76.95 ±0.31 | 77.17 ±0.52 |
| | Model sparsity 99% | | |
| Budget: | 2.5% | 5% | 10% |
| FT | 16.24 ±1.12 | 17.92 ±2.20 | 19.63 ±0.96 |
| LRW | 16.27 ±1.17 | 17.87 ±1.97 | 19.79 ±1.19 |
| SLR | 60.51 ±1.38 | 64.50 ±0.09 | 68.40 ±0.57 |
| CLR | 62.91 ±0.56 | 66.27 ±0.59 | 68.67 ±0.84 |
| LLR | 63.09 ±0.65 | 66.24 ±0.57 | 68.77 ±0.64 |
| ALLR | 62.93 ±0.53 | 65.74 ±0.51 | 68.72 ±0.04 |
| constant | 56.85 ±1.45 | 59.58 ±1.61 | 61.05 ±0.58 |
| stepped | 61.64 ±0.20 | 65.93 ±0.28 | 69.63 ±0.59 |
| cosine | 63.52 ±0.21 | 67.22 ±0.19 | 70.23 ±0.69 |
| linear | 63.93 ±0.36 | 67.34 ±0.70 | 69.50 ±0.71 |

Table 8: ResNet-50 on ImageNet (One Shot): Performance of the different learning rate translation schemes for IMP in the One Shot setting for target sparsity of 70%, 80%, 90%, 95% and a retrain time of 2.22% (2 epochs), 5.55% (5 epochs), 11.11% (10 epochs), 22.22% (20 epochs) of the initial training budget. The first, second, and third best values for the translation schemes are highlighted. Results are averaged over two seeds with the standard deviation indicated.

| | Model sparsity 70% | | | |
|---|---|---|---|---|
| Budget: | 2.22% | 5.55% | 11.11% | 22.22% |
| FT | 73.51 ±0.04 | 73.98 ±0.04 | 74.44 ±0.11 | 74.75 ±0.03 |
| LRW | 73.50 ±0.04 | 73.99 ±0.04 | 74.45 ±0.11 | 75.36 ±0.03 |
| SLR | 70.93 ±0.01 | 72.58 ±0.03 | 73.69 ±0.11 | 74.59 ±0.05 |
| CLR | 72.22 ±0.09 | 73.58 ±0.08 | 74.49 ±0.04 | 74.98 ±0.07 |
| LLR | 72.39 ±0.13 | 73.65 ±0.05 | 74.34 ±0.02 | 74.80 ±0.07 |
| ALLR | 73.69 ±0.03 | 74.37 ±0.05 | 74.89 ±0.04 | 75.27 ±0.16 |
| | Model sparsity 80% | | | |
| Budget: | 2.22% | 5.55% | 11.11% | 22.22% |
| FT | 70.45 ±0.20 | 71.81 ±0.11 | 72.68 ±0.07 | 72.82 ±0.13 |
| LRW | 70.45 ±0.20 | 71.82 ±0.12 | 72.67 ±0.07 | 74.24 ±0.02 |
| SLR | 70.48 ±0.04 | 72.37 ±0.02 | 73.44 ±0.18 | 73.95 ±0.13 |
| CLR | 71.96 ±0.09 | 73.30 ±0.08 | 74.24 ±0.08 | 74.38 ±0.05 |
| LLR | 72.07 ±0.09 | 73.41 ±0.05 | 74.23 ±0.10 | 74.12 ±0.02 |
| ALLR | 72.96 ±0.15 | 74.02 ±0.08 | 74.71 ±0.04 | 74.43 ±0.09 |
| | Model sparsity 90% | | | |
| Budget: | 2.22% | 5.55% | 11.11% | 22.22% |
| FT | 56.75 ±0.01 | 61.60 ±0.30 | 64.61 ±0.21 | 65.90 ±0.10 |
| LRW | 56.75 ±0.01 | 61.61 ±0.30 | 64.60 ±0.23 | 70.41 ±0.13 |
| SLR | 67.19 ±0.23 | 69.45 ±0.01 | 70.80 ±0.09 | 71.24 ±0.04 |
| CLR | 68.72 ±0.06 | 70.60 ±0.15 | 71.51 ±0.13 | 71.68 ±0.05 |
| LLR | 68.90 ±0.05 | 70.48 ±0.01 | 71.53 ±0.09 | 71.45 ±0.13 |
| ALLR | 69.56 ±0.07 | 71.19 ±0.01 | 71.99 ±0.07 | 71.76 ±0.09 |
| | Model sparsity 95% | | | |
| Budget: | 2.22% | 5.55% | 11.11% | 22.22% |
| FT | 27.24 ±0.30 | 40.91 ±0.05 | 48.56 ±0.17 | 51.69 ±0.08 |
| LRW | 27.24 ±0.30 | 40.91 ±0.04 | 48.56 ±0.17 | 62.55 ±0.09 |
| SLR | 61.42 ±0.10 | 64.60 ±0.17 | 66.31 ±0.00 | 66.93 ±0.16 |
| CLR | 63.35 ±0.11 | 66.07 ±0.12 | 67.34 ±0.09 | 67.44 ±0.03 |
| LLR | 63.58 ±0.06 | 66.02 ±0.16 | 67.20 ±0.17 | 67.34 ±0.04 |
| ALLR | 63.75 ±0.03 | 66.22 ±0.23 | 67.43 ±0.10 | 67.75 ±0.13 |

Table 9: ResNet-50 on ImageNet (Iterative): Performance of the different learning rate translation schemes for IMP in the iterative setting for target sparsity of 70%, 80%, 90% and retrain times as indicated in the Budget row. Here $2 \times 2.22\%$ indicates two prune-retrain cycles, each of which having length equal to 2.22% of the overall training budget. The **first**, **second**, and **third** best values for the translation schemes are highlighted. Results are averaged over two seeds with the standard deviation indicated.

| | | | Model sparsity 70% | | | |
|---|---|---|---|---|---|---|
| Budget: | $2 \times 2.22\%$ | $2 \times 5.55\%$ | $2 \times 11.11\%$ | $3 \times 2.22\%$ | $3 \times 5.55\%$ | $3 \times 11.11\%$ |
| FT | 73.59 ±0.04 | 74.16 ±0.02 | 74.50 ±0.10 | 73.77 ±0.01 | 74.24 ±0.03 | 74.67 ±0.09 |
| LRW | 73.59 ±0.04 | 74.16 ±0.02 | 74.50 ±0.10 | 73.77 ±0.01 | 74.24 ±0.03 | 74.67 ±0.09 |
| SLR | 70.71 ±0.14 | 72.36 ±0.06 | 73.67 ±0.05 | 70.94 ±0.14 | 72.35 ±0.05 | 73.55 ±0.06 |
| CLR | 72.30 ±0.04 | 73.52 ±0.00 | 74.43 ±0.08 | 72.27 ±0.05 | 73.50 ±0.01 | 74.56 ±0.11 |
| LLR | 72.46 ±0.15 | 73.59 ±0.05 | 74.42 ±0.10 | 72.40 ±0.02 | 73.80 ±0.08 | 74.63 ±0.06 |
| ALLR | 74.19 ±0.08 | 74.80 ±0.01 | 75.19 ±0.03 | 73.47 ±0.12 | 74.70 ±0.15 | 75.24 ±0.08 |
| | | | Model sparsity 80% | | | |
| Budget: | $2 \times 2.22\%$ | $2 \times 5.55\%$ | $2 \times 11.11\%$ | $3 \times 2.22\%$ | $3 \times 5.55\%$ | $3 \times 11.11\%$ |
| FT | 70.14 ±0.02 | 71.55 ±0.04 | 72.48 ±0.07 | 70.63 ±0.08 | 71.85 ±0.04 | 72.82 ±0.12 |
| LRW | 70.14 ±0.02 | 71.55 ±0.04 | 72.48 ±0.07 | 70.63 ±0.08 | 71.85 ±0.04 | 72.82 ±0.12 |
| SLR | 69.98 ±0.13 | 71.96 ±0.13 | 73.15 ±0.06 | 70.33 ±0.05 | 72.05 ±0.15 | 73.13 ±0.04 |
| CLR | 71.75 ±0.04 | 73.09 ±0.06 | 74.09 ±0.13 | 71.89 ±0.02 | 73.29 ±0.00 | 74.27 ±0.05 |
| LLR | 71.81 ±0.04 | 73.32 ±0.02 | 73.91 ±0.04 | 72.04 ±0.05 | 73.29 ±0.04 | 74.34 ±0.04 |
| ALLR | 73.10 ±0.02 | 73.99 ±0.25 | 74.69 ±0.16 | 72.50 ±0.16 | 74.48 ±0.06 | 75.02 ±0.04 |
| | | | Model sparsity 90% | | | |
| Budget: | $2 \times 2.22\%$ | $2 \times 5.55\%$ | $2 \times 11.11\%$ | $3 \times 2.22\%$ | $3 \times 5.55\%$ | $3 \times 11.11\%$ |
| FT | 56.96 ±0.06 | 61.90 ±0.12 | 64.80 ±0.12 | 59.42 ±0.17 | 63.72 ±0.15 | 66.30 ±0.14 |
| LRW | 56.96 ±0.06 | 61.90 ±0.12 | 64.80 ±0.12 | 59.42 ±0.17 | 63.72 ±0.15 | 66.30 ±0.14 |
| SLR | 68.16 ±0.02 | 70.16 ±0.15 | 71.68 ±0.07 | 68.49 ±0.13 | 70.61 ±0.13 | 72.09 ±0.12 |
| CLR | 69.69 ±0.01 | 71.38 ±0.01 | 72.41 ±0.04 | 70.18 ±0.01 | 71.79 ±0.12 | 72.93 ±0.05 |
| LLR | 69.88 ±0.24 | 71.35 ±0.07 | 72.24 ±0.09 | 70.29 ±0.07 | 71.85 ±0.04 | 72.91 ±0.08 |
| ALLR | 70.00 ±0.08 | 71.77 ±0.05 | 72.75 ±0.05 | 70.23 ±0.16 | 71.45 ±0.13 | 73.14 ±0.07 |

Table 10: MaxViT on ImageNet (One Shot): Performance of the different learning rate translation schemes for IMP in the One Shot setting for target sparsity of 75%, 80%, 85%, 90% and a retrain time of 1% (2 epochs), 2.5% (5 epochs), 5% (10 epochs), 10% (20 epochs) of the initial training budget. The **first**, **second**, and **third** best values for the translation schemes are highlighted. Results are averaged over two seeds with the standard deviation indicated.

| | | Model sparsity 75% | | |
|---|---|---|---|---|
| Budget: | 1% | 2.5% | 5% | 10% |
| FT | 76.64 ±0.58 | 76.90 ±0.15 | 77.04 ±0.23 | 77.02 ±0.45 |
| LRW | 76.64 ±0.58 | 76.90 ±0.15 | 77.04 ±0.23 | 77.02 ±0.45 |
| SLR | 75.29 ±0.29 | 76.31 ±0.21 | 76.68 ±0.16 | 76.99 ±0.66 |
| CLR | 76.10 ±0.49 | 76.98 ±0.05 | 77.33 ±0.15 | 77.52 ±0.54 |
| LLR | 76.21 ±0.45 | 77.04 ±0.00 | 77.39 ±0.13 | 77.50 ±0.52 |
| ALLR | 77.11 ±0.63 | 77.59 ±0.09 | 77.62 ±0.01 | 77.39 ±0.55 |
| | | Model sparsity 80% | | |
| Budget: | 1% | 2.5% | 5% | 10% |
| FT | 75.10 ±0.76 | 75.79 ±0.29 | 75.94 ±0.08 | 76.23 ±0.67 |
| LRW | 75.10 ±0.76 | 75.79 ±0.29 | 75.94 ±0.08 | 76.23 ±0.67 |
| SLR | 74.89 ±0.45 | 75.79 ±0.07 | 76.37 ±0.12 | 76.59 ±0.63 |
| CLR | 75.73 ±0.44 | 76.61 ±0.09 | 77.13 ±0.23 | 77.22 ±0.64 |
| LLR | 75.82 ±0.52 | 76.57 ±0.02 | 77.22 ±0.07 | 77.21 ±0.64 |
| ALLR | 76.70 ±0.60 | 77.21 ±0.04 | 77.37 ±0.08 | 77.31 ±0.66 |
| | | Model sparsity 85% | | |
| Budget: | 1% | 2.5% | 5% | 10% |
| FT | 71.49 ±0.95 | 72.97 ±0.40 | 73.61 ±0.24 | 74.14 ±0.58 |
| LRW | 71.49 ±0.95 | 72.97 ±0.40 | 73.61 ±0.24 | 74.14 ±0.58 |
| SLR | 74.02 ±0.29 | 75.15 ±0.21 | 75.75 ±0.09 | 75.86 ±0.69 |
| CLR | 74.85 ±0.40 | 76.11 ±0.18 | 76.55 ±0.06 | 76.59 ±0.58 |
| LLR | 75.05 ±0.34 | 76.05 ±0.03 | 76.56 ±0.02 | 76.59 ±0.51 |
| ALLR | 75.79 ±0.55 | 76.55 ±0.18 | 76.85 ±0.02 | 76.74 ±0.65 |
| | | Model sparsity 90% | | |
| Budget: | 1% | 2.5% | 5% | 10% |
| FT | 60.03 ±1.76 | 64.56 ±0.67 | 66.62 ±0.75 | 68.42 ±0.77 |
| LRW | 60.03 ±1.76 | 64.56 ±0.67 | 66.62 ±0.75 | 68.42 ±0.77 |
| SLR | 71.44 ±0.10 | 73.03 ±0.11 | 73.87 ±0.03 | 74.14 ±0.64 |
| CLR | 72.62 ±0.05 | 74.25 ±0.09 | 74.97 ±0.03 | 75.10 ±0.64 |
| LLR | 72.82 ±0.10 | 74.27 ±0.26 | 74.96 ±0.10 | 75.03 ±0.72 |
| ALLR | 73.43 ±0.26 | 74.66 ±0.22 | 75.22 ±0.04 | 75.28 ±0.53 |

Table 11: MaxViT on ImageNet (Iterative): Performance of the different learning rate translation schemes for IMP in the iterative setting for target sparsity of 75%, 80%, 85%, 90% and retrain times as indicated in the Budget row. Here $2 \times 2.5\%$ indicates two prune-retrain cycles, each of which having length equal to 2.5% of the overall training budget. The **first**, **second**, and **third** best values for the translation schemes are highlighted. Results are averaged over two seeds with the standard deviation indicated.

| | **Model sparsity 75%** | | | |
|---|---|---|---|---|
| Budget: | $2 \times 1\%$ | $2 \times 2.5\%$ | $3 \times 1\%$ | $3 \times 2.5\%$ |
| **FT** | 76.14 ±0.21 | 76.84 ±0.19 | 76.85 ±0.36 | 76.93 ±0.24 |
| **LRW** | 76.14 ±0.21 | 76.84 ±0.19 | 76.85 ±0.36 | 76.93 ±0.24 |
| **SLR** | 74.89 ±0.07 | 76.08 ±0.10 | 75.32 ±0.71 | 75.98 ±0.29 |
| **CLR** | 75.93 ±0.08 | 76.97 ±0.04 | 76.43 ±0.40 | 76.97 ±0.12 |
| **LLR** | 76.14 ±0.11 | 77.23 ±0.19 | 76.58 ±0.31 | 77.13 ±0.07 |
| **ALLR** | 76.81 ±0.28 | 77.56 ±0.25 | 77.37 ±0.55 | 77.32 ±0.27 |
| | **Model sparsity 80%** | | | |
| Budget: | $2 \times 1\%$ | $2 \times 2.5\%$ | $3 \times 1\%$ | $3 \times 2.5\%$ |
| **FT** | 74.75 ±0.24 | 75.69 ±0.13 | 75.48 ±0.30 | 75.85 ±0.19 |
| **LRW** | 74.75 ±0.24 | 75.69 ±0.13 | 75.48 ±0.30 | 75.85 ±0.19 |
| **SLR** | 74.46 ±0.12 | 75.76 ±0.03 | 74.92 ±0.47 | 75.59 ±0.13 |
| **CLR** | 75.49 ±0.01 | 76.68 ±0.20 | 76.11 ±0.29 | 76.52 ±0.04 |
| **LLR** | 75.66 ±0.04 | 76.78 ±0.07 | 76.10 ±0.31 | 76.80 ±0.15 |
| **ALLR** | 76.45 ±0.14 | 77.11 ±0.22 | 77.01 ±0.42 | 77.15 ±0.36 |
| | **Model sparsity 85%** | | | |
| Budget: | $2 \times 1\%$ | $2 \times 2.5\%$ | $3 \times 1\%$ | $3 \times 2.5\%$ |
| **FT** | 71.07 ±0.45 | 72.85 ±0.36 | 72.14 ±0.12 | 73.12 ±0.06 |
| **LRW** | 71.07 ±0.45 | 72.85 ±0.36 | 72.14 ±0.12 | 73.12 ±0.06 |
| **SLR** | 73.62 ±0.11 | 75.05 ±0.09 | 73.95 ±0.43 | 74.85 ±0.24 |
| **CLR** | 74.68 ±0.20 | 76.09 ±0.10 | 75.34 ±0.40 | 75.95 ±0.31 |
| **LLR** | 74.89 ±0.27 | 76.22 ±0.17 | 75.47 ±0.40 | 76.21 ±0.02 |
| **ALLR** | 75.43 ±0.08 | 76.45 ±0.08 | 76.14 ±0.33 | 76.53 ±0.12 |
| | **Model sparsity 90%** | | | |
| Budget: | $2 \times 1\%$ | $2 \times 2.5\%$ | $3 \times 1\%$ | $3 \times 2.5\%$ |
| **FT** | 59.72 ±1.15 | 64.38 ±0.64 | 61.75 ±0.35 | 64.96 ±0.20 |
| **LRW** | 59.72 ±1.15 | 64.38 ±0.64 | 61.75 ±0.35 | 64.96 ±0.20 |
| **SLR** | 71.04 ±0.19 | 73.04 ±0.20 | 72.12 ±0.43 | 73.50 ±0.00 |
| **CLR** | 72.39 ±0.36 | 74.33 ±0.16 | 73.31 ±0.32 | 74.47 ±0.17 |
| **LLR** | 72.58 ±0.16 | 74.36 ±0.00 | 73.69 ±0.05 | 74.78 ±0.03 |
| **ALLR** | 72.89 ±0.27 | 74.65 ±0.03 | 73.94 ±0.02 | 74.84 ±0.11 |

Table 12: DeepLabV3 on COCO (One Shot): Performance (measured as mIoU) of the different learning rate translation schemes for IMP in the One Shot setting for target sparsity of 50% - 80% and a retrain time of 3.33% (1 epochs), 6.66% (2 epochs), 16.66% (5 epochs) of the initial training budget. The **first**, **second**, and **third** best values for the translation schemes are highlighted. Results are averaged over two seeds with the standard deviation indicated.

| Model sparsity 50% | | |
|---|---|---|
| Budget: | 3.33% | 6.66% | 16.66% |

| | 3.33% | 6.66% | 16.66% |
|---|---|---|---|
| **FT** | 63.10 ±0.15 | 63.07 ±0.08 | 63.11 ±0.07 |
| **LRW** | 63.10 ±0.15 | 63.07 ±0.08 | 63.13 ±0.10 |
| **SLR** | 60.84 ±0.66 | 61.14 ±0.16 | 62.12 ±0.16 |
| **CLR** | 61.72 ±1.03 | 62.72 ±0.79 | 63.13 ±0.26 |
| **LLR** | 61.73 ±1.09 | 62.69 ±1.11 | 63.41 ±0.37 |
| **ALLR** | 63.02 ±0.36 | 63.21 ±0.27 | 63.58 ±0.08 |

| Model sparsity 60% | | |
|---|---|---|
| Budget: | 3.33% | 6.66% | 16.66% |

| | 3.33% | 6.66% | 16.66% |
|---|---|---|---|
| **FT** | 62.89 ±0.17 | 62.89 ±0.07 | 62.99 ±0.06 |
| **LRW** | 62.89 ±0.17 | 62.89 ±0.07 | 63.07 ±0.08 |
| **SLR** | 60.82 ±0.69 | 61.56 ±0.51 | 62.33 ±0.17 |
| **CLR** | 61.93 ±0.90 | 62.63 ±0.93 | 63.17 ±0.03 |
| **LLR** | 61.90 ±0.92 | 62.83 ±0.97 | 63.20 ±0.49 |
| **ALLR** | 62.87 ±0.43 | 63.09 ±0.32 | 63.46 ±0.12 |

| Model sparsity 70% | | |
|---|---|---|
| Budget: | 3.33% | 6.66% | 16.66% |

| | 3.33% | 6.66% | 16.66% |
|---|---|---|---|
| **FT** | 62.24 ±0.22 | 62.31 ±0.11 | 62.56 ±0.05 |
| **LRW** | 62.24 ±0.22 | 62.31 ±0.11 | 62.75 ±0.06 |
| **SLR** | 60.82 ±0.39 | 61.32 ±0.81 | 62.11 ±0.04 |
| **CLR** | 61.66 ±0.74 | 62.56 ±0.85 | 62.99 ±0.00 |
| **LLR** | 61.62 ±0.78 | 62.64 ±0.66 | 63.11 ±0.41 |
| **ALLR** | 62.53 ±0.54 | 62.78 ±0.45 | 63.08 ±0.06 |

| Model sparsity 80% | | |
|---|---|---|
| Budget: | 3.33% | 6.66% | 16.66% |

| | 3.33% | 6.66% | 16.66% |
|---|---|---|---|
| **FT** | 59.84 ±0.09 | 60.30 ±0.18 | 61.08 ±0.00 |
| **LRW** | 59.84 ±0.09 | 60.31 ±0.18 | 61.80 ±0.24 |
| **SLR** | 60.04 ±0.79 | 60.91 ±0.25 | 61.69 ±0.59 |
| **CLR** | 61.02 ±0.95 | 61.74 ±0.68 | 62.69 ±0.22 |
| **LLR** | 61.24 ±0.90 | 61.85 ±0.47 | 62.62 ±0.21 |
| **ALLR** | 61.68 ±0.75 | 62.05 ±0.93 | 62.55 ±0.03 |

Table 13: DeepLabV3 on COCO (Iterative): Performance of the different learning rate translation schemes for IMP in the iterative setting for target sparsity of 50 - 80% and retrain times as indicated in the Budget row. Here $2 \times 6.66\%$ indicates two prune-retrain cycles, each of which having length equal to 6.66% of the overall training budget. The **first**, **second**, and **third** best values for the translation schemes are highlighted. Results are averaged over two seeds with the standard deviation indicated.

| | Model sparsity 50% | | | |
|---|---|---|---|---|
| Budget: | $2 \times 6.66\%$ | $2 \times 10\%$ | $3 \times 6.66\%$ | $3 \times 10\%$ |
| **FT** | 63.10 ±0.01 | 63.26 ±0.39 | 63.18 ±0.10 | 63.30 ±0.38 |
| **LRW** | 63.10 ±0.01 | 63.26 ±0.39 | 63.18 ±0.10 | 63.30 ±0.38 |
| **SLR** | 61.50 ±0.69 | 61.86 ±0.54 | 61.54 ±0.45 | 62.39 ±0.35 |
| **CLR** | 62.72 ±0.42 | 62.65 ±0.27 | 62.70 ±0.07 | 63.16 ±0.34 |
| **LLR** | 62.97 ±0.63 | 62.85 ±0.14 | 63.06 ±0.28 | 63.32 ±0.44 |
| **ALLR** | 63.27 ±0.38 | 63.02 ±0.69 | 63.35 ±0.20 | 63.44 ±0.29 |
| | Model sparsity 60% | | | |
| Budget: | $2 \times 6.66\%$ | $2 \times 10\%$ | $3 \times 6.66\%$ | $3 \times 10\%$ |
| **FT** | 62.93 ±0.04 | 63.12 ±0.47 | 63.03 ±0.17 | 63.19 ±0.46 |
| **LRW** | 62.93 ±0.04 | 63.12 ±0.47 | 63.03 ±0.17 | 63.19 ±0.46 |
| **SLR** | 61.44 ±0.53 | 62.02 ±0.01 | 61.81 ±0.02 | 62.70 ±0.00 |
| **CLR** | 62.65 ±0.55 | 62.67 ±0.04 | 62.69 ±0.17 | 63.35 ±0.23 |
| **LLR** | 62.82 ±0.48 | 62.52 ±0.38 | 62.86 ±0.23 | 63.43 ±0.17 |
| **ALLR** | 63.19 ±0.34 | 63.00 ±0.53 | 63.27 ±0.16 | 63.38 ±0.37 |
| | Model sparsity 70% | | | |
| Budget: | $2 \times 6.66\%$ | $2 \times 10\%$ | $3 \times 6.66\%$ | $3 \times 10\%$ |
| **FT** | 62.33 ±0.04 | 62.58 ±0.50 | 62.51 ±0.11 | 62.71 ±0.37 |
| **LRW** | 62.33 ±0.04 | 62.58 ±0.50 | 62.51 ±0.11 | 62.71 ±0.37 |
| **SLR** | 61.57 ±0.18 | 62.00 ±0.11 | 61.83 ±0.15 | 62.17 ±0.10 |
| **CLR** | 62.58 ±0.77 | 62.51 ±0.01 | 62.64 ±0.02 | 62.72 ±0.13 |
| **LLR** | 62.78 ±0.64 | 62.38 ±0.29 | 63.10 ±0.23 | 63.05 ±0.32 |
| **ALLR** | 62.81 ±0.30 | 62.61 ±0.51 | 62.87 ±0.23 | 63.17 ±0.25 |
| | Model sparsity 80% | | | |
| Budget: | $2 \times 6.66\%$ | $2 \times 10\%$ | $3 \times 6.66\%$ | $3 \times 10\%$ |
| **FT** | 60.20 ±0.07 | 60.86 ±0.47 | 60.53 ±0.19 | 60.85 ±0.55 |
| **LRW** | 60.20 ±0.07 | 60.86 ±0.47 | 60.53 ±0.19 | 60.85 ±0.55 |
| **SLR** | 60.93 ±0.87 | 61.56 ±0.23 | 61.13 ±0.04 | 61.37 ±0.41 |
| **CLR** | 62.05 ±0.86 | 61.98 ±0.21 | 62.04 ±0.38 | 62.27 ±0.38 |
| **LLR** | 62.29 ±1.05 | 61.96 ±0.23 | 62.50 ±0.26 | 62.61 ±0.13 |
| **ALLR** | 62.21 ±0.81 | 61.77 ±0.72 | 62.19 ±0.22 | 62.59 ±0.18 |

Table 14: PSPNet on CityScapes (One Shot): Performance (measured as mIoU) of the different learning rate translation schemes for IMP in the One Shot setting for target sparsity of 60% - 90% and a retrain time of 0.66% (2 epochs), 1.66% (5 epochs), 3.33% (10 epochs), 5% (15 epochs) of the initial training budget. The **first**, **second**, and **third** best values for the translation schemes are highlighted. Results are averaged over two seeds with the standard deviation indicated.

| | **Model sparsity 60%** | | | |
|---|---|---|---|---|
| Budget: | 0.66% | 1.66% | 3.33% | 5% |
| **FT** | 58.03 ±0.08 | 58.15 ±0.22 | 58.18 ±0.07 | 58.12 ±0.10 |
| **LRW** | 58.03 ±0.08 | 58.15 ±0.22 | 58.18 ±0.07 | 58.11 ±0.13 |
| **SLR** | 57.27 ±0.12 | 57.70 ±0.79 | 58.10 ±0.07 | 57.95 ±0.19 |
| **CLR** | 57.58 ±0.22 | 57.80 ±0.25 | 58.23 ±0.16 | 58.18 ±0.72 |
| **LLR** | 57.52 ±0.33 | 58.17 ±0.53 | 57.84 ±0.05 | 58.30 ±0.02 |
| **ALLR** | 57.91 ±0.22 | 58.26 ±0.20 | 58.28 ±0.09 | 58.41 ±0.21 |
| | **Model sparsity 70%** | | | |
| Budget: | 0.66% | 1.66% | 3.33% | 5% |
| **FT** | 56.46 ±0.07 | 56.71 ±0.16 | 56.84 ±0.04 | 56.85 ±0.04 |
| **LRW** | 56.46 ±0.07 | 56.71 ±0.16 | 56.84 ±0.04 | 57.16 ±0.05 |
| **SLR** | 56.58 ±0.24 | 57.52 ±0.47 | 58.00 ±0.05 | 58.02 ±0.07 |
| **CLR** | 57.25 ±0.27 | 57.96 ±0.04 | 57.74 ±0.42 | 58.05 ±0.32 |
| **LLR** | 57.37 ±0.32 | 58.06 ±0.28 | 58.18 ±0.23 | 58.49 ±0.09 |
| **ALLR** | 57.67 ±0.10 | 58.09 ±0.08 | 58.04 ±0.29 | 58.30 ±0.30 |
| | **Model sparsity 80%** | | | |
| Budget: | 0.66% | 1.66% | 3.33% | 5% |
| **FT** | 49.59 ±0.18 | 51.08 ±0.19 | 52.17 ±0.15 | 52.77 ±0.38 |
| **LRW** | 49.59 ±0.19 | 51.08 ±0.20 | 52.17 ±0.15 | 54.34 ±0.37 |
| **SLR** | 55.88 ±0.59 | 57.07 ±0.10 | 57.41 ±0.01 | 57.87 ±0.34 |
| **CLR** | 56.68 ±0.12 | 57.68 ±0.37 | 57.84 ±0.11 | 58.16 ±0.12 |
| **LLR** | 56.78 ±0.18 | 57.53 ±0.46 | 57.95 ±0.07 | 58.31 ±0.38 |
| **ALLR** | 56.89 ±0.08 | 57.40 ±0.15 | 57.57 ±0.21 | 58.05 ±0.11 |
| | **Model sparsity 90%** | | | |
| Budget: | 0.66% | 1.66% | 3.33% | 5% |
| **FT** | 32.54 ±0.34 | 35.85 ±0.22 | 39.14 ±0.02 | 41.00 ±0.46 |
| **LRW** | 32.54 ±0.34 | 35.85 ±0.23 | 39.14 ±0.02 | 45.57 ±0.46 |
| **SLR** | 54.76 ±1.59 | 56.43 ±0.52 | 56.50 ±0.02 | 56.53 ±0.09 |
| **CLR** | 55.96 ±0.19 | 56.77 ±0.38 | 56.89 ±0.30 | 57.34 ±0.52 |
| **LLR** | 55.75 ±0.23 | 56.99 ±0.26 | 56.70 ±0.14 | 57.59 ±0.38 |
| **ALLR** | 56.12 ±0.12 | 56.48 ±0.07 | 56.85 ±0.32 | 57.25 ±0.06 |

Table 15: PSPNet on CityScapes (Iterative): Performance of the different learning rate translation schemes for IMP in the iterative setting for target sparsity of 60 - 90% and retrain times as indicated in the Budget row. Here $2 \times 1.66\%$ indicates two prune-retrain cycles, each of which having length equal to 1.66% of the overall training budget. The **first**, **second**, and **third** best values for the translation schemes are highlighted. Results are averaged over two seeds with the standard deviation indicated.

| | | Model sparsity 60% | | |
|---|---|---|---|---|
| Budget: | $2 \times 0.66\%$ | $2 \times 1.66\%$ | $3 \times 0.66\%$ | $3 \times 1.66\%$ |
| **FT** | 58.14 ±0.19 | 58.17 ±0.09 | 58.18 ±0.07 | 58.16 ±0.06 |
| **LRW** | 58.14 ±0.19 | 58.17 ±0.09 | 58.18 ±0.07 | 58.16 ±0.06 |
| **SLR** | 57.24 ±0.29 | 57.50 ±0.31 | 57.80 ±0.45 | 55.79 ±1.42 |
| **CLR** | 58.17 ±0.21 | 58.15 ±0.28 | 57.98 ±0.19 | 57.49 ±0.42 |
| **LLR** | 58.29 ±0.33 | 57.60 ±1.32 | 58.31 ±0.29 | 58.23 ±0.81 |
| **ALLR** | 58.47 ±0.32 | 58.46 ±0.17 | 58.59 ±0.20 | 58.51 ±0.40 |
| | | Model sparsity 70% | | |
| Budget: | $2 \times 0.66\%$ | $2 \times 1.66\%$ | $3 \times 0.66\%$ | $3 \times 1.66\%$ |
| **FT** | 56.62 ±0.10 | 56.79 ±0.04 | 56.60 ±0.19 | 56.75 ±0.12 |
| **LRW** | 56.62 ±0.10 | 56.79 ±0.04 | 56.60 ±0.19 | 56.75 ±0.12 |
| **SLR** | 57.24 ±0.48 | 57.63 ±0.28 | 57.06 ±0.08 | 56.68 ±0.18 |
| **CLR** | 58.18 ±0.10 | 57.07 ±0.59 | 58.05 ±0.04 | 57.93 ±0.55 |
| **LLR** | 58.09 ±0.08 | 58.13 ±0.72 | 58.07 ±0.14 | 58.47 ±0.97 |
| **ALLR** | 58.28 ±0.02 | 58.28 ±0.10 | 58.34 ±0.06 | 58.41 ±0.35 |
| | | Model sparsity 80% | | |
| Budget: | $2 \times 0.66\%$ | $2 \times 1.66\%$ | $3 \times 0.66\%$ | $3 \times 1.66\%$ |
| **FT** | 50.05 ±0.24 | 51.25 ±0.30 | 49.71 ±0.73 | 51.22 ±0.22 |
| **LRW** | 50.05 ±0.24 | 51.25 ±0.30 | 49.71 ±0.73 | 51.22 ±0.22 |
| **SLR** | 56.61 ±0.18 | 56.70 ±0.21 | 56.45 ±0.11 | 55.35 ±1.94 |
| **CLR** | 57.36 ±0.11 | 57.01 ±0.90 | 57.43 ±0.70 | 57.49 ±0.83 |
| **LLR** | 57.50 ±0.27 | 57.14 ±0.60 | 57.22 ±0.71 | 57.65 ±0.02 |
| **ALLR** | 57.64 ±0.54 | 57.93 ±0.24 | 57.61 ±0.83 | 58.04 ±0.32 |
| | | Model sparsity 90% | | |
| Budget: | $2 \times 0.66\%$ | $2 \times 1.66\%$ | $3 \times 0.66\%$ | $3 \times 1.66\%$ |
| **FT** | 33.09 ±0.29 | 36.22 ±0.13 | 33.70 ±0.87 | 37.69 ±0.44 |
| **LRW** | 33.08 ±0.28 | 36.22 ±0.13 | 33.70 ±0.87 | 37.69 ±0.44 |
| **SLR** | 55.37 ±1.24 | 55.89 ±0.77 | 55.47 ±0.89 | 54.53 ±0.88 |
| **CLR** | 56.44 ±0.49 | 56.95 ±0.45 | 55.56 ±1.18 | 56.20 ±0.28 |
| **LLR** | 56.48 ±0.28 | 56.79 ±0.97 | 55.84 ±0.90 | 56.47 ±0.48 |
| **ALLR** | 56.23 ±0.17 | 57.40 ±0.27 | 56.35 ±0.76 | 56.62 ±0.16 |

Table 16: T5-small on WMT16 (de-en) (One Shot): Performance (measured as test BLEU score) of the different learning rate translation schemes for IMP in the One Shot setting for target sparsity of 50% - 70% and a retrain time of 2% (0.1 epochs), 5% (0.25 epochs), 10% (0.5 epochs), 20% (1 epochs), 40% (2 epochs), 60% (3 epochs) of the initial training budget. The **first**, **second**, and **third** best values for the translation schemes are highlighted. Results are averaged over two seeds with the standard deviation indicated.

| | **Model sparsity 50%** | | | | | |
|---|---|---|---|---|---|---|
| Budget: | 2% | 5% | 10% | 20% | 40% | 60% |
| **FT** | 23.76 ±0.05 | 23.93 ±0.20 | 24.24 ±0.15 | 24.08 ±0.07 | 24.06 ±0.15 | 24.36 ±0.09 |
| **LRW** | 23.76 ±0.05 | 23.93 ±0.20 | 24.24 ±0.15 | 24.08 ±0.07 | 24.68 ±0.19 | 24.92 ±0.12 |
| **SLR** | 21.66 ±0.40 | 22.09 ±0.13 | 22.74 ±0.11 | 23.36 ±0.34 | 24.09 ±0.36 | 24.61 ±0.03 |
| **CLR** | 22.69 ±0.06 | 23.29 ±0.17 | 23.79 ±0.34 | 24.47 ±0.31 | 24.73 ±0.06 | 25.28 ±0.19 |
| **LLR** | 22.97 ±0.01 | 23.20 ±0.15 | 24.00 ±0.14 | 24.53 ±0.22 | 25.11 ±0.05 | 24.95 ±0.43 |
| **ALLR** | 24.11 ±0.17 | 24.29 ±0.02 | 24.51 ±0.00 | 24.63 ±0.21 | 25.27 ±0.30 | 25.17 ±0.39 |
| | **Model sparsity 60%** | | | | | |
| Budget: | 2% | 5% | 10% | 20% | 40% | 60% |
| **FT** | 22.18 ±0.18 | 22.75 ±0.03 | 23.07 ±0.31 | 23.36 ±0.03 | 23.47 ±0.04 | 23.59 ±0.39 |
| **LRW** | 22.18 ±0.18 | 22.75 ±0.03 | 23.07 ±0.31 | 23.36 ±0.03 | 23.90 ±0.06 | 24.62 ±0.11 |
| **SLR** | 21.61 ±0.08 | 21.95 ±0.45 | 22.47 ±0.10 | 23.19 ±0.17 | 23.78 ±0.05 | 24.18 ±0.08 |
| **CLR** | 22.51 ±0.19 | 23.10 ±0.15 | 23.51 ±0.07 | 23.96 ±0.22 | 24.67 ±0.15 | 24.87 ±0.16 |
| **LLR** | 22.57 ±0.05 | 23.19 ±0.18 | 23.48 ±0.24 | 23.96 ±0.01 | 24.59 ±0.19 | 24.60 ±0.36 |
| **ALLR** | 23.52 ±0.07 | 24.04 ±0.01 | 24.32 ±0.09 | 24.43 ±0.20 | 24.91 ±0.11 | 24.90 ±0.25 |
| | **Model sparsity 70%** | | | | | |
| Budget: | 2% | 5% | 10% | 20% | 40% | 60% |
| **FT** | 18.20 ±0.14 | 20.01 ±0.49 | 20.75 ±0.24 | 21.25 ±0.21 | 21.84 ±0.17 | 22.05 ±0.17 |
| **LRW** | 18.20 ±0.14 | 20.01 ±0.49 | 20.75 ±0.24 | 21.25 ±0.21 | 22.98 ±0.24 | 24.22 ±0.13 |
| **SLR** | 20.71 ±0.03 | 21.41 ±0.53 | 22.27 ±0.23 | 22.51 ±0.09 | 23.36 ±0.19 | 23.66 ±0.01 |
| **CLR** | 21.68 ±0.09 | 22.68 ±0.22 | 23.19 ±0.08 | 23.52 ±0.00 | 23.98 ±0.16 | 23.99 ±0.09 |
| **LLR** | 21.83 ±0.21 | 22.44 ±0.32 | 23.18 ±0.28 | 23.64 ±0.16 | 23.64 ±0.11 | 24.12 ±0.16 |
| **ALLR** | 22.33 ±0.47 | 22.83 ±0.08 | 23.72 ±0.18 | 23.85 ±0.21 | 24.36 ±0.15 | 23.99 ±0.14 |

Table 17: ResNet-56 on CIFAR-10 (One Shot, Filter Pruning): Performance of the different learning rate translation schemes in the One Shot setting for target sparsity of 50 - 80%, and a retrain time of 5% (10 epochs), 10% (20 epochs), 15% (30 epochs) of the initial training budget. The **first**, **second**, and **third** best values for the translation schemes are highlighted. Results are averaged over two seeds with the standard deviation indicated.

| | Model sparsity 50% | | |
|---|---|---|---|
| Budget: | 5% | 10% | 20% |
| **FT** | 90.78 ±0.14 | 90.88 ±0.31 | 91.36 ±0.03 |
| **LRW** | 90.76 ±0.19 | 91.00 ±0.15 | 91.80 ±0.13 |
| **SLR** | 90.53 ±0.16 | 91.27 ±0.08 | 91.78 ±0.28 |
| **CLR** | 91.14 ±0.06 | 91.90 ±0.18 | 91.85 ±0.25 |
| **LLR** | 91.30 ±0.13 | 91.74 ±0.02 | 91.75 ±0.24 |
| **ALLR** | 91.35 ±0.18 | 91.84 ±0.19 | 92.15 ±0.50 |
| | **Model sparsity 60%** | | |
| Budget: | 5% | 10% | 20% |
| **FT** | 89.62 ±0.18 | 90.09 ±0.32 | 90.33 ±0.23 |
| **LRW** | 89.79 ±0.23 | 90.13 ±0.37 | 91.21 ±0.13 |
| **SLR** | 89.47 ±0.14 | 90.60 ±0.06 | 91.02 ±0.01 |
| **CLR** | 90.28 ±0.33 | 91.19 ±0.19 | 91.54 ±0.25 |
| **LLR** | 90.51 ±0.10 | 91.35 ±0.33 | 91.45 ±0.01 |
| **ALLR** | 90.60 ±0.28 | 91.33 ±0.42 | 91.61 ±0.34 |
| | **Model sparsity 70%** | | |
| Budget: | 5% | 10% | 20% |
| **FT** | 87.63 ±0.18 | 88.03 ±0.16 | 88.74 ±0.46 |
| **LRW** | 87.61 ±0.49 | 88.03 ±0.46 | 90.23 ±0.14 |
| **SLR** | 88.34 ±0.09 | 89.41 ±0.06 | 89.93 ±0.26 |
| **CLR** | 88.97 ±0.02 | 89.98 ±0.08 | 90.59 ±0.33 |
| **LLR** | 89.06 ±0.28 | 89.92 ±0.13 | 90.72 ±0.03 |
| **ALLR** | 89.38 ±0.04 | 90.03 ±0.09 | 90.82 ±0.04 |
| | **Model sparsity 80%** | | |
| Budget: | 5% | 10% | 20% |
| **FT** | 78.32 ±3.99 | 81.80 ±0.69 | 84.08 ±1.08 |
| **LRW** | 78.71 ±3.55 | 82.09 ±0.16 | 85.73 ±1.41 |
| **SLR** | 82.90 ±2.82 | 86.38 ±0.40 | 87.08 ±0.76 |
| **CLR** | 84.47 ±1.72 | 86.86 ±0.76 | 87.67 ±0.73 |
| **LLR** | 84.45 ±1.64 | 86.69 ±0.75 | 87.51 ±0.22 |
| **ALLR** | 84.66 ±1.70 | 86.88 ±0.77 | 87.68 ±0.13 |

Table 18: ResNet-50 on ImageNet (One Shot, Filter Pruning): Performance of the different learning rate translation schemes in the One Shot setting for target sparsity of 30 - 50%, and a retrain time of 2.22% (2 epochs), 5.55% (5 epochs) and 11.11% (10 epochs) of the initial training budget. The **first**, second, and third best values for the translation schemes are highlighted. Results are averaged over two seeds with the standard deviation indicated.

| | Model sparsity 30% | | |
|---|---|---|---|
| Budget: | 2.22% | 5.55% | 11.11% |
| FT | 67.28 ±0.27 | 69.87 ±0.21 | 71.23 ±0.22 |
| LRW | 67.28 ±0.27 | 69.87 ±0.21 | 71.23 ±0.22 |
| SLR | 66.21 ±0.04 | 69.17 ±0.00 | 70.81 ±0.00 |
| CLR | 68.45 ±0.01 | 70.88 ±0.00 | 72.12 ±0.02 |
| LLR | 68.73 ±0.02 | 70.93 ±0.09 | 72.08 ±0.08 |
| ALLR | 69.11 ±0.05 | 71.23 ±0.10 | 72.31 ±0.04 |
| | Model sparsity 40% | | |
| Budget: | 2.22% | 5.55% | 11.11% |
| FT | 60.87 ±0.04 | 65.58 ±0.17 | 68.11 ±0.04 |
| LRW | 60.87 ±0.04 | 65.58 ±0.17 | 68.11 ±0.04 |
| SLR | 64.45 ±0.11 | 67.87 ±0.09 | 69.66 ±0.26 |
| CLR | 66.91 ±0.02 | 69.68 ±0.10 | 71.25 ±0.14 |
| LLR | 67.02 ±0.13 | 69.83 ±0.04 | 71.04 ±0.08 |
| ALLR | 67.52 ±0.19 | 70.12 ±0.04 | 71.32 ±0.01 |
| | Model sparsity 50% | | |
| Budget: | 2.22% | 5.55% | 11.11% |
| FT | 48.85 ±0.38 | 58.28 ±0.16 | 62.93 ±0.01 |
| LRW | 48.85 ±0.38 | 58.28 ±0.16 | 62.93 ±0.01 |
| SLR | 61.81 ±0.00 | 66.08 ±0.16 | 68.32 ±0.12 |
| CLR | 64.50 ±0.17 | 67.93 ±0.21 | 69.82 ±0.03 |
| LLR | 64.87 ±0.20 | 68.12 ±0.12 | 69.67 ±0.06 |
| ALLR | 65.04 ±0.09 | 68.32 ±0.12 | 69.98 ±0.08 |

Table 19: ResNet-50 on ImageNet (Iterative, Filter Pruning): Performance of the different learning rate translation schemes in the iterative setting for target sparsity of 30%, 40%, 50% and retrain times as indicated in the Budget row. Here $2 \times 2.22\%$ indicates two prune-retrain cycles, each of which having length equal to 2.22% of the overall training budget. The **first**, **second**, and **third** best values for the translation schemes are highlighted. Results are averaged over two seeds with the standard deviation indicated.

| | | Model sparsity 30% | | |
|---|---|---|---|---|
| Budget: | $2 \times 2.22\%$ | $2 \times 5.55\%$ | $3 \times 2.22\%$ | $3 \times 5.55\%$ |
| **FT** | 68.18 ±0.16 | 70.47 ±0.12 | 68.77 ±0.04 | 70.78 ±0.12 |
| **LRW** | 68.18 ±0.16 | 70.47 ±0.12 | 68.77 ±0.04 | 70.78 ±0.12 |
| **SLR** | 67.09 ±0.07 | 69.54 ±0.01 | 67.43 ±0.10 | 69.51 ±0.06 |
| **CLR** | 69.14 ±0.11 | 71.07 ±0.10 | 69.27 ±0.01 | 71.16 ±0.04 |
| **LLR** | 69.31 ±0.12 | 71.16 ±0.08 | 69.49 ±0.07 | 71.30 ±0.11 |
| **ALLR** | 69.32 ±0.00 | 71.11 ±0.09 | 69.60 ±0.12 | 72.38 ±0.07 |

| | | Model sparsity 40% | | |
|---|---|---|---|---|
| Budget: | $2 \times 2.22\%$ | $2 \times 5.55\%$ | $3 \times 2.22\%$ | $3 \times 5.55\%$ |
| **FT** | 62.83 ±0.04 | 66.87 ±0.10 | 63.94 ±0.09 | 67.55 ±0.01 |
| **LRW** | 62.83 ±0.04 | 66.87 ±0.10 | 63.94 ±0.09 | 67.55 ±0.01 |
| **SLR** | 65.74 ±0.02 | 68.32 ±0.05 | 66.23 ±0.04 | 68.54 ±0.02 |
| **CLR** | 67.94 ±0.05 | 70.14 ±0.17 | 68.25 ±0.01 | 70.18 ±0.03 |
| **LLR** | 68.08 ±0.03 | 70.17 ±0.05 | 68.41 ±0.14 | 70.38 ±0.13 |
| **ALLR** | 68.11 ±0.04 | 70.17 ±0.06 | 68.95 ±0.02 | 71.90 ±0.02 |

| | | Model sparsity 50% | | |
|---|---|---|---|---|
| Budget: | $2 \times 2.22\%$ | $2 \times 5.55\%$ | $3 \times 2.22\%$ | $3 \times 5.55\%$ |
| **FT** | 54.30 ±0.11 | 61.16 ±0.10 | 56.51 ±0.27 | 62.44 ±0.21 |
| **LRW** | 54.30 ±0.11 | 61.16 ±0.10 | 56.51 ±0.27 | 62.44 ±0.21 |
| **SLR** | 63.85 ±0.08 | 67.04 ±0.13 | 64.59 ±0.07 | 67.28 ±0.17 |
| **CLR** | 66.07 ±0.01 | 68.78 ±0.20 | 66.71 ±0.19 | 69.00 ±0.16 |
| **LLR** | 66.40 ±0.05 | 68.75 ±0.06 | 66.87 ±0.14 | 69.11 ±0.10 |
| **ALLR** | 66.37 ±0.03 | 68.63 ±0.16 | 67.86 ±0.04 | 70.47 ±0.18 |

Table 20: MaxViT on ImageNet (One Shot, Filter Pruning): Performance of the different learning rate translation schemes in the One Shot setting for target sparsity of 30 - 50%, and a retrain time of 1% (2 epochs), 2.5% (5 epochs), 5% (10 epochs) and 10% (20 epochs) of the initial training budget. The first, second, and third best values for the translation schemes are highlighted. Results are averaged over two seeds with the standard deviation indicated.

| | Model sparsity 30% | | | |
|---|---|---|---|---|
| Budget: | 1% | 2.5% | 5% | 10% |
| FT | 68.05 ±0.08 | 70.96 ±0.31 | 72.32 ±0.20 | 73.99 ±0.11 |
| LRW | 68.05 ±0.08 | 70.96 ±0.31 | 72.32 ±0.20 | 73.99 ±0.11 |
| SLR | 74.23 ±0.16 | 74.80 ±0.26 | 75.80 ±0.12 | 76.76 ±0.01 |
| CLR | 75.45 ±0.10 | 76.11 ±0.37 | 76.85 ±0.20 | 77.50 ±0.10 |
| LLR | 75.64 ±0.21 | 76.19 ±0.06 | 76.88 ±0.10 | 77.55 ±0.20 |
| ALLR | 76.31 ±0.18 | 76.77 ±0.19 | 77.19 ±0.08 | 77.58 ±0.03 |
| | Model sparsity 35% | | | |
| Budget: | 1% | 2.5% | 5% | 10% |
| FT | 64.18 ±0.19 | 68.38 ±0.19 | 70.50 ±0.32 | 72.43 ±0.07 |
| LRW | 64.18 ±0.19 | 68.38 ±0.19 | 70.50 ±0.32 | 72.43 ±0.07 |
| SLR | 73.70 ±0.22 | 74.61 ±0.33 | 75.57 ±0.09 | 76.49 ±0.05 |
| CLR | 74.98 ±0.09 | 75.78 ±0.31 | 76.75 ±0.06 | 77.39 ±0.04 |
| LLR | 75.24 ±0.10 | 75.99 ±0.42 | 76.78 ±0.05 | 77.30 ±0.01 |
| ALLR | 75.86 ±0.14 | 76.48 ±0.22 | 77.01 ±0.09 | 77.42 ±0.14 |
| | Model sparsity 40% | | | |
| Budget: | 1% | 2.5% | 5% | 10% |
| FT | 58.65 ±0.52 | 64.75 ±0.50 | 67.78 ±0.18 | 70.56 ±0.36 |
| LRW | 58.65 ±0.52 | 64.75 ±0.50 | 67.78 ±0.18 | 70.56 ±0.36 |
| SLR | 73.20 ±0.16 | 74.17 ±0.67 | 75.31 ±0.04 | 76.24 ±0.02 |
| CLR | 74.52 ±0.19 | 75.40 ±0.39 | 76.27 ±0.08 | 77.14 ±0.04 |
| LLR | 74.70 ±0.22 | 75.62 ±0.27 | 76.38 ±0.05 | 77.22 ±0.08 |
| ALLR | 75.40 ±0.18 | 76.18 ±0.39 | 76.68 ±0.01 | 77.11 ±0.01 |
| | Model sparsity 45% | | | |
| Budget: | 1% | 2.5% | 5% | 10% |
| FT | 51.20 ±0.15 | 60.19 ±0.76 | 64.56 ±0.17 | 68.34 ±0.40 |
| LRW | 51.20 ±0.15 | 60.19 ±0.76 | 64.56 ±0.17 | 68.34 ±0.40 |
| SLR | 72.34 ±0.09 | 73.71 ±0.37 | 74.82 ±0.04 | 75.80 ±0.18 |
| CLR | 74.03 ±0.16 | 75.05 ±0.38 | 76.04 ±0.06 | 77.01 ±0.06 |
| LLR | 74.05 ±0.08 | 75.19 ±0.32 | 76.19 ±0.08 | 77.00 ±0.04 |
| ALLR | 74.75 ±0.08 | 75.66 ±0.31 | 76.31 ±0.00 | 77.05 ±0.06 |
| | Model sparsity 50% | | | |
| Budget: | 1% | 2.5% | 5% | 10% |
| FT | 39.85 ±0.37 | 53.59 ±1.31 | 60.00 ±0.06 | 65.20 ±0.45 |
| LRW | 39.85 ±0.37 | 53.59 ±1.31 | 60.00 ±0.06 | 65.20 ±0.45 |
| SLR | 71.59 ±0.13 | 73.05 ±0.45 | 74.26 ±0.07 | 75.62 ±0.11 |
| CLR | 73.16 ±0.14 | 74.50 ±0.44 | 75.59 ±0.07 | 76.69 ±0.07 |
| LLR | 73.41 ±0.36 | 74.62 ±0.64 | 75.85 ±0.09 | 76.69 ±0.06 |
| ALLR | 74.04 ±0.17 | 75.17 ±0.42 | 75.99 ±0.10 | 76.75 ±0.08 |

Table 21: MaxViT on ImageNet (Iterative, Filter Pruning): Performance of the different learning rate translation schemes in the iterative setting for target sparsity of 30% - 50% and retrain times as indicated in the Budget row. Here $2 \times 2.5\%$ indicates two prune-retrain cycles, each of which having length equal to 2.5% of the overall training budget. The **first**, **second**, and **third** best values for the translation schemes are highlighted. Results are averaged over two seeds with the standard deviation indicated.

| | | Model sparsity 30% | | |
|---|---|---|---|---|
| Budget: | $2 \times 1\%$ | $2 \times 2.5\%$ | $3 \times 1\%$ | $3 \times 2.5\%$ |
| FT | 68.56 ±0.62 | 71.84 ±0.31 | 70.01 ±0.98 | 72.15 ±0.78 |
| LRW | 68.56 ±0.62 | 71.84 ±0.31 | 70.01 ±0.98 | 72.15 ±0.78 |
| SLR | 73.48 ±0.37 | 75.03 ±0.30 | 74.28 ±0.48 | 75.19 ±0.43 |
| CLR | 74.73 ±0.19 | 76.30 ±0.22 | 75.51 ±0.52 | 76.62 ±0.15 |
| LLR | 75.01 ±0.11 | 76.56 ±0.42 | 75.79 ±0.33 | 76.72 ±0.14 |
| ALLR | 75.93 ±0.24 | 76.96 ±0.18 | 76.40 ±0.34 | 76.72 ±0.27 |

| | | Model sparsity 35% | | |
|---|---|---|---|---|
| Budget: | $2 \times 1\%$ | $2 \times 2.5\%$ | $3 \times 1\%$ | $3 \times 2.5\%$ |
| FT | 65.35 ±0.27 | 69.53 ±0.19 | 67.29 ±0.84 | 70.40 ±0.71 |
| LRW | 65.35 ±0.27 | 69.53 ±0.19 | 67.29 ±0.84 | 70.40 ±0.71 |
| SLR | 73.04 ±0.23 | 74.77 ±0.33 | 73.89 ±0.56 | 74.90 ±0.24 |
| CLR | 74.42 ±0.27 | 76.06 ±0.26 | 75.14 ±0.43 | 76.28 ±0.17 |
| LLR | 74.57 ±0.01 | 76.25 ±0.25 | 75.38 ±0.31 | 76.47 ±0.20 |
| ALLR | 75.62 ±0.22 | 76.61 ±0.28 | 76.17 ±0.35 | 76.59 ±0.13 |

| | | Model sparsity 40% | | |
|---|---|---|---|---|
| Budget: | $2 \times 1\%$ | $2 \times 2.5\%$ | $3 \times 1\%$ | $3 \times 2.5\%$ |
| FT | 60.97 ±0.11 | 66.62 ±0.23 | 63.67 ±0.76 | 67.77 ±0.74 |
| LRW | 60.97 ±0.11 | 66.62 ±0.23 | 63.67 ±0.76 | 67.77 ±0.74 |
| SLR | 72.48 ±0.09 | 74.51 ±0.20 | 73.50 ±0.53 | 74.66 ±0.31 |
| CLR | 73.89 ±0.28 | 75.70 ±0.23 | 74.94 ±0.55 | 76.17 ±0.31 |
| LLR | 74.29 ±0.18 | 75.91 ±0.28 | 75.19 ±0.54 | 76.26 ±0.17 |
| ALLR | 74.99 ±0.14 | 76.38 ±0.33 | 75.68 ±0.41 | 76.39 ±0.41 |

| | | Model sparsity 45% | | |
|---|---|---|---|---|
| Budget: | $2 \times 1\%$ | $2 \times 2.5\%$ | $3 \times 1\%$ | $3 \times 2.5\%$ |
| FT | 55.23 ±0.57 | 63.10 ±0.62 | 58.66 ±1.45 | 64.53 ±1.30 |
| LRW | 55.23 ±0.57 | 63.10 ±0.62 | 58.66 ±1.45 | 64.53 ±1.30 |
| SLR | 71.81 ±0.19 | 74.01 ±0.18 | 72.85 ±0.44 | 74.41 ±0.16 |
| CLR | 73.44 ±0.27 | 75.37 ±0.22 | 74.43 ±0.49 | 75.71 ±0.14 |
| LLR | 73.66 ±0.15 | 75.52 ±0.25 | 74.61 ±0.53 | 75.84 ±0.27 |
| ALLR | 74.40 ±0.20 | 75.90 ±0.29 | 75.07 ±0.32 | 76.01 ±0.40 |

| | | Model sparsity 50% | | |
|---|---|---|---|---|
| Budget: | $2 \times 1\%$ | $2 \times 2.5\%$ | $3 \times 1\%$ | $3 \times 2.5\%$ |
| FT | 48.40 ±1.29 | 58.29 ±0.95 | 53.10 ±1.34 | 60.36 ±1.41 |
| LRW | 48.40 ±1.29 | 58.29 ±0.95 | 53.10 ±1.34 | 60.36 ±1.41 |
| SLR | 71.08 ±0.27 | 73.30 ±0.30 | 72.30 ±0.58 | 73.88 ±0.19 |
| CLR | 72.74 ±0.02 | 74.74 ±0.24 | 73.83 ±0.66 | 75.61 ±0.42 |
| LLR | 73.05 ±0.06 | 75.01 ±0.35 | 74.21 ±0.46 | 75.55 ±0.14 |
| ALLR | 73.77 ±0.18 | 75.54 ±0.29 | 74.54 ±0.37 | 75.53 ±0.37 |

## B.2 BUDGETING THE RETRAINING PHASE

Similarly to Figure 3 in the main part, Figure 4 and Figure 5 display the envelope when retraining with LLR and ALLR, respectively. Here, the weight decay values, including those used for the baseline, were individually tuned for each datapoint using a grid search over 1e-4, 2e-4 and 5e-4. All results are averaged over two seeds with max-min-bands indicated.

Although LLR and ALLR show slightly different behaviour, we note that both reach the performance of the baseline with significantly less retraining epochs than required to establish that baseline.

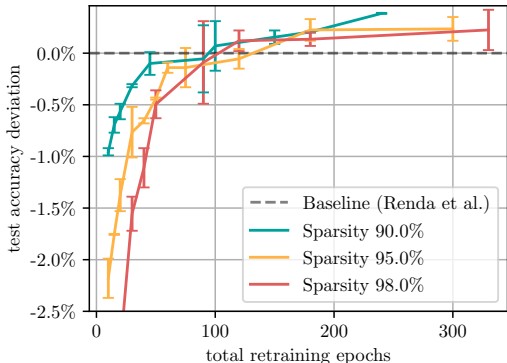

Figure 4: ResNet-56 on CIFAR-10: Envelope of the performance of IMP using LLR compared to the baseline of Renda et al. (2020) shown over the total number of epochs used for retraining. Results are averaged over two seeds with max-min-bands indicated.

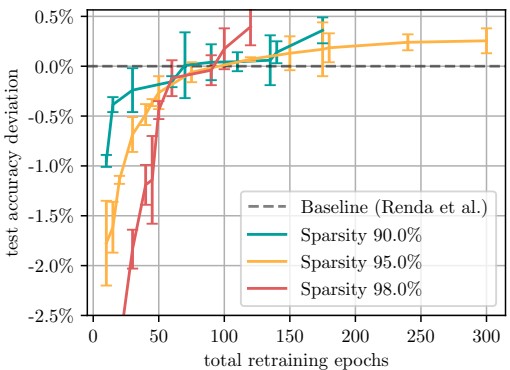

Figure 5: ResNet-56 on CIFAR-10: Envelope of the performance of IMP using ALLR compared to the baseline of Renda et al. (2020) shown over the total number of epochs used for retraining. Results are averaged over two seeds with max-min-bands indicated.

### B.3   BUDGETING THE INITIAL TRAINING

Figure 6 and Figure 7 show the behaviour of One Shot as well as iterative pruning when budgeting the initial training, where retraining is performed with LLR and ALLR, respectively. Here, the initial training is budgeted between 5% up to 100% of 200 epochs, which we consider the 'full' training. The initial training follows a linearly decaying learning rate schedule which starts from 0.1. After initial training, IMP is applied One Shot (retraining for 30 epochs) or iteratively (3 cycles of 15 epochs each). The individual datapoints are given by the length of the initial training, namely 10, 25, 50, 75, 100, 125, 150, 175 and 200 training epochs. We keep weight decay fixed at 5e-4.

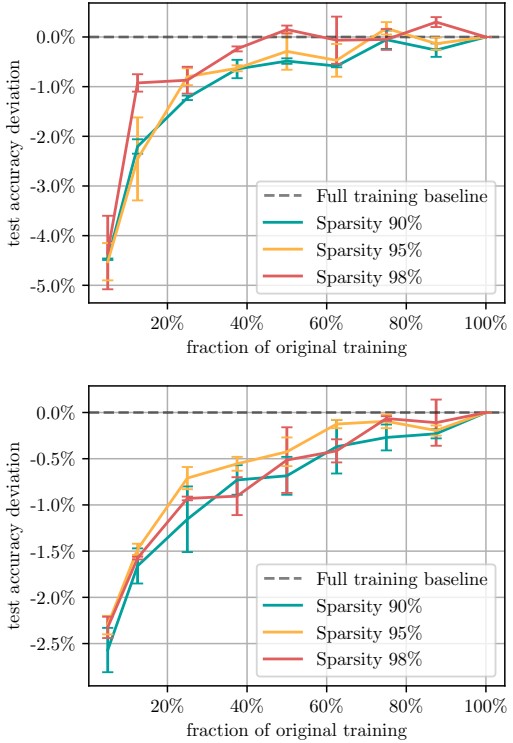

Figure 6: ResNet-56 on CIFAR-10: Test accuracy achieved by IMP when retraining One Shot for 30 epochs (above) or iteratively for three cycles of 15 epochs each (below) with LLR after budgeting the initial training length. Each line depicts a different goal sparsity and values are indicated as deviation from the performance of IMP when applied to a network trained with the full budget. Results are averaged over two seeds with max-min-bands indicated.

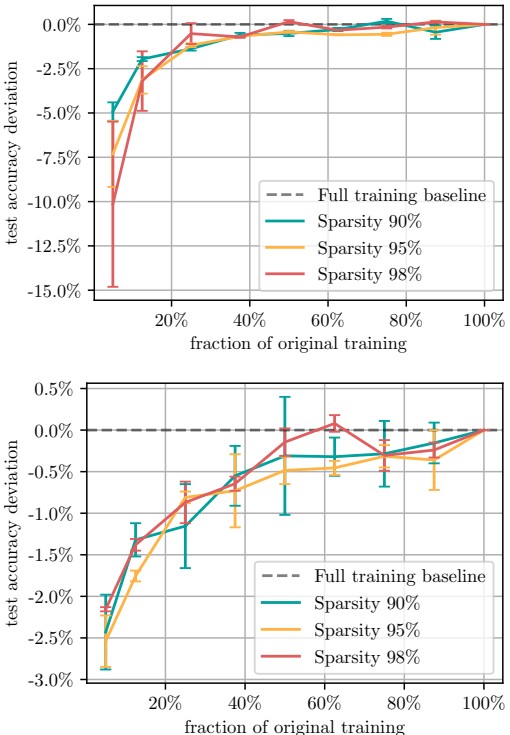

Figure 7: ResNet-56 on CIFAR-10: Test accuracy achieved by IMP when retraining One Shot for 30 epochs (above) or iteratively for three cycles of 15 epochs each (below) with ALLR after budgeting the initial training length. Each line depicts a different goal sparsity and values are indicated as deviation from the performance of IMP when applied to a network trained with the full budget. Results are averaged over two seeds with max-min-bands indicated.

## B.4 COMPARISON TO STATE-OF-THE-ART PRUNING-STABLE METHODS

Table 22 lists all methods that take part in the comparative study between BIMP and state-of-the-art pruning-stable methods (cf. Section 3.3), where we added IMP as a method of reference. In the following subsections, we list the exact hyperparameter grids used for each pruning-stable method.

### B.4.1 RESNET-56 ON CIFAR-10

For each method (including BIMP) we tune weight decay over 1e-4, 5e-4, 1e-3 and 5e-3 and keep momentum fixed at 0.9. Since the learning rate schedule might need additional tuning, we vary the initial learning rate between 0.05, 0.15, 0.1 and 0.2 for all methods except CS. The decay of the schedule follows the same pattern as listed in Table 3. Since CS required the broadest grid, we fixed the learning rate schedule to the one in Table 3. Otherwise, we used the following grids.

**BIMP**
Initial training budget epochs: $\{20, 60, 100\}$.
Number of pruning phases of equal length: $\{1, 2, 3\}$.

**GMP**
Equally distributed pruning steps: $\{20, 100\}$.

**GSM**
Momentum: $\{0.9, 0.95, 0.99\}$.

Table 22: Overview of sparsification methods. CS, STR and DST control the sparsity implicitly via additional hyperparameters. IMP is the only method that is pruning instable by design, i.e., it loses its performance right after the ultimate pruning. Further, IMP is the only method that is sparsity agnostic throughout the regular training; the sparsity does not play a role while training to convergence. All other methods require training an entire model when changing the goal sparsity.

| | Sparsity specifiable | Pruning stable | Sparsity agnostic training |
|---|---|---|---|
| IMP | ✓ | ✗ | ✓ |
| BIMP | ✓ | ✓ | ✗ |
| GMP | ✓ | ✓ | ✗ |
| GSM | ✓ | ✓ | ✗ |
| LC | ✓ | ✓ | ✗ |
| DPF | ✓ | ✓ | ✗ |
| DNW | ✓ | ✓ | ✗ |
| CS | ✗ | ✓ | ✗ |
| STR | ✗ | ✓ | ✗ |
| DST | ✗ | ✓ | ✗ |

**LC**
We only tune the weight decay.

**DPF**
As for GMP, we tune the number of pruning steps, i.e., $\{20, 100\}$, and the weight decay.

**DNW**
We only tune the weight decay, since there are no additional hyperparameters.

**CS**
As recommended by Savarese et al. (2020), we fix the temperature $\beta$ at 300. We tune the mask initialization $s_0$ over $\{-0.3, -0.25, -0.2, -0.1, -0.05, 0, 0.05, 0.1, 0.2, 0.25, 0.3\}$ and the $\ell_1$ penalty $\lambda$ over $\{$1e-8, 1e-7$\}$.

**STR**
We tune the initial threshold value $s_{\text{init}} \in \{-100, -50, -5, -2, -1, 0, 5, 50, 100\}$. In an extended grid search, we also used weight decays in $\{$5e-05, 1e-4$\}$ and varied $s_{\text{init}} \in \{-40, -30, -20, -10\}$.

**DST**
We tune the sparsity-controlling regularization parameter $\alpha \in \{$5e-6, 1e-5, 5e-5, 1e-4, 5e-4$\}$. In an extended grid search, we used weight decays in $\{$0, 1e-4$\}$ and tuned $\alpha$ over $\{$1e-7, 5e-7, 1e-6$\}$.

### B.4.2  WIDERESNET ON CIFAR-100

For each method (including BIMP) we tune weight decay over 1e-4, 2e-4 and 5e-4 and keep momentum fixed at 0.9. We vary the initial learning rate between 0.05, 0.1 and 0.15 for all methods and we use the learning rate schedule of Table 3 but additionally include a linear schedule with initial learning rate value set to 0.1. Otherwise, we used the following grids.

**BIMP**
Initial training budget epochs: $\{20, 60, 100\}$.
Number of pruning phases of equal length: $\{1, 2, 3, 4\}$.

**GMP**
Equally distributed pruning steps: $\{20, 100\}$.

**GSM**
Momentum: $\{0.9, 0.95, 0.99\}$.

**LC**
We only tune the weight decay.

**DPF**
Equally distributed pruning steps: $\{20, 100\}$.

**DNW**
We only tune the weight decay, since there are no additional hyperparameters.

**CS**
As recommended by Savarese et al. (2020), we fixed the temperature $\beta$ at 300, but increased it to 500 upon noticing the pruning instability of CS on this dataset. We tune the mask initialization $s_0$ over $\{-0.3, -0.25, -0.2, -0.1, -0.05, -0.03, -0.01, -0.005, -0.003, -0.001, 0\}$ and the $\ell_1$ penalty $\lambda$ over $\{1e\text{-}9, 1e\text{-}8, 1e\text{-}7\}$.

**STR**
We tune the initial threshold value $s_{\text{init}} \in \{-5000, -3000, -2000, -1000, -500\}$. In an extended grid search, we also varied $s_{\text{init}} \in \{-200, -150, -100, -80, -50, -40, -25, -10, 0\}$.

**DST**
We tuned $\alpha$ over $\{5e\text{-}6, 1e\text{-}5, 5e\text{-}5, 1e\text{-}4, 5e\text{-}4\}$. In an extended grid search, we tuned $\alpha$ over $\{1e\text{-}6, 3e\text{-}6, 8e\text{-}6, 3e\text{-}5, 8e\text{-}5, 2e\text{-}4, 3e\text{-}4, 4e\text{-}4, 5e\text{-}4, 6e\text{-}4, 7e\text{-}4, 8e\text{-}4, 9e\text{-}4, 1e\text{-}3\}$.

### B.4.3 RESNET-50 ON IMAGENET

If not otherwise specified, we fix the weight decay at 1e-4 and momentum at 0.9. For all methods, we use the learning rate schedule of Table 3 but additionally include a linear schedule with the same initial learning rate value to rule out too strong dependence on the training schedule. Note that for GSM and LC, we choose the best value training from scratch or using a pretrained model. In the latter case, we apply the methods for 10, 20 or 40 epochs. Otherwise, we used the following grids.

**BIMP**
Initial training budget epochs: $\{60, 75\}$.
Number of pruning phases of equal length: $\{1, 2, 3\}$.

**GMP**
Equally distributed pruning steps: $\{5, 9, 18, 45\}$.

**GSM**
Momentum: $\{0.9, 0.95\}$.
Weight decay: $\{1e\text{-}4, 1e\text{-}5\}$.

**LC**
Weight decay: $\{1e\text{-}4, 1e\text{-}5\}$.

**DPF**
Equally distributed pruning steps: $\{9, 18\}$.

**DNW**
Weight decay: $\{1e\text{-}4, 1e\text{-}5\}$.

**STR**
Weight decay: $\{1e\text{-}5, 2e\text{-}5, 3e\text{-}5, 4e\text{-}5, 1e\text{-}4\}$.
$s_{\text{init}}$: $\{-20, -15, -13, -10, -5, -4, -2\}$.

**DST**
Weight decay: $\{1e\text{-}4, 1e\text{-}5\}$.
$\alpha$: $\{1e\text{-}7, 5e\text{-}7, 1e\text{-}6, 2e\text{-}6, 5e\text{-}6, 8e\text{-}6, 1e\text{-}5, 1e\text{-}4\}$

### B.4.4 RESULTS

Table 23 and Table 24 show the full results when comparing BIMP to pruning-stable methods in the case of ResNet-56 on CIFAR-10 and WideResNet on CIFAR-100, respectively.

Table 23: ResNet-56 on CIFAR-10: Results of the comparison between BIMP and pruning stable methods for the sparsity range between 90% and 99.5%. The columns are structured as follows: First the method is stated. Secondly, we denote the images-per-second throughput through training, i.e., a higher number indicates a faster algorithm. The following columns are substructured as follows: Each column corresponds to one goal sparsity and each subcolumn denotes the Top-1 accuracy, the theoretical speedup and the actual sparsity reached. All results include standard deviations. Missing values (indicated by —) correspond to cases where we were unable to obtain results in the desired sparsity range, i.e., there did not exist a training configuration with average final sparsity within a .25% interval around the goal sparsity and the closest one is too far away or belongs to another column.

| Method | # img/s | Model sparsity 90% | | | Model sparsity 93% | | |
| | | Accuracy | Speedup | Sparsity | Accuracy | Speedup | Sparsity |
|---|---|---|---|---|---|---|---|
| **BIMP** | 3638 | 93.35 ±0.13 | 7 ±0.2 | 90.00 ±0.00 | 93.00 ±0.26 | 10 ±0.8 | 93.00 ±0.00 |
| **GMP** | 3536 | 92.87 ±0.13 | 9 ±0.0 | 90.00 ±0.00 | 92.47 ±0.33 | 13 ±0.0 | 93.00 ±0.00 |
| **GSM** | 3251 | 91.27 ±0.69 | 11 ±2.5 | 90.00 ±0.00 | 91.12 ±1.07 | 15 ±3.5 | 93.00 ±0.00 |
| **DPF** | 3560 | 93.32 ±0.11 | 7 ±0.0 | 90.00 ±0.00 | 93.23 ±0.05 | 9 ±0.5 | 93.00 ±0.00 |
| **DNW** | 3335 | 91.81 ±1.83 | 6 ±0.8 | 90.00 ±0.00 | 91.97 ±0.20 | 5 ±0.2 | 93.09 ±0.00 |
| **LC** | 3467 | 90.51 ±0.16 | 5 ±0.1 | 90.00 ±0.00 | 89.67 ±0.55 | 7 ±0.6 | 93.00 ±0.00 |
| **STR** | 2864 | 89.25 ±1.23 | 8 ±0.8 | 90.15 ±0.76 | 90.27 ±0.86 | 20 ±4.0 | 93.01 ±0.58 |
| **CS** | 2725 | 91.87 ±0.30 | 13 ±0.3 | 90.52 ±0.76 | 89.64 ±1.43 | 16 ±2.9 | 92.78 ±0.18 |
| **DST** | 1972 | 92.41 ±0.28 | 10 ±0.7 | 89.55 ±0.41 | — | — | — |

| Method | # img/s | Model sparsity 95% | | | Model sparsity 98% | | |
| | | Accuracy | Speedup | Sparsity | Accuracy | Speedup | Sparsity |
|---|---|---|---|---|---|---|---|
| **BIMP** | 3638 | 92.57 ±0.32 | 12 ±0.8 | 95.00 ±0.00 | 90.43 ±0.11 | 26 ±4.9 | 98.00 ±0.00 |
| **GMP** | 3536 | 92.20 ±0.07 | 18 ±0.0 | 95.00 ±0.00 | 90.08 ±0.32 | 43 ±0.0 | 98.00 ±0.00 |
| **GSM** | 3251 | 90.07 ±1.67 | 21 ±5.0 | 95.00 ±0.00 | 87.21 ±1.24 | 44 ±10.5 | 98.00 ±0.00 |
| **DPF** | 3560 | 92.68 ±0.14 | 12 ±0.1 | 95.00 ±0.00 | 90.49 ±0.23 | 29 ± 1.2 | 98.00 ±0.00 |
| **DNW** | 3335 | 91.95 ±0.06 | 7 ±0.3 | 95.09 ±0.00 | 34.87 ±43.08 | 26 ±2.8 | 98.10 ±0.00 |
| **LC** | 3467 | 89.16 ±0.60 | 8 ±0.5 | 95.00 ±0.00 | 85.11 ±0.51 | 16 ±0.0 | 98.00 ±0.00 |
| **STR** | 2864 | 89.77 ±1.75 | 31 ±10.3 | 95.11 ±0.28 | 89.15 ±0.26 | 66 ±4.9 | 98.00 ±0.04 |
| **CS** | 2725 | 91.36 ±0.23 | 21 ±2.9 | 95.38 ±0.19 | 90.04 ±0.36 | 50 ±7.2 | 98.12 ±0.06 |
| **DST** | 1972 | 89.17 ±0.00 | 18 ±0.0 | 94.42 ±0.00 | 88.22 ±0.36 | 53 ± 3.6 | 98.04 ±0.21 |

| Method | # img/s | Model sparsity 99% | | | Model sparsity 99.5% | | |
| | | Accuracy | Speedup | Sparsity | Accuracy | Speedup | Sparsity |
|---|---|---|---|---|---|---|---|
| **BIMP** | 3638 | 87.17 ±0.59 | 56 ±7.2 | 99.00 ±0.00 | 82.02 ±0.07 | 91 ±12.8 | 99.50 ±0.00 |
| **GMP** | 3536 | 86.72 ±0.04 | 77 ±0.0 | 99.00 ±0.00 | 80.26 ±0.80 | 127 ±0.0 | 99.50 ±0.00 |
| **GSM** | 3251 | 83.00 ±0.90 | 77 ±18.9 | 99.00 ±0.00 | 76.52 ±1.82 | 156 ±41.4 | 99.50 ±0.00 |
| **DPF** | 3560 | 86.76 ±0.33 | 63 ±3.7 | 99.00 ±0.00 | 80.03 ±0.64 | 146 ±34.3 | 99.50 ±0.00 |
| **DNW** | 3335 | 83.67 ±0.24 | 15 ±0.1 | 99.17 ±0.00 | 34.71 ±24.17 | 34 ±4.4 | 99.67 ±0.00 |
| **LC** | 3467 | 81.63 ±0.74 | 30 ±1.5 | 99.00 ±0.00 | 74.44 ±2.03 | 64 ±4.5 | 99.50 ±0.00 |
| **STR** | 2864 | 83.68 ±0.94 | 159 ±31.9 | 99.13 ±0.02 | 77.34 ±2.68 | 420 ±167.4 | 99.66 ±0.09 |
| **CS** | 2725 | 86.55 ±0.92 | 69 ±7.1 | 98.90 ±0.02 | — | — | — |
| **DST** | 1972 | 86.99 ±0.00 | 63 ±0.0 | 98.36 ±0.00 | — | — | — |

Table 24: WideResNet on CIFAR-100: Results of the comparison between BIMP and pruning stable methods for the sparsity range between 90% and 98%. The columns are structured as follows: First the method is stated. Secondly, we denote the images-per-second throughput through training, i.e., a higher number indicates a faster algorithm. The following columns are substructured as follows: Each column corresponds to one goal sparsity and each subcolumn denotes the Top-1 accuracy, the theoretical speedup and the actual sparsity reached. All results include standard deviations.

| Method | # img/s | Model sparsity 90% | | | Model sparsity 93% | | |
|---|---|---|---|---|---|---|---|
| | | Accuracy | Speedup | Sparsity | Accuracy | Speedup | Sparsity |
| **BIMP** | 2318 | **76.10 ±0.00** | 8 ±0.1 | 90.00 ±0.00 | **75.92 ±0.42** | 12 ±0.2 | 93.00 ±0.00 |
| **GMP** | 2278 | **76.09 ±0.29** | 8 ±0.1 | 90.00 ±0.00 | **74.91 ±0.16** | 9 ±0.2 | 93.00 ±0.00 |
| **GSM** | 2190 | 74.25 ±0.31 | 7 ±0.0 | 90.00 ±0.00 | 73.88 ±1.19 | 8 ±0.0 | 93.00 ±0.00 |
| **DPF** | 2299 | 75.44 ±0.44 | 7 ±0.8 | 90.00 ±0.00 | 74.78 ±0.52 | 11 ±1.3 | 93.00 ±0.00 |
| **DNW** | 258 | **76.84 ±0.72** | 7 ±0.1 | 90.00 ±0.00 | **75.75 ±0.10** | 8 ±0.1 | 93.00 ±0.00 |
| **LC** | 2207 | 72.82 ±0.57 | 5 ±0.1 | 90.00 ±0.00 | 68.87 ±0.13 | 6 ±0.1 | 93.00 ±0.00 |
| **STR** | 2148 | 73.06 ±1.22 | 15 ±0.0 | 90.97 ±0.00 | 74.11 ±0.21 | 13 ±0.2 | 92.36 ±0.17 |
| **CS** | 2077 | 73.50 ±0.47 | 7 ±0.0 | 90.82 ±0.09 | 73.52 ±0.21 | 10 ±0.0 | 92.96 ±0.03 |
| **DST** | 1868 | 72.84 ±0.42 | 12 ±0.1 | 90.00 ±0.00 | 72.16 ±0.32 | 20 ±0.3 | 93.00 ±0.00 |

| Method | # img/s | Model sparsity 95% | | | Model sparsity 98% | | |
|---|---|---|---|---|---|---|---|
| | | Accuracy | Speedup | Sparsity | Accuracy | Speedup | Sparsity |
| **BIMP** | 2318 | **75.64 ±0.29** | 16 ±0.2 | 95.00 ±0.00 | **73.73 ±0.20** | 39 ±0.3 | 98.00 ±0.00 |
| **GMP** | 2278 | **74.80 ±0.60** | 20 ±0.0 | 95.00 ±0.00 | 72.07 ±0.03 | 24 ±0.1 | 98.00 ±0.00 |
| **GSM** | 2190 | 73.25 ±0.52 | 10 ±0.1 | 95.00 ±0.00 | 68.79 ±0.58 | 18 ±0.0 | 98.00 ±0.00 |
| **DPF** | 2299 | 73.87 ±0.69 | 17 ±0.6 | 95.00 ±0.00 | **72.18 ±0.29** | 42 ±0.9 | 98.00 ±0.00 |
| **DNW** | 258 | **74.94 ±0.25** | 9 ±0.2 | 95.00 ±0.00 | **72.13 ±0.12** | 17 ±0.5 | 98.00 ±0.00 |
| **LC** | 2207 | 60.29 ±0.75 | 8 ±0.2 | 95.00 ±0.00 | 29.30 ±0.20 | 17 ±0.5 | 98.00 ±0.00 |
| **STR** | 2148 | 70.66 ±0.52 | 24 ±0.0 | 94.40 ±0.02 | 65.17 ±0.64 | 22 ±0.1 | 98.50 ±0.03 |
| **CS** | 2077 | 72.81 ±0.13 | 12 ±0.0 | 95.22 ±0.00 | 72.29 ±0.34 | 28 ±0.5 | 97.99 ±0.00 |
| **DST** | 1868 | 70.66 ±0.33 | 24 ±0.5 | 95.00 ±0.00 | 68.46 ±0.00 | 40 ±0.0 | 97.96 ±0.00 |

## C   ABLATION STUDIES

### C.1   ALLR: THE INITIAL VALUE OF THE LEARNING RATE SCHEDULE

We analyze the impact of our choices regarding the design of ALLR. First of all, we justify the usage of the proxy to determine the initial learning rate. Recall that ALLR discounts the initial value $\eta_1$ by a factor $d \in [0, 1]$ to account for both the available retraining time (similar to LRW) and the actual increase in loss induced by pruning. It does so by choosing $d = \max(d_1, d_2)$, where

$$d_1 = \frac{\|\mathcal{W} - \mathcal{W}^p\|_2}{\|\mathcal{W}\|_2 \cdot \sqrt{s}} \in [0, 1] \tag{2}$$

measures the relative $L_2$-norm change in the weights due to pruning and $d_2 = T_{rt}/T$ accounts for the length of the retrain phase in comparison to the original training length.

The motivation behind choosing these factors lies in handling different retraining scenarios. When choosing the initial value, two aspect have to be taken into account.

1. **The number of retraining epochs available, $T_{rt}$, might be very limited.** A large initial learning rate might yield too large oscillations of the loss, from which we cannot recover in a too short retraining timeframe. To find highly generalizing minima, we need both phases: a large-step and a small-step retraining regime (Jastrzębski et al., 2017; Li et al., 2019; You et al., 2019; Leclerc & Madry, 2020). If $T_{rt}$ is too small, we do not have enough time to do both. This is the advantage of LRW: The magnitude of the initial learning rates depends directly on $T_{rt}$.

2. **The pruning-induced decrease in accuracy might be very small, depending on the fraction of weights we remove.** Pruning only a small fraction of the weights will most likely have little impact on the loss: in a highly over-parameterized network, removing a small fraction of the weights will not drive the parameters far away from the current (local) optimum. In that case, convergence is accelerated by performing small learning rate steps. We expect that especially in the higher sparsity regime, where the loss increases dramatically by pruning, a larger initial learning rate is desirable to being able to compensate the drop in accuracy and approach the optimum faster.

In other words, we seek to choose the learning rate to initially be as large as possible (but bounded by the largest learning rate used throughout training), where as large as possible means taking two factors into account: How much *increase in loss* do we have to compensate and do we have enough *time* to properly perform both a large-step regime and a small-step regime?

The metric $d_2$ is an immediate proxy to the duration of retraining phase, motivated by LRW. Note that we clip $d_2$ to not become larger than 1. When we are allowed to retrain at least as long as the initial training, we prefer the initial value of the original training, which is the maximum value in the problem setting. In any other case, we take a fraction of it. However, regarding the metric $d_1$, we choose to measure the drop in $L_2$-norm induced by pruning instead of measuring the actual drop in accuracy, since the latter is only available after performing an entire forward pass on the train dataset.

To investigate whether this replacement is justified, we compare ALLR to an accuracy drop based variant of it, namely *AccALLR*. AccALLR works exactly like ALLR, but we instead choose $d_1$ as follows:

$$d_1 = \frac{\mathrm{Acc}(\Phi_d) - \mathrm{Acc}(\Phi_s)}{\mathrm{Acc}(\Phi_d)} \in [0, 1], \tag{3}$$

where $\mathrm{Acc}(\Phi)$ denotes the train accuracy of model $\Phi$ and $\Phi_d$, $\Phi_s$ denote the dense and sparse model, respectively. We clip $d_1$ between 0 and 1. The metric $d_1$ measures the relative drop in accuracy compared to the dense model. It hence requires a complete forward pass of the sparse model on the training data before commencing retraining. For CIFAR-10, Table 25 shows that despite AccALLR having a slight advantage over the 'less-informed' ALLR, these often lie within the margin of the standard deviation. The discounting factor of ALLR taking the norm-drop into account is indeed a well-functioning proxy for the pruning-induced drop in accuracy. Similarly, Table 26 shows the comparison on ImageNet in the iterative setting. Especially for moderate sparsities such as 70% we

see that the accuracy is a more precise metric when determining the initial learning rate, however it becomes unprecise at higher sparsities. High distortions of the parameters will lead to the accuracy dropping entirely to that of a random classifier, resulting in the largest possible learning rate being taken as the initial value. ALLR gives a more robust estimate.

Table 25: ResNet-56 on CIFAR-10 (One Shot): Performance of ALLR versus its accuracy-drop based variant AccALLR for IMP in the One Shot setting for target sparsity of 80%, 90%, 98% and a retrain time of 2% (2 epochs), 5% (5 epochs), 20% (20 epochs) of the initial training budget. The **first** and **second** best values for the translation schemes are highlighted. Results are averaged over two seeds with the standard deviation indicated.

| | Model sparsity 80% | | |
|---|---|---|---|
| Budget: | 2% | 5% | 20% |
| ALLR | 90.85 ±0.66 | 91.40 ±1.02 | 91.96 ±0.70 |
| AccALLR | 90.92 ±1.17 | 91.27 ±1.24 | 91.71 ±1.02 |
| | Model sparsity 90% | | |
| Budget: | 2% | 5% | 20% |
| ALLR | 89.56 ±0.55 | 90.16 ±1.24 | 90.87 ±1.32 |
| AccALLR | 89.75 ±1.36 | 90.34 ±1.23 | 90.89 ±1.21 |
| | Model sparsity 98% | | |
| Budget: | 2% | 5% | 20% |
| ALLR | 80.47 ±1.09 | 82.60 ±1.48 | 84.75 ±1.24 |
| AccALLR | 80.11 ±1.23 | 82.48 ±1.72 | 84.89 ±1.08 |

Table 26: ResNet-50 on ImageNet (Iterative): Performance of ALLR versus its accuracy-drop based variant AccALLR for IMP in the iterative setting for target sparsity of 70% and 90%. The **first** and **second** best values for the translation schemes are highlighted. Results are averaged over two seeds with the standard deviation indicated.

| | Model sparsity 70% | | | |
|---|---|---|---|---|
| Budget: | $2 \times 2.22\%$ | $2 \times 3.33\%$ | $3 \times 2.22\%$ | $3 \times 3.33\%$ |
| ALLR | 74.19 ±0.03 | 74.60 ±0.11 | 73.80 ±0.01 | 74.71 ±0.06 |
| AccALLR | 74.64 ±0.21 | 75.03 ±0.21 | 74.94 ±0.09 | 75.23 ±0.24 |
| | Model sparsity 90% | | | |
| Budget: | $2 \times 2.22\%$ | $2 \times 3.33\%$ | $3 \times 2.22\%$ | $3 \times 3.33\%$ |
| ALLR | 70.08 ±0.03 | 71.77 ±0.18 | 69.96 ±0.21 | 72.36 ±0.19 |
| AccALLR | 69.19 ±0.18 | 70.51 ±0.11 | 70.54 ±0.15 | 72.28 ±0.09 |

We further show the impact of the two factors $d_1$ and $d_2$ of ALLR by considering all four cases of selectively disabling a subset of the metrics, that is, we compare ALLR to *ALLRd1* (ALLR only using $d_1$), *ALLRd2* (ALLR only using $d_2$) and *LLR* (which is the same as applying no discounting factor at all) when training ResNet-56 on CIFAR-10. Table 27 displays the performance of the four different variations when testing against the different scenarios as outlined above, i.e., with low to high sparsity (80-98%) and short to long retraining time (2-20% budget), where we stick to the One Shot setting. First of all, we observe ALLR consistently performs best or second best among all variants. Note that the discounting factor $d = \max(d_1, d_2)$ of ALLR is attained at either $d_1$ or $d_2$ and ALLR always matches the performance of the better $d_1$- or $d_2$-disabling variant. Further, we observe that ALLR behaves exactly as designed and addresses the different retraining scenarios as outlined above. In the low sparsity, short retraining regime, both ALLR variants disabling one of the two discounting

factor perform equally well and are superior than selecting a large initial learning rate as LLR does. When the retraining time is increased, larger initial learning rates are more suited despite low sparsity, as visible in the marginalized difference to LLR. With further decreasing sparsity and increasing retraining time, the effect of ALLRd1 would vanish to the benefit of ALLRd2. On the other hand, in the high sparsity regime, we notice that it is beneficial to start with a high initial learning rate and it is not sufficient to only include the length of the retraining time, as ALLRd2 shows with a difference of ten percent in test accuracy to its competitors. LRW would behave similarly in this particular case. Overall, ALLR addresses these issues by accounting both for the increase in loss as well as the available retraining time.

Table 27: ResNet-56 on CIFAR-10 (One Shot): Performance of the ALLR derivations for IMP in the One Shot setting for target sparsity of 90%, 98% and a retrain time of 2% (2 epochs), 5% (5 epochs), 20% (20 epochs) of the initial training budget. The first, second, and third best values for the translation schemes are highlighted. Results are averaged over two seeds with the standard deviation indicated.

| | Model sparsity 80% | | |
|---|---|---|---|
| Budget: | 2% | 5% | 20% |
| ALLR | 90.92 ±0.81 | 91.40 ±1.07 | 91.82 ±0.87 |
| ALLRd1 | 90.73 ±0.71 | 91.23 ±1.19 | 91.94 ±0.85 |
| ALLRd2 | 90.78 ±1.16 | 91.28 ±1.19 | 91.66 ±0.70 |
| LLR | 90.07 ±0.72 | 90.84 ±1.29 | 91.76 ±0.95 |
| | Model sparsity 90% | | |
| Budget: | 2% | 5% | 20% |
| ALLR | 89.75 ±0.89 | 90.20 ±1.08 | 91.03 ±1.15 |
| ALLRd1 | 89.66 ±1.18 | 90.10 ±1.03 | 90.72 ±1.56 |
| ALLRd2 | 87.94 ±1.07 | 89.70 ±1.29 | 90.85 ±1.11 |
| LLR | 88.68 ±1.12 | 89.89 ±1.29 | 90.72 ±1.17 |
| | Model sparsity 98% | | |
| Budget: | 2% | 5% | 20% |
| ALLR | 80.11 ±1.28 | 82.59 ±1.60 | 84.89 ±1.26 |
| ALLRd1 | 80.16 ±1.19 | 82.51 ±1.75 | 84.84 ±1.04 |
| ALLRd2 | 70.05 ±0.31 | 79.04 ±0.62 | 85.12 ±1.08 |
| LLR | 79.75 ±1.18 | 82.43 ±1.60 | 84.55 ±1.48 |

## C.2 PRUNING SELECTION CRITERIA

We compare the original GLOBAL pruning criterion of IMP to the previously introduced proposed alternatives. In the case of ResNet-56 on CIFAR-10 (Figure 8) and VGG-16 on CIFAR-10 (Figure 11), we report the weight decay config with highest accuracy, where we optimized over the values 1e-4, 5e-4 and 1e-3. For WideResNet on CIFAR-100 (Figure 9) and ResNet-50 on ImageNet (Figure 10) we relied on a weight decay value of 1e-4 for both architectures. The CIFAR-10 and CIFAR-100 results are averaged over three seeds and max-min-bands are indicated. For ImageNet, the results are based on a single seed.

We tested both FT (Han et al., 2015) and SLR (Le & Hua, 2021) to see whether the learning rate scheme during retraining has any impact on the performance of the pruning selection scheme. Surprisingly the simple global selection criterion performs at least on par with the best out of all tested methods at any sparsity level for every combination of dataset and architecture tested here when considering the sparsity of the pruned network as the relevant measure. Using SLR during retraining compresses the results by equalizing performance, but otherwise does not change the overall picture. We note that the results on CIFAR-100 using FT largely track with those reported by Lee et al. (2020), with the exception of the strong performance of the global selection criterion. Apart from slightly different network architectures, we note that they used significantly more retraining epochs, e.g., 100 instead of 30, and that they use AdamW (Loshchilov & Hutter, 2019) instead of SGD. Comparing the impact different optimizers can have on the pruning selection schemes seems like a potentially interesting direction for future research.

While the sparsity-vs.-performance tradeoff has certainly been an important part of the justification of modifications to global selection criterion, let us also directly address two further points that are commonly made in this context. First, the global selection criterion has previously been reported to suffer from a pruning-induced collapse at very high levels of sparsity in certain network architectures that is avoided by other approaches. This phenomenon has been studied in the *pruning before training* literature and was coined *layer-collapse* by Tanaka et al. (2020), who hypothesize that it can be avoided by using smaller pruning steps since gradient descent restabilizes the network after pruning by following a layer-wise magnitude conservation law. To verify whether these observations also hold in the *pruning after training* setting, we trained a VGG-16 network on CIFAR-10, as also done by Lee et al. (2020), both in the One Shot and in the iterative setting. The results are reported in Figure 11 and show that layer collapse is clearly occurring for both FT and SLR for the global selection criterion at sparsity levels above 99% in the One Shot setting, but disappears entirely when pruning iteratively. This indicates that layer collapse, while a genuine potential issue, can be avoided even using the global selection criterion. We also remark that SLR needs less prune-retrain-cycles to avoid layer-collapse than FT, possibly indicating that the retraining strategy impacts the speed of restabilization of the network in the hypothesis posed by Tanaka et al. (2020).

The second important aspect to consider is that layer-dependent selection criteria are also intended to address the inherent tradeoff not just between the achieved sparsity of the pruned network and its performance, but also the theoretical computational speedup. Figure 8, Figure 9 and Figure 10 include plots highlighting the achieved performance in relation to the theoretical speedup. The key takeaway here is that for both the ResNet-56 and the WideResNet network architecture, there is overall surprisingly little distinction between all five tested methods, with Uniform+ and ERK taking the lead and the global selection criterion performing well to average. For the ResNet-50 architecture however a much more drastic separation occurs, with Uniform performing the best, followed by Uniform+ and then the global selection criterion. Overall, the picture is significantly less clear. However, despite its simplicity, the global approach performs on par with respect to managing the accuracy vs. speedup tradeoff, where we observe that for ResNet-50 on ImageNet it even outperforms methods such as LAMP and ERK regarding both objectives, performance and speedup.

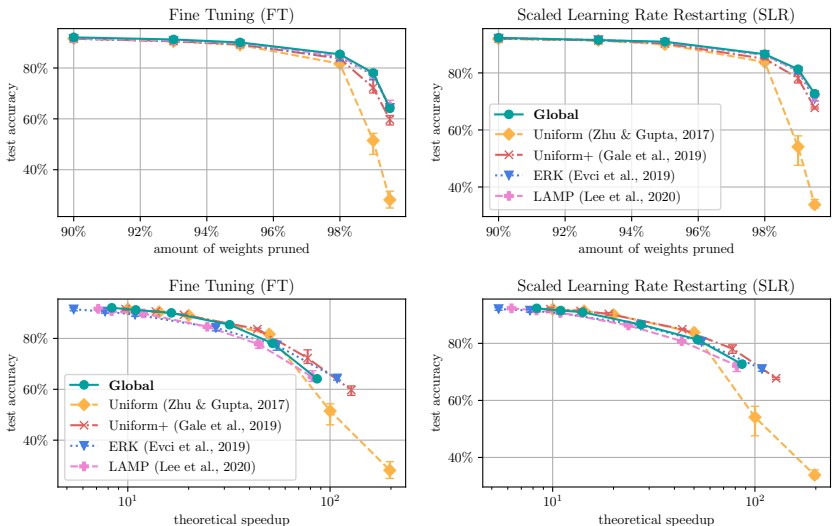

Figure 8: ResNet-56 on CIFAR-10 (One Shot): Sparsity-vs.-performance (above) and theoretical speedup-vs.-performance (below) tradeoffs in the One Shot setting with FT (left) and SLR (right) as retraining methods. Retraining is done for 30 epochs. The plot includes max-min confidence intervals.

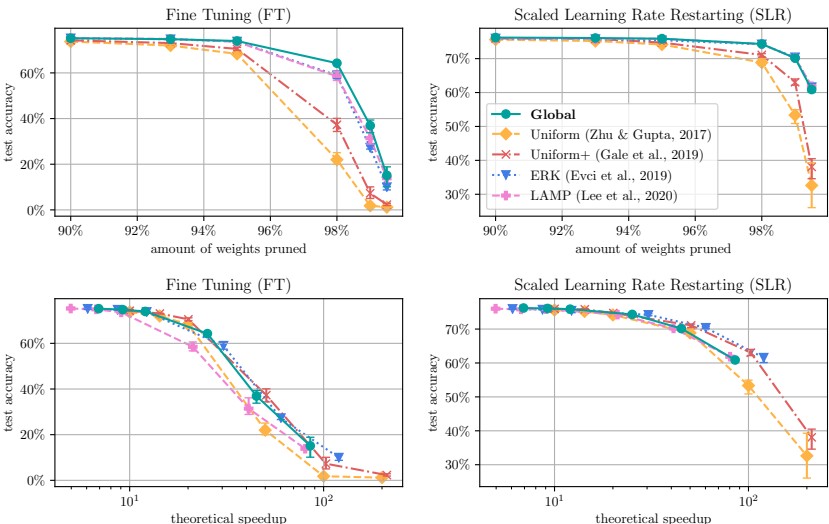

Figure 9: WideResNet on CIFAR-100 (One Shot): Sparsity-vs.-performance (above) and theoretical speedup-vs.-performance (below) tradeoffs in the One Shot setting with FT (left) and SLR (right) as retraining methods. Retraining is done for 30 epochs. The plot includes max-min confidence intervals.

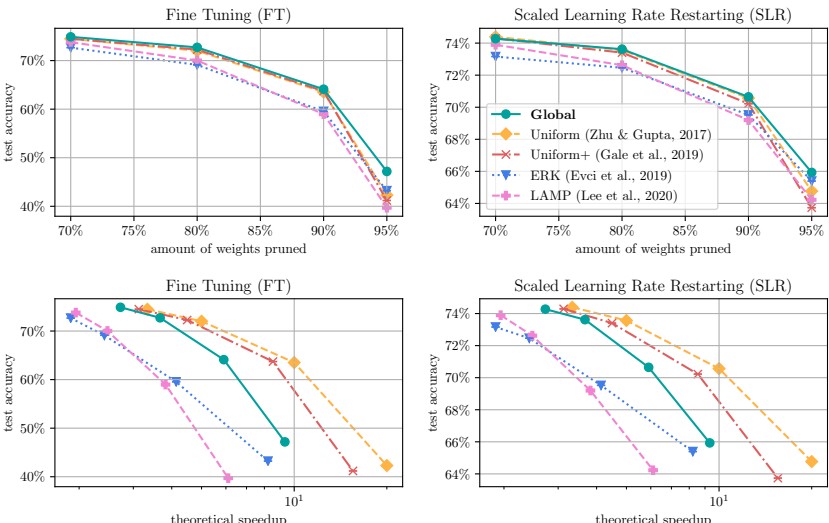

Figure 10: ResNet-50 on ImageNet (One Shot): Sparsity-vs.-performance (above) and theoretical speedup-vs.-performance (below) tradeoffs in the One Shot setting with FT (left) and SLR (right) as retraining methods. Retraining is done for 10 epochs.

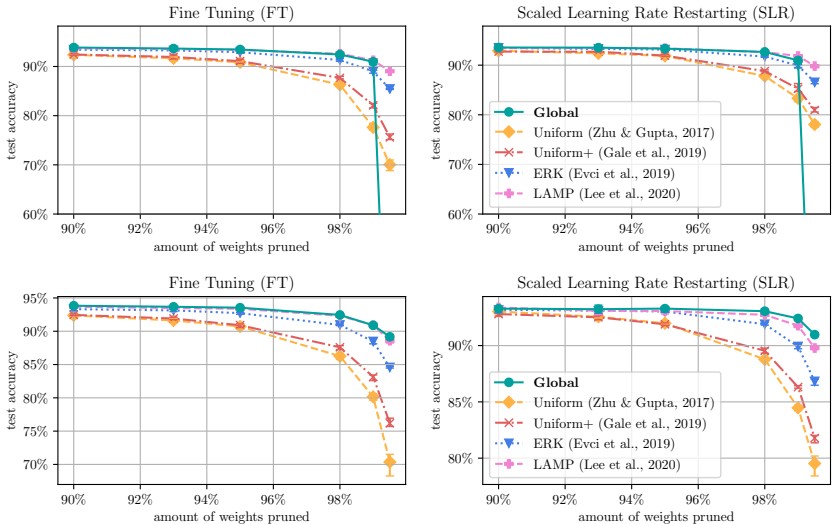

Figure 11: VGG-16 on CIFAR-10: Performance-vs.-sparsity tradeoffs in the One Shot (above) and Iterative (below) setting with FT (left) and SLR (right) as retraining methods. In the One Shot setting the model is retrained for 30 epochs after pruning and the iterative setting consists of 3 prune-retrain cycles with 10 epochs each. For One Shot we observe *layer-collapse* while the iterative splitting into less severe pruning steps avoids the problem. Note that the total amount of retraining epochs between the two settings is identical here.

### C.3 COMPARING BIMP TO GMP

We note that BIMP has a fair number of similarities to GMP (Zhu & Gupta, 2017) and we will therefore seek the direct comparison between the two. In particular, both methods effectively prune and retrain (although that terminology is commonly reserved to IMP) at distinct predetermined points during the overall training process. Let us start by highlighting the particular design decisions by which BIMP and GMP differ:

1. BIMP 'hard' prunes while GMP 'soft' prunes. In GMP, pruned weights are zeroed out during the forward and backward pass, but the mask is recomputed at the next pruning step, hence allowing for previously pruned weights to recover. However, it is unclear for example how the momentum buffer of SGD or the memory of optimizers like Adam (Kingma & Ba, 2014) are supposed to take this soft pruning into consideration, but we believe that this aspect overall contributes very little to explain the different performance of GMP and BIMP.

2. BIMP requires sufficiently long retraining time between two pruning points to recover from the last pruning step. GMP on the other hand can have much shorter distances between the equally distributed pruning points with as little as pruning every 100 training iterations (Zhu & Gupta, 2017).

3. Both BIMP and GMP rely on magnitude pruning, however we have decided to use the simple GLOBAL selection criterion of Han et al. (2015) for both while GMP was originally proposed using the UNIFORM selection and further improved by Gale et al. (2019) with the UNIFORM+ criterion, see Appendix C.2 for a discussion. The version of GMP included in the main part of this text in fact relies on the same global criterion as BIMP, since we found it to yield better results than UNIFORM+ with respect to the final accuracy.

4. BIMP relies on a cyclic linear learning rate schedule where the cycles coincide with the points at which the network is pruned, while GMP can use any kind of learning rate scheme that would commonly be employed for that kind of architecture and data set, i.e., normally not a cyclic one. Zhu & Gupta (2017) note that GMP can be quite sensitive to the learning rate. In the main body of the text we have mostly relied on a common stepped learning rate schedule. Given the importance of the learning rate schedule, we have also tested a version of GMP using both a linearly decaying schedule, as well as a cyclic learning rate schedule similar to what we have suggested for BIMP, i.e., the cycles are chosen to exactly end at the next pruning step.

5. We have relied on the simple exponential pruning schedule suggested by Renda et al. (2020) for BIMP while GMP relies on a particular schedule defined by a cubic polynomial that effectively leads to pruning larger amounts initially and progressively smaller amounts later in training when compared to BIMP. While we think that the pruning schedule probably has a significant impact on the performance of the pruning method and can possibly interact with the learning schedule in particularly interesting ways, we have so far not attempted exchanging the schedule in BIMP for either the one of GMP or a novel one.

In Table 28 we have included the previously mentioned modifications of GMP and compared them to BIMP for WideResNet trained on CIFAR-100. In particular, for each variant of GMP we indicate which learning rate schedule we use and whether we prune 'hard' or 'soft'. The models are trained according to the same settings as indicated in Table 3, where *stepped* refers to the stepped learning rate case. On the other hand, *linear* indicates a linear learning rate schedule and *cyclic* refers to a linearly decaying learning rate schedule that is restarted after every pruning point. We use a weight decay value of 1e-4 and set the initial value of the learning rate to 0.1. For GMP we prune every 5 or 10 epochs, while for BIMP we fix the initial training length to a 100 epochs and split the remaining 100 epochs equally between 1 to 4 cycles. All results are averaged over two seeds with standard deviations indicated.

We note that there seems to be surprisingly little difference between hard and soft pruning. The impact of the learning rate schedule is more nuanced: using a linearly decaying schedule throughout training can give a slight increase in performance, albeit the classical stepped learning rate schedule seems to work better in the high sparsity regime. Whether the cyclic restarting of the learning rate works crucially depends on the distance between two pruning points and the impact of pruning, which BIMP with ALLR seems to leverage. For the medium sparsity of 90% a cyclical learning rate seems to be detrimental, while in the high sparsity regime we see improvements.

Table 29 further reports results for ResNet-50 trained on ImageNet, where we sticked to the 'hard' pruning for each GMP variant. Similarly, we use the same settings as in Table 3, set the weight decay to 1e-4 and the initial value of the learning rate to 0.1. GMP prunes every 5 or 10 epochs, whereas BIMP leverages an initial training length of 60 or 75 epochs, splitting the remaining 30 or 15 epochs into 1 to 4 cycles of equal length. Here, the stepped learning rate schedule seems to be in advantage over a linear one, with the slight exception of the highest sparsity. Interestingly, the cyclic learning rate schedule seems to be in conflict with and too aggressive for the sparsification schedule of GMP. Overall we think that there is a fair amount of nuance here that deserves further exploration and will probably require a more diverse testbed to draw any definitive conclusions.

Table 28: WideResNet on CIFAR-100: Comparison between BIMP and GMP variants for goal sparsity levels of 90% and 95%, denoted in the main columns. Each subcolumn denotes the Top-1 accuracy, the theoretical speedup and the actual sparsity achieved by the method. All results are averaged over two seeds and include standard deviations.

| Method | Model sparsity 90% | | | Model sparsity 95% | | |
|---|---|---|---|---|---|---|
| | Accuracy | Speedup | Sparsity | Accuracy | Speedup | Sparsity |
| **GMP (stepped, hard)** | 75.19 ±0.25 | 8 ±0.0 | 90.00 ±0.00 | 74.62 ±0.02 | 15 ±0.2 | 95.00 ±0.00 |
| **GMP (linear, hard)** | 75.44 ±0.23 | 8 ±0.0 | 90.00 ±0.00 | 74.40 ±0.01 | 16 ±0.1 | 95.00 ±0.00 |
| **GMP (cyclic, hard)** | 75.09 ±0.25 | 8 ±0.0 | 90.00 ±0.00 | 74.81 ±0.47 | 16 ±0.1 | 95.00 ±0.00 |
| **GMP (stepped, soft)** | 75.32 ±0.04 | 8 ±0.0 | 90.00 ±0.00 | 74.29 ±0.32 | 15 ±0.2 | 95.00 ±0.00 |
| **GMP (linear, soft)** | 75.52 ±0.05 | 8 ±0.1 | 90.00 ±0.00 | 74.86 ±0.34 | 16 ±0.0 | 95.00 ±0.00 |
| **GMP (cyclic, soft)** | 74.79 ±0.12 | 8 ±0.0 | 90.00 ±0.00 | 74.65 ±0.19 | 16 ±0.0 | 95.00 ±0.00 |
| **BIMP** | 75.77 ±0.23 | 8 ±0.1 | 90.00 ±0.00 | 75.30 ±0.34 | 15 ±0.1 | 95.00 ±0.00 |

Table 29: ResNet-50 on ImageNet: Comparison between BIMP and GMP variants for goal sparsity levels of 70%, 80% and 90%, denoted in the main columns. Each subcolumn denotes the Top-1 accuracy, the theoretical speedup and the actual sparsity achieved by the method. All results are averaged over two seeds and include standard deviations.

| Method | Model sparsity 70% | | | Model sparsity 80% | | | Model sparsity 90% | | |
|---|---|---|---|---|---|---|---|---|---|
| | Accuracy | Speedup | Sparsity | Accuracy | Speedup | Sparsity | Accuracy | Speedup | Sparsity |
| **GMP (stepped)** | 74.62 ±0.08 | 2 ±0.0 | 70.00 ±0.00 | 74.19 ±0.17 | 4 ±0.0 | 80.00 ±0.00 | 72.74 ±0.06 | 7 ±0.0 | 90.00 ±0.00 |
| **GMP (cyclic)** | 72.91 ±0.19 | 2 ±0.0 | 70.00 ±0.00 | 72.67 ±0.10 | 4 ±0.0 | 80.00 ±0.00 | 71.76 ±0.00 | 7 ±0.1 | 90.00 ±0.00 |
| **GMP (linear)** | 74.58 ±0.01 | 2 ±0.0 | 70.00 ±0.00 | 73.90 ±0.04 | 4 ±0.0 | 80.00 ±0.00 | 72.80 ±0.03 | 7 ±0.1 | 90.00 ±0.00 |
| **BIMP** | 75.62 ±0.02 | 2 ±0.0 | 70.00 ±0.00 | 75.08 ±0.16 | 3 ±0.0 | 80.00 ±0.00 | 73.53 ±0.05 | 6 ±0.0 | 90.00 ±0.00 |

