# OpenReview forum: "How I Learned to Stop Worrying and Love Retraining"
_ICLR.cc/2023/Conference — ICLR 2023 poster_

### Official Review · Reviewer_HL8u · 2022-10-23

**Confidence:** 5
**Correctness:** 3
**Technical Novelty And Significance:** 3
**Empirical Novelty And Significance:** 3
**Recommendation:** 8

**Clarity, Quality, Novelty And Reproducibility:**

This paper is clearly up to standards in paper quality, with it being very well written and organized. As far as I'm aware applying such a training budget and achieving such good results is novel, and of interest to the broader research community. Although I worry the originality of the work will be debated due to its simplicity, the simplicity of the method and strength of the results make this a stronger paper.

**Strength And Weaknesses:**

# Strengths
* Well-written and well-organized paper, with a clear motivation, excellent background/references
* Method is simple and explained well overall
* Figures and results are presented well
* Empirical evaluation is (mostly) on reasonable datasets/models (exception being VGG, see below)
* Questions existing wisdom in the sparse neural network community that "pruning stable" approaches (i.e. methods without hard/fixed masks during training) to the finding sparse neural networks are preferable to "pruning unstable" approaches (i.e. methods with hard masks).
* Strong results mean this could be a new baseline for "pruning unstable" methods

# Weaknesses
* The authors significantly weaken the impact of their work by essentially over-claiming that their work shows that "pruning stable" methods are pointless, without the results to back it up. In a notable and questionable omission in the results, the authors do not compare to the state-of-the-art dynamic sparse training methods despite citing them, for example RiGL. On ImageNet/ResNet-50 the BIMP results are massively below those of RiGL. For example, in Table 2, at 90% sparsity, BIMP has a Top-1 accuracy of 70.25%, while RiGL achieves 72-76.4% (saying based on ERK/non-ERK and length of training).
* The authors point out rightly that RiGL and other such methods are massively more computationally complex and require longer training - and if they instead focused on this argument in their results, rather than generalization alone while omitting RiGL, this would be a much stronger paper.
* Intuition/reasoning behind the ALLR method of using the normalized distance between weight vectors to decide an initial learning rate is not made clear, and this reads as quite arbitrary/heuristic as-is.
* VGG is much too over-parameterized to be in a sparsity-related paper in 2022.

**Summary Of The Paper:**

The authors propose that recent work on budgeted training in dense models applies in the context of both the dense training and sparse re-training phases of Iterative Magnitude Pruning (IMP) - perhaps the most common form of pruning. In particular the authors propose two methods of pruning/re-training consisting of relatively short total training duration compared to existing work, and yet achieving strong results compared to those much longer-trained baselines.

**Summary Of The Review:**

Overall the paper is well-written, with a good background, experimental setup and analysis. The paper sets strong new baselines for "pruning unstable" methods, such as Iterative Magnitude Pruning. Although these results are used by the authors to compare with "pruning stable" methods, such as Dynamic Sparse Training methods (in general), in not comparing to the widely known DST methods with the best generalization results (in particular RiGL on ImageNet/ResNet-50) weakens this argument significantly, and avoiding comparing to such a well-known and strong baseline in general when its cited by the authors raises questions. The authors do make very good points on the relative computational complexity of such methods however which even in the face of RiGL results still leaves this work motivating, and of interest to the sparse neural network research community.

---

> ### Author Response · Authors · 2022-11-16
> **Reply to Reviewer HL8u #1**
>
> We thank you for your detailed review and interest in our research. In the following, let us address your concerns.
>
> > The authors significantly weaken the impact of their work by essentially over-claiming that their work shows that "pruning stable" methods are pointless, without the results to back it up. In a notable and questionable omission in the results, the authors do not compare to the state-of-the-art dynamic sparse training methods despite citing them, for example RiGL. On ImageNet/ResNet-50 the BIMP results are massively below those of RiGL. For example, in Table 2, at 90\% sparsity, BIMP has a Top-1 accuracy of 70.25\%, while RiGL achieves 72-76.4\% (saying based on ERK/non-ERK and length of training).
>
> Before addressing your criticism in detail, please let us make clear that it is in fact not our claim that "pruning-stable methods are pointless", such a statement would not backed up by evidence. To the contrary, we are trying to argue that methods that need retraining, such as IMP, are neither inefficient nor deficient when compared to pruning-stable methods, as is often argued in the literature. That being said, let us clarify some general points regarding our work.
>
> First of all, our goal has always been to compare pruning-after-training algorithms to those that fall in the category of pruning-during-training, as we state in the first paragraph of section 2. However, both of these paradigms are based on a dense-to-sparse training scheme (cf. e.g. Kusupati et al., Hoefler et al.), being entirely different than approaches that follow a 'sparse-to-sparse' approach, which can further be divided into *Dynamic Sparse Training* methods (such as RiGL, SNFS) or pruning-before-training approaches (such as e.g. SNIP). All of the methods compared in our work fall into the category 'dense-to-sparse', a supposed exception being DST (an acronym for Dynamic Sparse Training), which however relies on the dense model in every single iteration, zeroing out weights in the forward-pass while computing dense gradients at all times. Kusupati et al. for example name DST as concurrent to their work, since it is similarly based on learnable sparsification masks, however belonging to the realm of dense-to-sparse training. That being said, we think that RiGL is entirely different in its approach since it starts with a sparse model (i.e., rigging the lottery) and was therefore intentionally excluded in our comparison. Sparse-to-sparse methods have the clear advantage of profiting from the sparsity-induced speedups throughout training (assuming exploitable sparsity patterns), making the comparison more complicated.
>
> Secondly, we think that using numeric results from other publications itself poses a problem, since the exact training settings are never the same. In fact, RiGL is trained for more epochs using a different learning rate schedule. The authors also relied on an entirely different batch size as well as different augmentations of the train data such as label smoothing. For that reason we implemented each of these methods from scratch (relying on existing implementations whenever possible) and compared them in a unified setting in order to allow for a fair comparison.
>
> Last but not least, we take your concerns serious and do not want to give any reader the impression of our computational study purposely avoiding certain methods to strengthen our claims. To that end and despite the aforementioned concerns regarding the comparison of sparse-to-sparse to dense-to-sparse methods, we implemented RiGL and have run preliminary experiments on ImageNet in the same setting as the other methods at stake. We followed the authors' recommendations and included their suggested hyperparameters in a larger grid search. We performed both Uniform as well as ERK-pruning, varied the alpha in the decay function between 0.3 and 0.5, and tuned the pruning/growing interval between 100 and 200 iterations. Given the limited rebuttal timeframe, we restricted the learning rate and weight decay to the default setting. We noted that uniform pruning typically results in higher theoretical speedups, albeit at the price of the final performance. This aligns with our previously reported findings on the pruning selection criteria (cf. Appendix C.2). Our results for RiGL are as follows, where we report the best configuration for each goal sparsity:
> - Sparsity 70\%: Test Accuracy 72.24\%($\pm$ 0.07\%) | Speedup 1.87x
> - Sparsity 80\%: Test Accuracy 71.06\%($\pm$ 0.04\%) | Speedup 2.41x
> - Sparsity 90\%: Test Accuracy 68.09\%($\pm$ 0.02\%) | Speedup 4.13x
>
> Clearly, RiGL is a strong method, but profits from much longer training times (the authors use a factor 5), which might be justifiable since the network is sparse at all times, drastically reducing the number of FLOPs training requires.

---

> > ### Comment · Reviewer_HL8u · 2022-11-22
> > **Author Rebuttal**
> >
> > First I'd like to thank the authors for their detailed and significant rebuttal comments. My apologies that this reply comes late in the discussion period.
> >
> > I agree with your point as you have clarified it here, and your claims/motivation as explained here is perfectly reasonable given your results/analysis, and would result in a much stronger paper overall. Unfortunately in reading your original paper however I was very clearly left with the view that your claim was that pruning stable methods are pointless. I would significantly revise your writing to make it clear this is not your intent/claim.
> >
> > Similarly I think your paper will be much clarified if you explicitly explain what you did here, in that you only compared to dense-to-sparse and not sparse-to-sparse (and I agree none of these definitions are as concrete as that, but we can only try!). As your paper was you didn't discuss DST methods much at all in relation to your work, and so the reader is left to fill in the blanks. Given this, I personally read it as an intentional oversight. Clearly touching upon the relationship of DST methods to the area you are looking for will clarify this a great deal and I believe address this problem.
> >
> > I do agree that comparing numerical results between papers can be problematic. However given the RiGL results are so much better at face value than those presented here, it's simply something again that must be addressed or the reader will be left to fill in the gaps. I would note that even for the non-5x (i.e. 1x) RiGL training schedule, the results are significantly better, so I'm not convinced this is just due to length of training. In any case I appreciate the author's attempt to address this with results given the short rebuttal period. I think the summary is that the results are close in the best case, and likely practically RiGL is better - however I think the authors have arguments to make on the simplicity of their approach, training time, and potentially the fact that dense gradient are required for RiGL at mask updates, etc. They are very different approaches, and I like that the authors are pointing out how well the simple approach works.
> >
> > I'm hopeful that the authors can revise the text of the paper to address the issues I had exactly as they did in the strong rebuttal above. Conditional upon such revisions I would lean towards an accept.

---

> > > ### Author Response · Authors · 2022-11-24
> > > **Reply to Reviewer HL8u #3**
> > >
> > > Thank you for your fast reply! We are happy to hear that we agree on these issues and are working on making the discussed changes to avoid that the reader is 'left to fill in the blanks'! We are glad to incorporate the suggested changes since we agree that that they improve the clarity of our work. At this point of the review phase we are unfortunately unable to upload further revisions of the manuscript, but we will fully address your concerns in a potential camera-ready version. We thank you again for the fruitful discussion!

---

> ### Author Response · Authors · 2022-11-16
> **Reply to Reviewer HL8u #2**
>
> > The authors point out rightly that RiGL and other such methods are massively more computationally complex and require longer training - and if they instead focused on this argument in their results, rather than generalization alone while omitting RiGL, this would be a much stronger paper.
>
> As stated in the above comment, we think that other dense-to-sparse methods can be computationally much more complex - however RiGL, being a sparse-to-sparse method, can in fact require less FLOPs for computing a sparse solution. As Evci et al. themselves state, the cost of RiGL's pruning-and-growing can be amortized. The point that many pruning-stable methods in fact more computationally complex is reflected by the \#images-per-second throughput we included in every table comparing such methods. We will consider making this argument more present in the current manuscript and are happy to take suggestions.
>
> > Intuition/reasoning behind the ALLR method of using the normalized distance between weight vectors to decide an initial learning rate is not made clear, and this reads as quite arbitrary/heuristic as-is.
>
> Please see the official comment on this issue, where we also refer to the ablation studies in Appendix C.1, which provides more detailed reasoning for ALLR.
>
> > VGG is much too over-parameterized to be in a sparsity-related paper in 2022.
>
> We agree, VGG is not an appropriate architecture for this purpose. For this reason, we did not use it in the main computational study and only relied on it in Appendix C.2, where we analyzed the effect of retraining schemes given different pruning selection criteria (such as Uniform, Global, ..). VGG was purposefully chosen here  to illustrate the effect of layer-collapse.
>
> We are happy to hear your thoughts and to discuss further on how to improve our work.

---

### Official Review · Reviewer_cREN · 2022-10-27

**Confidence:** 4
**Correctness:** 4
**Technical Novelty And Significance:** 3
**Empirical Novelty And Significance:** 3
**Recommendation:** 8

**Clarity, Quality, Novelty And Reproducibility:**

Quality: The experiments are very well thought out and conducted with great care. I do miss some variety in the selected models/datasets.

Clarity: In my opinion, this paper is very well-written. It does a great job of positioning itself in the context of related work. Most steps are very well motivated.

Originality: There is little technical/methodological novelty in the proposed approach. However, the budgeted training perspective and the careful experimental comparison gives very useful insights.

**Strength And Weaknesses:**

### Strengths

- The paper does a very good job of explaining existing work and positioning itself in that context. This makes the work accesible to readers not familiar with minute details of pruning methods.

- While one could see the proposed technique as only a slight variation of some existing methods, the simple linear step size schedule is well-motivated from the "budgeted training" perspective. It is simpler than the schedules used in prior work while performing at least as good, and often better, across a wide range of "retraining budgets" and target sparsities.

- The experimental comparison is conducted with great care. The experimental set-up is motivated and explained in detail, including hyperparameter settings and/or tuning protocols. The results are presented in very clearly and their significance is discussed in detail.

- I want to explicitly commend the authors for making clear the limitations of the proposed method and to focus on generating useful insights rather than trying to coin another acronym.

### Weaknesses

- I find the motivation for **A**LLR quite unsatisfying. The relative distance computed in Eq. (1) strikes me as quite a poor proxy for what we are really interested in, which is to quote the paper "how much of an increase in loss do we have to compensate for". At the same time, this increase in loss could actually be computed at the expense of an additional validation epoch. In contrast to what's stated in the paper, I don't think that would be a big computational burden. The evaluation could be done on the validation set (or possibly even a subset thereof) and requires only a forward pass. Its cost should therefore be minor compared to the actual retraining for multiple passes over the training set.

- The experiments are restricted to image classification tasks. For breadth, it would have been desirable to include another data modality and/or task.

**Summary Of The Paper:**

This paper studies the retraining steps in iterative magnitude pruning from the "budgeted training" perspective. This is used to derive an efficient learning rate schedule for the retraining phase.

**Summary Of The Review:**

I think this is a very well-written paper with a solid contribution to the model pruning literature. I was slightly disappointed by the motivation for **A**LLR and would encourage the authors to respond to this during the rebuttal.

---

> ### Author Response · Authors · 2022-11-16
> **Reply to Reviewer cREN**
>
> We thank reviewer cREN for the interest in our work and are happy to answer the questions and address the concerns raised.
>
> > I find the motivation for ALLR quite unsatisfying. The relative distance computed in Eq. (1) strikes me as quite a poor proxy for what we are really interested in, which is to quote the paper "how much of an increase in loss do we have to compensate for". At the same time, this increase in loss could actually be computed at the expense of an additional validation epoch. In contrast to what's stated in the paper, I don't think that would be a big computational burden. The evaluation could be done on the validation set (or possibly even a subset thereof) and requires only a forward pass. Its cost should therefore be minor compared to the actual retraining for multiple passes over the training set.
>
> We agree that one could argue that a single validation epoch is not much of a big computational burden. However, since all existing retraining schemes rely on very simple, easily implementable mechanisms, we decided to follow this path to provide a straightforward way to selecting the initial value of the learning rate. Further, introducing such a validation epoch could provide for an unfair comparison to the existing schemes. In fact, Appendix C.1 contains several ablation studies regarding our design choices, including a comparison between ALLR and a variant of it close to what you suggest (despite computing the performance degradation on the train set). We just extended these ablation studies to contain ImageNet and think that the results highlight that Eq. (1) is a good proxy for the performance degradation. Please also see the official comment in that regard.
>
> > The experiments are restricted to image classification tasks. For breadth, it would have been desirable to include another data modality and/or task.
>
> We absolutely agree with you on this point, which we had noted in the discussion of our work. To strengthen our claims, we used the review- and rebuttal-phase to extend our experiments by a large body of experiments, including MaxVisionTransformers on ImageNet, Semantic Segmentation experiments on the benchmark Coco and Cityscapes datasets as well as Neural Machine Translation on WMT16 (DE-EN) using a T5-transformer architecture. Please see the official comment for a more detailed account.

---

### Official Review · Reviewer_Ybn7 · 2022-10-31

**Confidence:** 4
**Correctness:** 3
**Technical Novelty And Significance:** 2
**Empirical Novelty And Significance:** 3
**Recommendation:** 6

**Clarity, Quality, Novelty And Reproducibility:**

### Clarity
There are a few areas in the paper that suffer from a lack of clarity:
- Point 2 of the major takeaways is awkwardly phrased.  If there is more evidence supporting a linear LR schedule, why not come out and say so?  Putting it in parentheses weakens the idea.
- The first sentence after stating the major takeaways states the authors' claim that retraining should be considered in a budgeted scenario, but at this stage, It’s not clear why.  Do the authors believe this because budgeted results give the best improvement per parameter update?  Do they have better generalization performance?  They should call out

This section has laid out the cases for not using a truncated learning rate schedule, and not using a fixed learning rate schedule, but it has not argued successfully for why the budgeted learning rate schedule is more advantageous.

- Section 2.2 lays out two commonly stated advantages of pruning-stable methods, and goes on to argue why IMP should prevail.  I think the authors can make the case for Budgeted IMP more clearly.  They cast doubt on the potential of pruning stable algorithms, and propose that instead of aiming at a complicated method to prune in one shot, instead the cost of IMP should be reduced by training on a budget.  This way, the main drawback of IMP can be removed, and the network can achieve good performance though a more careful choice of an initially aggressive learning rate coupled with the right choice of initial LR in each subsequent phase of  post-pruning recovery training.

Figure 2 repeats much of the story of Table 1, which is that the task only gets hard enough to differentiate the methods at 80% sparsity and up.  It is somewhat interesting that there is a noticeable difference in behaviour between the top performers (ALLR,  LLR, CLR) between the 80% sparsity task and the 90% sparsity task.  In the former, ALLR is universally dominant, while in the latter, it is indistinguishable from LLR and CLR until at least 20 epochs of retraining are budgeted.  Is this to be expected?  And what does this say about  how the advantage of adjusting the initial learning rate carefully (as ALLR enjoys over LLR) interacts with retraining budget to and task difficulty?

### Novelty

- The main novelty here is in extending the definition of what constitutes a training interval under Budgeted training, and of adaptively choosing a new learning rate after each phrase of retraining pursuant to magnitude based pruning.  It's clear that the development of ALLR is an improvement on LLR, but I'm not sure that the addition of a heuristic-based adaptive learning rate scheme is truly novel, given the extensive literature on adaptive learning rates for neural networks.




**Strength And Weaknesses:**

### Strengths
- The authors demonstrate a solid understanding of the literature on pruning
- The figures and tables are quite clear, establishing the benefit of adaptive LLR in certain settings.
- The experiments, within their original parameters, are extensive enough to demonstrate the advantage of LLR and ALLR over the competing methods.

### Weaknesses
- There are a number of claims in the paper that do not agree with the published literature.  For instance, in the first paragraph of the introduction, they claim that a heavily pruned model will normally be less performant than its dense (or moderately pruned) counterpart.  But this isn't always true.  For example, Lottery-Ticket hypothesis [works](https://proceedings.mlr.press/v119/frankle20a.html) show that pruning even up to 80 to 90% of the weights of an original network.  I think the authors should either be more careful in their claims, or more precise in their writing.
- At the end of the introduction, the authors state their results build upon work by Renda et al, Le & HUa by proposing  how to choose the initial value of the learning rate, a problem not previously addressed.  Key works missing here are: *in a pruning context*.  This is a fine point to make, but a necessary one to avoid claiming that the problem of choosing an initial value of the learning rate for retraining a network has not previously been addressed.  For instance, every simple transfer learning problem has to solve this issue, by fine-tuning new classification weights (or more output-proximal representation layer weights) to solve a new task, which i do not see as fundamentally different than retraining following pruning.  I would walk back this claim. I also note that the reviewers of Le and Hua found the same criticism.
- In section 2.1, the authors declare themselves for Li et al in the battle of ideas about re-training a learned network.  That may be so, but what I have yet to see here, (and to be fair in the short readings of any of the competitions like Renda et al., or Le & Hua), is the question of how this problem differs form the fine-tuning problem of any base model (or foundation model).

I think the authors would help separate their work from these other lines by clearly delineating what aspects of the retraining problem they think are the most salient, and then arguing why the budgeted approach yields the best results.

- The connection of the learning rate scheduler to generalization (in Li et al as well) is empirical, and not well established.  Appealing to this as justification is wishful thinking. Over all, section 2.1 feels poorly structured.  Where does the related work summary end, and the authors’ own work begin?  It could be made more clear by the use of sub-sectioning to delineate the concepts of pruning, learning rate schedules, and generalization.

- A minor quibble about equation (1): you cannot have the interval of $s$ include $0$, as that leaves $d_1$ undefined by virtue of division by zero.

- It’s also unclear how this is supposed to stand as a proxy for the degradation in on-task training accuracy, which the authors claim in the preamble to equation (1). There might be a more grounded argument to be made here (perhaps approximating the function of $W$ -> accuracy for a fixed evaluation set?

- The final paragraph of section 2 cursorily sketches out ideas in selecting weights to be pruned, but only considers magnitude based pruning.  There are competing approaches that are sufficiently different, and deserve to be mentioned in here, such as  [Tanaka et al.](https://proceedings.neurips.cc/paper/2020/file/46a4378f835dc8040c8057beb6a2da52-Paper.pdf) who propose a flow-based method for pruning weights.

- Finally, I am sure I’m not the first to raise this issue, but there is surely an in-built bias resulting from the authors’ choice to study only multi-class classification in image data sets, using one family of convolutional networks.  It would be more convincing if the authors could have included some diversity in either their model families chosen, or their tasks considered.  The authors acknowledge this failing in their work  as a footnote in the concluding discussion, but if they are to follow in the steps of  Li et al 2020 (who use tasks from four different domains), then they should present more evidence that their method has an empirical benefit beyond that of this narrow domain.

**Summary Of The Paper:**

The authors address the problem of network pruning under a fixed training budget.  Earlier work on network pruning relied on computationally expensive iterative training & pruning regimes.  The authors show that when operating under a fixed training regime, a lot of savings can be found by eschewing complex learning rate schedules for a simple but aggressive linear learning rate schedule.  The authors further propose improvements to the initial choice of learning rate in the schedule, as well as adding more structure to the budget of the dense network training, which together result in real value per unit of computation.  They demonstrate good performance against alternative methods which induce sparsity during training, and challenge the community to reconsider the commonly held tenet that retraining is wasteful.

**Summary Of The Review:**

While the authors convinced me that the addition of budgeting can save IMP, I am less than convinced of the generality of their results to other settings than well-studied vision data sets.  I think that the authors should either extend their set of empirical results to encompass different problems (and different networks), or to reach into theory of generalization papers to try and support their claims for adaptive learning rates and budgeted retraining.

---

> ### Author Response · Authors · 2022-11-16
> **Reply to Reviewer Ybn7 #1**
>
> Thank you for your extensive review. In the following, let us try to address your concerns.
>
> **Difference to general adaptive learning rate schedules and experimental diversity.**
> > At the end of the introduction, the authors state their results build upon work by Renda et al, Le \& Hua by proposing how to choose the initial value of the learning rate, a problem not previously addressed. Key works missing here are: in a pruning context.
>
> Thank you for your remark. We agree and changed the phrasing accordingly. As is well known, choosing a suitable learning rate schedule (including starting value) is essential for every training task in the field of Deep Learning. Our intention was to very specifically address this issue in the context of pruning and not more broadly (though this certainly could be of interest for future research).
>
> > In section 2.1, the authors declare themselves for Li et al in the battle of ideas about re-training a learned network. That may be so, but what I have yet to see here, (and to be fair in the short readings of any of the competitions like Renda et al., or Le \& Hua), is the question of how this problem differs from the fine-tuning problem of any base model (or foundation model).
>
> Comparing the issue of retraining after pruning to finetuning a base model to a new task is an interesting perspective, we however think that there are indeed several differences which we want to highlight. First of all, pruning changes the current parameters while the global loss landscape remains unchanged. From our understanding, the problem of finetuning to a new tasks involves a new dataset and consequently an entirely different problem (the initial learning rate for finetuning is hence also dependent on the domain similarity, cf. [2]). Secondly, finetuning typically restricts the retraining to specific layers, e.g., a last fully connected layer, the selection of which is considered to be a major problem in finetuning itself [1].  Pruning affects all layers and it is unclear whether it is sufficient to retrain just a few of them. Lastly, the full potential of pruning is unveiled when performing it iteratively, i.e., in multiple cycles. At first sight, these prune-retrain-cycles seem independent of one another, however the retraining schedule of each cycle clearly affect subsequent cycles, not only w.r.t. the final performance but also regarding the stability to pruning after each cycle, since the learning rate affects the stabilization of the network inbetween cycles (cf. e.g. Tanaka et al.). To the best of our knowledge, there is no analogue to this interdependent cyclic behaviour in the context of finetuning to a new task.
>
> Nevertheless, since there is indeed some similarity between the two problems, we think that this further supports a central point we are trying to make, namely that, similarly to the finetuning problem you suggest, the retraining phase is first and foremost a training phase which requires a suitable chosen learning rate schedule. Consequently, we show that given the limited timeframe it is desirable to choose a linear one and given the special setting of different timeframes and different pruning impacts we propose to adaptively choose the initial value, resulting in ALLR. As outlined in section 2, we think that previous works on the retraining issue were unable to explain the success of their respective methods. Especially LRW (Renda et al.) is a highly cited retraining variant, which is used throughout various other works as a de facto default routine for retraining, despite being a variant of Weight Rewinding which is applied in a different context, namely that of the Lottery Ticket Hypothesis and the search for 'winning tickets'. As we show, it is possible to significantly improve upon LRW and other approaches when viewing the problem as a budgeted training scenario.
>
> > I am sure I’m not the first to raise this issue, but there is surely an in-built bias resulting from the authors’ choice to study only multi-class classification in image data sets, using one family of convolutional networks. It would be more convincing if the authors could have included some diversity in either their model families chosen, or their tasks considered.
>
> We absolutely agree with you on this issue. To strengthen our claims, we used the review- and rebuttal-phase to extend our experiments by a large body of experiments, including MaxVisionTransformers on ImageNet, Semantic Segmentation on the benchmark Coco and Cityscapes datasets as well as Neural Machine Translation on WMT16 (DE-EN). Please see the official comment for a more detailed account.

---

> > ### Author Response · Authors · 2022-11-16
> > **Reply to Reviewer Ybn7 #2**
> >
> > **Clarity and other weaknesses.**
> > > Over all, section 2.1 feels poorly structured. Where does the related work summary end, and the authors’ own work begin?
> >
> > We agree that the presentation of section 2.1 can be improved. We thank you for that remark and will try to improve the clarity and structure.
> > > There are a number of claims in the paper that do not agree with the published literature. For instance, in the first paragraph of the introduction, they claim that a heavily pruned model will normally be less performant than its dense (or moderately pruned) counterpart. But this isn't always true.
> >
> > The specific example you mention is lacking some context that we give immediately prior: "Although it has been observed that pruning might have a regularizing effect and be beneficial to the generalization capacities (Blalock et al., 2020; Hoefler et al., 2021), a very heavily pruned model will normally be less performant than its dense (or moderately pruned) counterpart." Of course 'very heavily pruned' is somewhat vague and we will make sure to improve this phrase, but overall we think this aligns with existing literature (cf. e.g. Hoefler et al., Figure 4) as well as with your observations.
> > > A minor quibble about equation (1): you cannot have the interval of $s$ include $0$, as that leaves $d_1$ undefined by virtue of division by zero.
> >
> > Thank you for pointing this out. We have changed it accordingly.
> > > It’s also unclear how this is supposed to stand as a proxy for the degradation in on-task training accuracy, which the authors claim in the preamble to equation (1).
> >
> > Please see the official comment on this issue, where we also refer to an ablation study we conducted on CIFAR-10 and ImageNet.
> > > The final paragraph of section 2 [...] only considers magnitude based pruning. There are competing approaches [...] such as Tanaka et al. who propose a flow-based method for pruning weights.
> >
> > We agree that there is a variety of different criteria, with magnitude pruning arguably being among the most popular ones. Our pruning-after-training experiments focus on IMP and hence magnitude pruning and, in the case of filter pruning, on an $L_2$-norm based criterion. Please note that we cited Tanaka et al. in this context in Appendix C.2.
> > > Point 2 of the major takeaways is awkwardly phrased. If there is more evidence supporting a linear LR schedule, why not come out and say so? Putting it in parentheses weakens the idea.
> >
> > This sentence is in fact more defensively phrased than necessary and we will change it. Thank you for pointing that out. The initial intention of this phrasing was to highlight that this explains the success of CLR as stated in the following sentence.
> >
> > > The first sentence after stating the major takeaways states the authors' claim that retraining should be considered in a budgeted scenario, but at this stage, It’s not clear why.
> >
> > The content of the two bullet points of the major takeaways is that the findings of Li et al. w.r.t. to the learning rate schedule in a budgeted training context almost perfectly resemble the development and improvement of retraining schedules in the context of pruning, that is CLR $>$ SLR $>$ LRW $>$ FT. We believe and show in our submission that it makes sense to view retraining as a budgeted training scenario, resulting in an optimal usage of the limited retraining timeframe.
> >
> >
> > > Figure 2 repeats much of the story of Table 1, which is that the task only gets hard enough to differentiate the methods at 80\% sparsity and up. It is somewhat interesting that there is a noticeable difference in behaviour between the top performers (ALLR, LLR, CLR) between the 80\% sparsity task and the 90\% sparsity task. [...] Is this to be expected?
> >
> > No, this is not expected and does not even align with the previous tables, we thank you for pointing that out. Unfortunately, in Fig. 2 some experiments were not taken into account, leading to incorrect data for sparsities 80\% and 90\%. We have updated the submission with the correct numbers and apologize. Note that for the 90\% plot we also changed the y-axis to emphasize the differences between the top performers. In the 90\% sparsity case ALLR is clearly distinguishable for less than 20 epochs of retraining time, with an exception at 15 epochs. Generally, we think that for high sparsities it is advisable to start with an initially large learning rate to rapidly compensate pruning-induced performance degradation. In the short retraining regime, this however lead to a too large increase for LLR and CLR, a fact that ALLR addresses. For longer amounts of retraining, the three approaches begin to converge.
> >
> > We hope to have addressed your concerns and are happy to discuss if further clarification is needed.
> >
> > **References.**
> >
> > [1] Vrbančič, Grega, and Vili Podgorelec. "Transfer learning with adaptive fine-tuning." IEEE Access 8 (2020): 196197-196211
> >
> > [2] Li, Hao, et al. "Rethinking the hyperparameters for fine-tuning." arXiv preprint arXiv:2002.11770 (2020)

---

> > > ### Comment · Reviewer_Ybn7 · 2022-11-22
> > > **Response to authors continued**
> > >
> > > Thanks for taking the time to attend to my review.
> > >
> > > > Thank you for pointing this out. We have changed (the definition of $d_1$) accordingly.
> > >
> > > I don't see that in the updated manuscript.  The definition of $d_1$ still suffers from this edge case in the updated manuscript I'm reading, and moreover there's a typo in the definition of the interval of the fraction of weights pruned: $s \in ]0,1]$ should read $s \in (0, 1]$.  Maybe an oversight in the revisions?
> > >
> > > >The content of the two bullet points of the major takeaways is that the findings of Li et al. w.r.t. to the learning rate schedule in a budgeted training context almost perfectly resemble the development and improvement of retraining schedules in the context of pruning, that is CLR  SLR  LRW  FT. We believe and show in our submission that it makes sense to view retraining as a budgeted training scenario, resulting in an optimal usage of the limited retraining timeframe.
> > >
> > > The major takeaways points are now *much* clearer.
> > >
> > > > Generally, we think that for high sparsities it is advisable to start with an initially large learning rate to rapidly compensate pruning-induced performance degradation. In the short retraining regime, this however lead to a too large increase for LLR and CLR, a fact that ALLR addresses. For longer amounts of retraining, the three approaches begin to converge.
> > >
> > > I thank the authors for this inclusion, it is a valuable observation.  I wonder if this can help explain the inconsistent result in Table 16, or if another phenomenon leads to the inconsistent results?
> > >
> > >
> > > Overall, I'm pleased that the authors have  improved the clarity of their writing and the breadth of their experiments during the rebuttal period, and am increasing my score accordingly.

---

> > > > ### Author Response · Authors · 2022-12-12
> > > > **Reply to Reviewer response**
> > > >
> > > > Thanks for your fast reply! And thanks again for pointing out the edge case in the definition of $d_1$, it will be changed to standard notation in the next revision.
> > > >
> > > > We agree that there is no clear 'winner' for the translation experiment (Table 16) in the high sparsity and long retraining time regime. Especially CLR and LLR are fairly close for all settings of that experiment, in contrast to the other results. We had a closer look and ran some additional experiments to further understand this: the decisive aspect appears to be that for the experiments of Table 16 the relative budgets are significantly larger (i.e. 20-60\% of the original training) than in all other experiments. This is mainly due to the fact that our code previously only allowed us to specify entire epochs for the retraining time and the overall training time was comparatively short at only five epochs. We have updated the code and run experiments using smaller budgets as well. The results track much more closely with the rest of the paper: for smaller budgets, the advantage of ALLR over its competitors becomes larger and larger, with an improvement in BLEU Score of up to 2, while for larger retraining budgets the approaches start to converge (since the initial learning rate value of ALLR will by definition of $d_1$ and $d_2$ more closely resemble that of LLR).
> > > >
> > > > We will include a discussion of this in the next revision of our paper!

---

> > ### Comment · Reviewer_Ybn7 · 2022-11-22
> > **Response to authors**
> >
> > >Nevertheless, since there is indeed some similarity between the two problems, we think that this further supports a central point we are trying to make, namely that, similarly to the finetuning problem you suggest, the retraining phase is first and foremost a training phase which requires a suitable chosen learning rate schedule. Consequently, we show that given the limited timeframe it is desirable to choose a linear one and given the special setting of different timeframes and different pruning impacts we propose to adaptively choose the initial value, resulting in ALLR.
> >
> > Yes!  I was really hoping that the authors would take my prompt and refine their arguments for what motivates ALLR, and I'm glad to read they've done so here.
> >
> > >We absolutely agree with you on this issue. To strengthen our claims, we used the review- and rebuttal-phase to extend our experiments by a large body of experiments, including MaxVisionTransformers on ImageNet, Semantic Segmentation on the benchmark Coco and Cityscapes datasets as well as Neural Machine Translation on WMT16 (DE-EN). Please see the official comment for a more detailed account.
> >
> > Excellent, this greater diversity of tasks strengthens the paper's arguments.  Tables 10 through 15 show a convincing advantage for CLR, LLR, ALLR as sparsity increases.  But I think the most important experiment added is the translation experiment (results in Table 16), which cut against the main conclusion of the other vision task experiments, and shows that for the translation task, no one method is universally better as the model sparsity constraint increases.  There's surely a lesson to be extracted here, I would have loved to see some discussion space reserved for this contrasting result.

---

### Author Response · Authors · 2022-11-16
**Official comment regarding the current revision**

We thank all reviewers for their interest in our work and their insightful comments and constructive feedback, which helped us to further improve our work. This comment contains an overview of changes in the revision of our work, addressing the two most common concerns.

**Generalization to other learning domains and architectures.**
Throughout the review- as well as rebuttal-phase we extended our computational study with further experiments, adding more architectures and datasets in order to show that our results transfer to other domains.

- We complemented our experiments on ImageNet-1k using ResNet-50 with an additional network architecture, namely the Vision-Transformer model *MaxViT*. Our results contain both the weight as well as filter pruning case, each covering the OneShot and iterative pruning cases. Similar to the results presented in our original submission, we observe large advantages of ALLR over the other approaches (cf. Table 10, 11, 20, 21).
- We performed experiments on the neural machine translation dataset WMT-16 (DE-EN) using a T5 transformer architecture, limiting ourselves to the OneShot pruning case given the rebuttal timeframe. The results can be found in Table 16 in the appendix, where we report the BLEU score on the test set. Similarly, our claims are also supported in this setting.
- We included the semantic segmentation and object detection dataset COCO, on which we trained the DeepLabV3 architecture. The retraining schedule comparison for OneShot- and iterative weight pruning can be found in Tables 12 and 13 in the appendix, where we report the mean IoU (Intersection over Union) metric.
- We similarly added the common semantic segmentation benchmark CityScapes, learned by a PSPNet architecture. Results can be found in Tables 14 and 15.

The revision contains the results of these experiments (all in the appendix). We changed the text in the main body accordingly to reflect the changed experimental setup. The new results further show that the approach we propose is effective for a wide range of different architectures and learning tasks.

**Motivation for ALLR.**
The intuition behind our adaptive scheme was critized for not being as clear as we intended it to be. Our goal was to incorporate both the length of retraining (reflected by $d_2$) as well as the impact of pruning into the choice of an initial learning rate value. For the latter, a direct way would be to simply measure the performance degradation on a hold-out set, which was our initial thought and has now further been suggested and emphasized by the reviewers. Nevertheless, we decided to take the drop in $L_2$-norm as a means for measuring the pruning-induced distortion of the model, quantifying the intuition that weights that are already close to zero will have a minor impact on the model output when pruned. To account for the former approach and its differences to our approach more explicitly, we changed the ablation study in Appendix C.1 to further contain results on ImageNet, which give a more clear picture. Given these results, we think that both approaches have their strengths and weaknesses. What we term 'AccALLR', an ALLR-variant which considers the performance degradation as measured by the drop in train accuracy, seems to give a more precise estimator for a good initial learning rate value in the medium sparsity regime, surpassing the results of ALLR. On the other hand, in the higher sparsity regime, ALLR seems to be more robust in the sense that the performance easily degradates to that of a random classifier, essentially resulting in 'AccALLR' selecting the largest possible learning rate as LLR and CLR do, resulting in an overshooting and performance deficit compared to ALLR. We thus think it is best to stick to our proposed criterion: it is simple, efficiently computable and does not require additional evaluation passes, which could potentially be viewed as unfair in the context of comparing BIMP to other pruning-stable methods. Given our experimental results, we think that the drop in $L_2$-norm as measured in Eq. (1) serves as a good measure for the impact of pruning and clearly, as we were able to show empirically, as a good indicator for the initial value of the learning rate.


**Other updates to the manuscript.**
Thanks to reviewer Ybn7, we noticed that the plots for sparsities 80\% and 90\% in Figure 2 were not correct and we updated them accordingly.

---

### Decision · Program_Chairs · 2023-01-20

**Decision:**

Accept: poster

**Justification For Why Not Higher Score:**

This is a solid contribution but cannot say currently it is high impact, but at the same time hard to rule out.

**Justification For Why Not Lower Score:**

N/A

**Metareview: Summary, Strengths And Weaknesses:**

The paper proposes a simple and efficient method for improving iterative prunning. The method focuses on tuning the learning rate schedule during the retraining phase, which has been shown to significantly impact the performance of pruned networks. The experiments show improvement upon some much more complex methods.

The simplicity of the method and its impact on the robustness of the field are the main strong sides. However, some reviewers noted a lack of clarity in the writing and poor motivation behind the learning rate scheduling.

After discussion, all reviewers were supportive of accepting the paper. All in all, it is my pleasure to recommend acceptance. Thank you for your submission. Please make sure to address the authors’ feedback in the camera-ready version.

**Note From Pc:**

if the above contains the word "oral" or "spotlight" please see: "oral" presentation means -> notable-top-5% and "spotlight" means -> notable-top-25%. As stated in our emails, we are disassociating presentation type from AC recommendations